# Best of Both Worlds Model Selection

**Aldo Pacchiano**
Microsoft Research, NYC
apacchiano@microsoft.com

**Christoph Dann**
Google, NYC
cdann@cdann.net

**Claudio Gentile**
Google, NYC
cgentile@google.com

## Abstract

We study the problem of model selection in bandit scenarios in the presence of nested policy classes, with the goal of obtaining simultaneous adversarial and stochastic ("best of both worlds") high-probability regret guarantees. Our approach requires that each base learner comes with a candidate regret bound that may or may not hold, while our meta algorithm plays each base learner according to a schedule that keeps the base learner's candidate regret bounds balanced until they are detected to violate their guarantees. We develop careful mis-specification tests specifically designed to blend the above model selection criterion with the ability to leverage the (potentially benign) nature of the environment. We recover the model selection guarantees of the CORRAL [3] algorithm for adversarial environments, but with the additional benefit of achieving high probability regret bounds. More importantly, our model selection results also hold simultaneously in stochastic environments under gap assumptions. These are the first theoretical results that achieve best of both world (stochastic and adversarial) guarantees while performing model selection in contextual bandit scenarios.

## 1 Introduction

A fundamental challenge in sequential decision-making is the ability of the learning agent to adapt to the unknown properties of the environment they interact with. While adversarial environments may require some caution, we would also like to leverage situations where more benign scenarios may be disclosed as a result of this interaction.

In the literature on multi-armed bandits, this adaptation capabilities has often taken two forms:
**1. Best-of-both worlds guarantees**, which were pioneered by [10] and subsequently studied by a number of authors [e.g., 29, 5, 28, 30, 33, 21]. Here, the goal is to design algorithms achieving both stochastic and adversarial environment guarantees simultaneously, without knowing the type of environment in advance. Similar in spirit is the stream of literature on stochastic rewards with adversarial corruptions [23, 17, 34, 21, 31], where an adversary is assumed to corrupt the stochastic rewards observed by the algorithm, and the regret guarantees are expected to degrade gracefully with the total amount of corruption, without knowing this amount in advance.
**2. Model selection guarantees**, which were initiated by [3], and subject since then to intense investigations [e.g., 15, 1, 27, 4, 16, 12, 8, 14, 22, 26, 13]. Here, we assume we have access to a pool of $M$ base bandit algorithms each operating, say, within a different class of models or under different assumptions on the environment, and the aim is to design a bandit meta-algorithm that learns to simulate the best base algorithm in hindsight, without knowing in advance which one will be best for the environment at hand. This approach has often been used in the literature to capture bandit model selection problems. In fact, a natural instantiation of this framework is when we have a sequence of nested policy classes, and the goal is to single out the best policy within this nested family, by paying as price only the complexity of the policy class the optimal policy falls into. In some sense, Item 2 is more general than Item 1, since one may attempt to achieve a best-of-both-world performance by pooling a stochastic bandit algorithm with an adversarial bandit one, and expect the meta-algorithm on top of them to eventually learn to follow one of the two. Similarly, in the setting of

36th Conference on Neural Information Processing Systems (NeurIPS 2022).

stochastic rewards with adversarial corruptions, one can pool together stochastic algorithms operating with increasing guessed levels of corruptions, and let the meta-algorithm learn to single out the one corresponding to the true corruption level.[1]

In this paper, we combine the two items above into a bandit algorithm that exhibits both model selection and best-of-both worlds regret guarantees simultaneously. Our framework encompasses in particular a well-known linear bandit model selection scenario, where the action set $\mathcal{A}$ is a (finite but large) subset of $\mathbb{R}^{d_M}$, for some maximal dimension $d_M > 0$, and we deal with a hierarchy of possible dimensions $d_1 < \ldots < d_M$. At time $t$, the learner plays an action $\mathbf{a}_t \in \mathcal{A}$ and receives as reward either $r_t = \mathbf{a}_t^\top \boldsymbol{\omega}_t$, where $\boldsymbol{\omega}_t \in \mathbb{R}^{d_M}$ is an adversarially-generated reward vector (adversarial setting), or $r_t = \mathbf{a}_t^\top \boldsymbol{\omega}$ + white noise, where $\boldsymbol{\omega} \in \mathbb{R}^{d_M}$ is a fixed but unknown reward vector (stochastic setting). Associated with dimensions $d_1 < \ldots < d_M$ are a nested family of policy classes $\Pi_1 \subset \ldots \subset \Pi_M$, and $M$ base algorithms $\mathbb{A}_1, \ldots, \mathbb{A}_M$, where the $i$-th algorithm $\mathbb{A}_i$ is a linear bandit algorithm that works under the assumption that only the first $d_i$ components of $\boldsymbol{\omega}_t$ (or $\boldsymbol{\omega}$) are nonzero. Hence $\mathbb{A}_i$ operates with dimension $d_i$ in that all policies $\pi \in \Pi_i$ are probability distributions which are projections over action set $\mathcal{A}_i$ of the set of probability distributions in $\Pi_M$. Here, $\mathcal{A}_i$ is the projection of the full-dimensional action set $\mathcal{A}$ onto its first $d_i$ dimensions. Notice that this implies that the policy classes are nested: $\Pi_1 \subseteq \Pi_2 \ldots \subseteq \Pi_M$. If only the first $d_{i_\star}$ dimensions ($i_\star$ being unknown to the learner) of each $\boldsymbol{\omega}_t$ (adversarial setting) or $\boldsymbol{\omega}$ (stochastic setting) are nonzero, we design an algorithm whose regret bound scales as $\text{poly}(d_{i_\star})\sqrt{T}$ in the adversarial case and, simultaneously, as $\frac{\text{poly}(d_M)\log T}{\Delta}$ if the environment happens to be stochastic. Specific consideration is given to the dependence on $d_{i_\star}$. We show through a lower bounding argument that in the stochastic case a guarantee of the form $\frac{d_M \log T}{\Delta}$, that is, where $d_M$ replaces the more desirable factor $d_{i_\star}$, is *inevitable*, if we want to insist on obtaining a $\frac{\log T}{\Delta}$-like result. Thus our bounds reflect the best achievable regret rates depending on $d_M$ in the stochastic case and on $\text{poly}(d_{i_\star})$ in the adversarial case.

In order to achieve these best-of-both-world results, one cannot easily rely on general corralling techniques, like the one contained in [3], since the granularity offered by adversarial aggregation algorithms is no better than $\sqrt{T}$, which is not adequate for stochastic settings; hence our choice towards the model selection technique known as *regret balancing*. Yet, even in this case, the literature does not provide off-the-shelf solutions: We first need to extend regret balancing from stochastic rewards [13, 26, 1] and corrupted stochastic rewards [31] to adversarial rewards. This involves adding extra actions to the action space at a given level of the hierarchy so as to enable the base learner operating at that level to compete with higher levels. This seems to be needed because in the adversarial case, base algorithms may even incur a negative regret, thus making it hard to compare the relative performance of algorithms operating at different levels during the regret balancing operations. Specific technical hurdles arise in the linear bandit case, where mis-specification may cause low-level base learners to behave in a maliciously erratic way. In this case, mis-specification tests have to be designed with care so as to ensure that base learners that are mis-specified but not yet eliminated incur manageable regret. At the same time, these tests should also detect as soon as possible if the environment is stochastic through ad hoc regret balancing gap estimation procedures.

For our results to hold, some technical conditions are required on the base algorithms, like a notion of (high probability) stability and a notion of action space extendability, formally defined in Section 2. We show that these conditions are fulfilled by known algorithms, like an anytime variant of the GeometricHedge.P algorithm in [7], and a high-probability variant of the EXP4 algorithm in [6].

**Related work.**     Among the references we mentioned above, those which are most relevant to our work, as directly related to model selection and best-of-both-worlds guarantees are perhaps [4, 13, 21, 31]. In both [4] and [13], model selection regret guarantees for stochastic contextual bandits are given which take the form $\text{poly}(d_{i_\star})\sqrt{T}$. Yet, no combination of best-of-both worlds and model selection results are contained in those papers. Closer in spirit are [21] and [31].

In [21], the authors consider model selection problems on top of adversarially corrupted stochastic linear bandit problems, where the total level of corruption $C$ is unknown in advance. It is worth stressing that this is a model selection problem which is substantially different (and actually easier, see Section 4) than ours, as the selection applies to $C$ instead of the complexity term $R(\Pi_i)$. In fact, the model selection procedure in [21] looks very different from ours, and can be roughly seen as

---

[1]Recent relevant papers, working with model mis-specification instead of reward corruption, include [27, 14].

a robust version of a doubling trick applied to $C$. Our algorithm is more general than theirs, as it applies to settings beyond linear bandits, and the same is true of our best of both worlds procedures.

In [31], the authors also consider RL settings, and more general forms of corruptions than [21]. Like ours, their model selection algorithm follows the idea of regret balancing and elimination of [1, 13]. Yet, importantly our work applies to the fully adversarial setting with $\sqrt{T}$ regret while the corruption-robust approach by [31] suffers a linear in $T$ regret if the world is fully adversarial ($C = \Omega(T)$). To enable this sublinear regret, several important technical challenges needed to solved that are not present in the corruption setting. Most importantly, the corruption setting can, for the most part, be handled similar to the stochastic setting. For instance, since [31] measures regret always w.r.t. to the uncorrupted environment, by definition, no learner can achieve negative regret, which entails that setting does not require linking base learners as we do here – see Section 2.

On the lower bound side, relevant papers include [32, 24], where the authors investigate the Pareto frontier of model selection for (contextual) stochastic bandits. It is shown in particular that a regret upper bound of the form $\sqrt{d_{i_\star} T}$ cannot be achieved. However, these papers do not explicitly cover gap-dependent regret guarantees. A more thorough discussion of our contributions is postponed to Section 3, after introducing our main notation and assumptions.

## 2 Problem Setting and Assumptions

We start off by defining the adversarial scenario. An adversarial contextual bandit problem is a repeated game between a learner $\mathbb{A}$ and an environment $\mathbb{B}$. We consider the general decision-making scenario where the learner $\mathbb{A}$ has at its disposal a class of policies $\Pi_{\mathbb{A}}$ made up of functions $\pi$ of the form $\pi : \mathcal{X} \to \Delta_{\mathcal{A}}$, where $\Delta_{\mathcal{A}}$ denotes the set of probability distributions over $\mathcal{A}$. At each round $t$, the interaction between $\mathbb{A}$ and $\mathbb{B}$ is as follows:

1. Learner $\mathbb{A}$ selects a policy $\pi_t \in \Pi_{\mathbb{A}}$
2. Simultaneously, the environment $\mathbb{B}$ selects context $x_t \in \mathcal{X}$ and reward function $r_t : \mathcal{A} \times \mathcal{X} \to [0, 1]$, and reveals $x_t$ to $\mathbb{A}$
3. Learner $\mathbb{A}$ takes an action $a_t \sim \pi_t(\cdot \,|\, x_t)$ and observes reward $o_t = r_t(a_t, x_t)$.

The regret $\mathrm{Reg}_{\mathbb{A}}(\mathcal{T}, \Pi')$ of algorithm $\mathbb{A}$ in rounds $\mathcal{T} \subseteq \mathbb{N}$ against a policy class $\Pi'$ is the difference between the learner's accumulated reward in $\mathcal{T}$ and the performance of the best *fixed* policy from $\Pi'$:

$$\mathrm{Reg}_{\mathbb{A}}(\mathcal{T}, \Pi') = \max_{\pi \in \Pi'} \sum_{\ell \in \mathcal{T}} \left[ \mathbb{E}_{a \sim \pi} \left[ r_\ell(a, x_\ell) \right] - r_\ell(a_\ell, x_\ell) \right] \;,$$

where $\mathbb{E}_{a \sim \pi}[r_\ell(a, x_\ell)] = \sum_{a \in \mathcal{A}} \pi(a|x_\ell) r_\ell(a, x_\ell)$ denotes the expectation over the action drawn from the given policy $\pi$ which may itself be a random quantity. For ease of notation, we will omit the comparator policy class when $\Pi' = \Pi_{\mathbb{A}}$ is the policy class of $\mathbb{A}$ and replace $\mathcal{T}$ by $t$ when $\mathcal{T} = [t] := \{1, \ldots, t\}$. Hence, $\mathrm{Reg}_{\mathbb{A}}(t)$ is the regret of $\mathbb{A}$ against its own policy class up to round $t$.

The stochastic scenario we consider is similar to the above, except in the way contexts $x_t$ and reward values $r_t(a_t, x_t)$ are generated. Specifically, $\mathbb{B}$ generates contexts $x_t$ in an i.i.d. fashion according to a fixed (but arbitrary and unknown) distribution $\mathcal{D}$ over $\mathcal{X}$, while the reward $r_t(a_t, x_t)$ is such that for all fixed $(a, x) \in \mathcal{A} \times \mathcal{X}$, we have $\mathbb{E}[r_t(a, x)] = r(a, x)$, for some fixed function $r(\cdot, \cdot)$ in some known class of reward functions. In this case, we measure performance through pseudo-regret

$$\mathrm{PseudoReg}_{\mathbb{A}}(\mathcal{T}, \Pi') = \max_{\pi \in \Pi'} \sum_{\ell \in \mathcal{T}} \left[ \mathbb{E}_{x \sim \mathcal{D}, a \sim \pi} \left[ r(a, x) \right] - \mathbb{E}_{x \sim \mathcal{D}, a \sim \pi_\ell} \left[ r(a, x) \right] \right] \;,$$

where $\mathbb{E}_{x \sim \mathcal{D}, a \sim \pi}$ is the expectation over contexts $x$ and the action $a \sim \pi(\cdot|x)$ drawn from a policy $\pi$ that may itself be a random quantity. We say an environment $\mathbb{B}$ is stochastic with gap $\Delta > 0$ if there is a policy $\pi_\star \in \Pi$ such that

$$\mathbb{E}_{x \sim \mathcal{D}, a \sim \pi_\star} \left[ r(a, x) \right] \geq \max_{\pi \in \Pi \setminus \{\pi^*\}} \mathbb{E}_{x \sim \mathcal{D}, a \sim \pi} \left[ r(a, x) \right] + \Delta \;.$$

A notable example of the above is the following linear bandit scenario. In the adversarial case ("adversarial linear bandits") the action space $\mathcal{A}$ is a subset of $\mathbb{R}^d$, for some dimension $d$, the context $x_t$ is irrelevant and, upon playing action $\mathbf{a}_t \in \mathcal{A}$, the environment generates a reward which is a *linear* function of the actions, $r_t(\mathbf{a}_t, x_t) = r_t(\mathbf{a}_t) = \mathbf{a}_t^\top \boldsymbol{\omega}_t$, where $\boldsymbol{\omega}_t$ is chosen adversarially at every round

within some known class of $d$-dimensional vectors. In the stochastic case ("stochastic linear bandits"), the only difference is that we simply have $r_t(\mathbf{a}_t, x_t) = r_t(\mathbf{a}_t) = \mathbf{a}_t^\top \boldsymbol{\omega} + \epsilon_t$, where $\boldsymbol{\omega}$ is a fixed and unknown vector in the class of $d$-dimensional vectors, and $\epsilon_t$ is a sub-Gaussian noise. Moreover, we have gap $\Delta$ if there is $\pi^\star \in \Pi$ such that $\mathbb{E}_{a \sim \pi^\star} \left[ \mathbf{a}^\top \boldsymbol{\omega} \right] \geq \max_{\pi \in \Pi \setminus \{\pi^\star\}} \mathbb{E}_{a \sim \pi} \left[ \mathbf{a}^\top \boldsymbol{\omega} \right] + \Delta$ .

Many algorithms $\mathbb{A}$ in the literature exist that enjoy a regret bound against their policy class $\Pi_\mathbb{A}$, holding in all environments $\mathbb{B}$ satisfying certain conditions. If this regret bound holds, we say that $\mathbb{A}$ is *adapted* to this environment. Here, we allow algorithms to also compete against policy classes $\Pi'$ other than $\Pi_\mathbb{A}$, and focus on algorithms that come with regret bounds of the following form.

**Definition 1** (Adapted). *We call $\mathbb{A}$ adapted to environment $\mathbb{B}$ and policy class $\Pi'$ if, with probability at least $1 - \delta$, the following bound holds simultaneously for all $t \in \mathbb{N}$:*[2]

$$\mathrm{Reg}_\mathbb{A}(t, \Pi') = \mathcal{O}\left( R(\Pi_\mathbb{A})\sqrt{t \ln \frac{t}{\delta}} \right) \ .$$

*The term $R(\Pi_\mathbb{A})$ is a (known) measure of complexity of the policy class $\Pi_\mathbb{A}$ used by $\mathbb{A}$.*

Examples of algorithms satisfying Definition 1 include a version of the Geometric Hedge algorithm in linear bandits [7] and the EXP4 algorithm [6] in contextual bandits with finite action sets that operates over policies mapping contexts to probability distributions over actions. We will discuss them in detail below. Before introducing the model selection questions addressed in this work, we present two stronger versions of the above definition, which we call *high-probability stability* and *extendability*. These will be useful for model selection among multiple learners. As we will show with examples later, these stricter conditions can be established for several common settings under the same conditions for which adaptivity from Definition 1 is guaranteed.

**Additional conditions.** The first condition, *high-probability stability* (or *h-stability*) generalizes adaptivity to the case where $\mathbb{A}$ only observes a certain noisy version of the reward. To state the condition formally, consider a more general interaction protocol between $\mathbb{A}$ and the environment $\mathbb{B}$, where Step 3 above is replaced by

3a. Let $b_t \sim \mathrm{Bernoulli}(\rho)$ for a fixed and known $\rho \in (0, 1]$. Learner $\mathbb{A}$ takes an action $a_t \sim \pi_t(\cdot \mid x_t)$ and observes an importance-sampled version of the reward $o_t = b_t \frac{r_t(a_t, x_t)}{\rho}$ .

This encompasses the original protocol with $\rho = 1$ as a special case. An algorithm is *h-stable* if it maintains its regret guarantee up to a $1/\sqrt{\rho}$ penalty in this more general interaction protocol:

**Definition 2** (h-stability). *An algorithm $\mathbb{A}$ is high probability stable (**h-stable**) in an environment $\mathbb{B}$ against policy class $\Pi'$ if it satisfies for any constant $\rho \in (0, 1]$ a regret bound*

$$\mathrm{Reg}_\mathbb{A}(t, \Pi') = \mathcal{O}\left( R(\Pi_\mathbb{A})\sqrt{\frac{t}{\rho} \ln \frac{t}{\delta}} \right)$$

*that holds with probability at least $1 - \delta$ simultaneously for all $t \in \mathbb{N}$.*

Note that $\mathbb{A}$ is adapted to $\mathbb{B}$ and $\Pi'$ if it is h-stable for $\rho = 1$. Compared to the notion of stability proposed in [3], the one in Definition 2 is more demanding, in that it requires the importance-weighted regret bound to hold with high probability, rather than in expectation. Since we aim for high-probability regret bounds in adversarial and stochastic settings, this stronger notion is natural.

The second condition, *extendability*, generalizes h-stability to environments with additional actions, as specified next. Let $\mathbb{B}$ be the original environment with action set $\mathcal{A}$ and let $\bar{\mathcal{A}}_k = \mathcal{A} \cup \{a'_1, a'_2, \ldots, a'_k\}$ be the action set extended by $k$ extra special actions $a'_1, a'_2, \ldots, a'_k$. Further, let $\bar{\Pi} = \{\pi : \mathcal{X} \to \Delta_{\bar{\mathcal{A}}_k}\}$ be an extended version of the original policy set $\Pi = \{\pi : \mathcal{X} \to \Delta_\mathcal{A}\}$ that contains all policies of the original policy set and the single-action policies $\mathbf{1}\{a'_i\}$ that always choose a certain special action $a'_i$, i.e., $\bar{\Pi} \supseteq \Pi \cup \{\mathbf{1}\{a'_1\}, \mathbf{1}\{a'_2\}, \ldots, \mathbf{1}\{a'_k\}\}$.[3] We further allow the environment on the extended action space $\bar{\mathcal{A}}_k$ to choose any values for rewards $r_t(a'_i, x_t)$, possibly depending on the entire history and all rewards $r_t(a, x_t)$ assigned to other actions $a \neq a'_i$ in that round. We denote the set of all such extended environments by $\mathcal{B}_k(\mathbb{B}, \bar{\mathcal{A}}_k)$. An algorithm $\mathbb{A}$ is now extendable if we can run it with the extended policy set $\bar{\Pi}_k$ and it competes well against $\Pi'$ and policies $\mathbf{1}\{a'_i\}$ in all such extended environments.

---

[2]Here and throughout, the $\mathcal{O}$-notation only hides absolute constants.

[3]Policies $\pi \in \Pi$ are naturally extended from $\Delta_\mathcal{A}$ to $\Delta_{\bar{\mathcal{A}}_k}$ by assigning probability 0 to all special actions $a'_i$.

**Definition 3** (Extendability). *Consider an algorithm $\mathbb{A}$ with policy set $\Pi_{\mathbb{A}} \subseteq \mathcal{X} \to \Delta_{\mathcal{A}}$ in environment $\mathbb{B}$. We call $\mathbb{A}$ $k$-extendable in $\mathbb{B}$ against $\Pi'$ if there is an extended policy set $\bar{\Pi}_k \supseteq \Pi_{\mathbb{A}} \cup \{\mathbf{1}\{a_1'\}, \ldots \mathbf{1}\{a_k'\}\}$ such that $\mathbb{A}$ equipped with the extended policy set $\bar{\Pi}_k$ is h-stable against $\Pi' \cup \{\mathbf{1}\{a_1'\}, \ldots \mathbf{1}\{a_k'\}\}$ in all environments $\mathbb{B}' \in \mathcal{B}(\mathbb{B}, \bar{\mathcal{A}}_k)$ that extend $\mathbb{B}$ from action space $\mathcal{A}$ to action space $\bar{\mathcal{A}}_k = \mathcal{A} \cup \{a_1', \ldots, a_k'\}$.*

One relevant example of h-stable and extendable algorithm working in the adversarial linear bandit scenario is an anytime variant of the Geometric Hedge algorithm from [7] with exploration ruled by John's ellipsoid (e.g., [9]) – see Appendix B. Another example is a high-probability variant of the EXP4 algorithm from [6], which operates with finite sets of policies.

## 2.1 Model selection and best-of-both-worlds regret

Our model selection for best-of-both worlds regret guarantees can be described as follows. We have a nested family of policy classes $\Pi_1 \subseteq \ldots \subseteq \Pi_M$, with (known) complexities $R(\Pi_1) \leq \ldots \leq R(\Pi_M)$. A meta-learning algorithm $\mathbb{M}$ has access to $M$ base algorithms $\mathbb{A}_1, \ldots, \mathbb{A}_M$, algorithm $\mathbb{A}_i$ operating with policy class $\Pi_i$. These algorithms we sometimes refer to as *base learners*. Let $i_\star$ be the smallest index of the base learner that competes against the largest policy class $\Pi_M$ in the following sense:

$$i_\star = \min\left\{i \in [M] \colon \mathbb{A}_i \text{ is } (M-i)\text{-extendable and h-stable in } \mathbb{B} \text{ against } \Pi_M\right\} .$$

The goal of model selection is to devise a meta-algorithm $\mathbb{M}$ that has access to the base learners $\mathbb{A}_1, \ldots, \mathbb{A}_M$, and which achieves with probability $1 - \delta$ a regret bound of the form[4]

$$\text{Reg}_{\mathbb{M}}(t, \Pi_M) = \mathcal{O}\left(\text{poly}\left(M, R(\Pi_{i_\star}), \ln R(\Pi_M)\right)\sqrt{t \ln \frac{t}{\delta}}\right) ,$$

holding for all $t$, whenever $\mathbb{A}_M$ is h-stable and the environment $\mathbb{B}$ is adversarial. Simultaneously, if $\mathbb{B}$ is stochastic with gap $\Delta$, then we must have

$$\text{PseudoReg}_{\mathbb{M}}(t, \Pi_M) = \mathcal{O}\left(\frac{\text{poly}(M, R(\Pi_M))}{\Delta} \log \frac{t}{\delta}\right) .$$

Notice that the above requirement on the pseudo-regret in stochastic environments only requires a dependence on $R(\Pi_M)$, instead of $R(\Pi_{i_\star})$. This is motivated by the fact that in stochastic environments with gap $\Delta$ it is generally impossible to obtain model selection guarantees of the form $\frac{R(\Pi_{i_\star}) \log t}{\Delta}$ – see Appendix A for a proof of this claim.

# 3 Summary of Our Contributions and Discussion

Our contributions can be summarized as follows. (i) We introduce an algorithm, called Arbe (Algorithm 1), for high probability model selection. This is the first high-probability model selection result for adversarial contextual bandit algorithms. Arbe satisfies a high probability guarantee as long as each of the base algorithms also satisfies one. Our algorithm takes inspiration from the balancing and elimination techniques in [13], which have been designed for stochastic contextual bandit (and RL) scenarios. Yet, as mentioned above, several technical hurdles had to be overcome in the algorithm's design to make it usable with adversarial base learners. We believe the model selection rates that our algorithm achieves are minimax optimal in their dependency on $R(\Pi_{i_\star})$. Recall that, as shown by the lower bounds of [24], it is not possible to achieve a model selection rate scaling linearly with $R(\Pi_{i_\star})$ – the best one can hope for is a quadratic dependence on $R(\Pi_{i_\star})$. In the setting of adversarial linear bandits this turns into a rate with a multiplier of the form $d_{i_\star} \log(\mathcal{A})$ instead of $\sqrt{d_{i_\star} \log(\mathcal{A})}$. In particular, when $|\mathcal{A}| = \Omega(2^{d_{i_\star}})$ or[5] $|\mathcal{A}| = \infty$, this multiplier becomes $d_{i_\star}^2$. In the simpler case of EXP4 base learners, this multiplier takes the form $\log(\Pi_{i_\star})$ instead of $\sqrt{\log(\Pi_{i_\star})}$.

(ii) We introduce the first algorithm for best-of-both-worlds model selection. By leveraging the high-probability guarantees of Arbe, we develop the first best-of-both-worlds model selection algorithm

---

[4]Here, $\text{poly}(a, b, c)$ is a polynomial function of the three arguments separately.

[5]Despite we do not explicitly work out the details here, applying our results to the infinite arm case for adversarial linear bandits can be accomplished by a standard covering argument at each dimension $d_i$. This turns factor $\log(\mathcal{A})$ into one of the form $d_{i_\star} \log T$.

---

**Algorithm 1:** Arbe$(\delta, s = 1, t_0 = 0)$ Adversarial Regret Balancing and Elimination

---
**1** **Input:** initial time $t_0$, index $s$ of smallest active base learner, failure probability $\delta$

**2** Initialize base learners $\mathbb{A}_s, \mathbb{A}_{s+1}, \ldots, \mathbb{A}_M$ with extended policy classes $\tilde{\Pi}_s, \tilde{\Pi}_{s+1}, \ldots, \tilde{\Pi}_M$

**3** Set sampling probabilities $\rho_i = \frac{R(\tilde{\Pi}_i)^{-2}}{\sum_{j=s}^{M} R(\tilde{\Pi}_j)^{-2}}$ for all $i \in \{s, \ldots, M\}$, and $\rho_i = 0$ for $i < s$

**4** **for** *round* $t = t_0 + 1, t_0 + 2, \ldots$ **do**

**5**      Sample base learner index $b_t \sim \text{Categorical}(\rho_1, \cdots, \rho_M)$

**6**      Get context $x_t$ and compute $a_t^i \sim \pi_t^i(\cdot|x_t)$, the action each base learner $\mathbb{A}_i$ proposes for $x_t$

**7**      Play action $a_t = a_t^{b_t}$ (resolve linked actions if necessary) and receive reward $r_t(a_t, x_t)$

**8**      Update all base learners $\mathbb{A}_i$ with reward $\frac{\mathbf{1}\{b_t = i\} r_t(a_t, x_t)}{\rho_i}$

**9**      Set

$$\widetilde{\text{CRew}}_i(t_0, t) = \sum_{\ell = t_0 + 1}^{t} \frac{\mathbf{1}\{b_\ell = i\} r_\ell(a_\ell, x_\ell)}{\rho_i}, \quad D_i(t_0, t) = \mathcal{O}\left(\sqrt{\frac{t - t_0}{\rho_i} \ln \frac{\ln t}{\delta}} + \frac{1}{\rho_i} \ln \frac{\ln t}{\delta}\right)$$

$$(1)$$

     Test for all $i, j \in \{s, \ldots, M\}$ with $i < j$:

$$\widetilde{\text{CRew}}_j(t_0, t) > \widetilde{\text{CRew}}_i(t_0, t) + D_i(t_0, t) + D_j(t_0, t) + R(\tilde{\Pi}_i)\sqrt{\frac{t - t_0}{\rho_i} \ln \frac{t}{\delta}} \quad (2)$$

**10**      **if** test triggers for $\mathbb{A}_i$ **then restart** algorithm by running Arbe$(\delta, i + 1, t)$

---

(Arbe-Gap + Arbe-GapExploit, Algorithm 2 + Algorithm 4) that can retain model selection rates when the environment is adversarial, and obtain logarithmic rates when the environment is stochastic with a gap. The logarithmic gap-dependent rate of this algorithm exhibits an optimal quadratic dependence w.r.t. the largest policy class $R(\Pi_M)$ (see also Item (iii) below). Our algorithm is quite complex and requires a couple of main innovations: First, a careful design of a gap identification subroutine aimed at identifying a candidate optimal policy and gap estimator (Arbe-Gap) and, second, a very precise schedule of play for exploiting this knowledge and test its truthfulness (Arbe-GapExploit).

(iii) As already mentioned, we show via a lower bound for stochastic environments that, in the presence of a gap, perfect model selection between multiple logarithmic rate learners is impossible. This can be found in Appendix A. A dependence on the complexity of the largest class $R(\Pi_M)$ is inevitable, and this dependence must be quadratic. Our algorithms (Arbe-Gap + Arbe-GapExploit) achieve exactly this dependence when the environment is stochastic and has a gap (Theorem 5).

## 4 Adversarial Model Selection Using Regret Balancing

We now introduce our algorithm for model selection in adversarial bandit problems with high-probability regret guarantees. The algorithm is shown in Algorithm 1, and follows the regret balancing principle. This principle has been applied successfully to model selection in bandit and RL problems with stochastic rewards [13, 26, 1] and with corrupted stochastic rewards [31]. Our work is the first to extend this approach to adversarial rewards.

**Regret Balancing.** In each round, we choose the index of a base learner $b_t$ by sampling from a categorical distribution with probabilities $\rho_s, \rho_{s+1}, \ldots, \rho_M$ that remain fixed throughout the epoch. The policy of learner $\mathbb{A}_{b_t}$ is then used to sample the action $a_t$ that is passed to the environment. After receiving the reward $r_t(a_t, x_t)$, Algorithm 1 updates each base learner $\mathbb{A}_i$ with $r_t(a_t, x_t)$ importance-weighted by the probability that the learner was selected, i.e., $\frac{\mathbf{1}\{b_t = i\} r_t(a_t, x_t)}{\rho_i}$. Thus, we update all base learners. This is closer to the Corral algorithm [3] than to the regret balancing approaches for the stochastic setting, which only update the selected learner $\mathbb{A}_{b_t}$.

If the probabilities are set to $\rho_i \propto \frac{1}{R(\Pi_i)^2}$ then after $t$ rounds, each learner $\mathbb{A}_i$ is followed roughly $\rho_i t$ times. To see why the regrets of all learners are balanced if they are $h$-stable against $\Pi_M$ in this case, consider the following. Denote by $\text{reg}_t(a) = [\mathbb{E}_{a \sim \pi^\star}[r_t(a, x_t)] - r_t(a, x_t)]$ the regret in round $t$ of action $a$ where $\pi^\star \in \text{argmax}_{\pi \in \Pi_M} \sum_{t=1}^{T} \mathbb{E}_{a \sim \pi}[r_t(a, x_t)]$ is the best policy after all rounds $T$. The

regret in rounds where $\mathbb{A}_i$ was selected can be bounded using standard concentration arguments as

$$\sum_{t=1}^{T} \mathbf{1}\{b_t = i\}\operatorname{reg}_t(a_t) = \rho_i \sum_{t=1}^{T} \frac{\mathbf{1}\{b_t = i\}}{\rho_i}\operatorname{reg}_t(a_t^i) \approx \rho_i \sum_{t=1}^{T} \operatorname{reg}_t(a_t^i) \leq R(\tilde{\Pi}_i)\tilde{O}\left(\sqrt{\rho_i T}\right),$$

where the final inequality holds because $\mathbb{A}_i$ is h-stable. From the definition of $\rho_i$ we have $R(\tilde{\Pi}_i)\sqrt{\rho_i} = \left(\sum_{j=s}^{M} R(\tilde{\Pi}_j)^{-2}\right)^{-1/2} \leq R(\tilde{\Pi}_s)$, which shows that the total regret in those rounds is bounded by $\tilde{O}(R(\tilde{\Pi}_s)\sqrt{T})$. Thus, if base learners are all h-stable, then the regret incurred by each of them is comparable to the regret incurred by $\mathbb{A}_s$, the learner with the smallest complexity $R(\Pi_s)$.

**Eliminating Base Learners.** If a learner $\mathbb{A}_i$ is not h-stable and may have linear regret, then the probabilistic schedule above which plays this learner roughly $\rho_i t$ times yields linear regret. We therefore monitor the performance of each learner and terminate the epoch whenever a learner performs significantly worse than expected, and thus cannot be h-stable in the environment. To identify such cases, we compare estimates of the rewards of all pairs of base learners as follows. For each learner $\mathbb{A}_i$, $\widetilde{\operatorname{CRew}}_i(t_0, t)$ in Equation 1 is an unbiased estimate of the learners reward sequence $\operatorname{CRew}_i(t_0, t) = \sum_{\ell=t_0+1}^{t} r_\ell(a_\ell^i, x_\ell)$ (see the appendix for details about how these estimates are computed). Further, using confidence bounds $\mathrm{D}_i(t - t_0)$ from Equation 1, we can constructs a confidence interval for $\operatorname{CRew}_i(t_0, t)$ as $\left[\widetilde{\operatorname{CRew}}_i(t_0, t) - \mathrm{D}_i(t - t_0), \widetilde{\operatorname{CRew}}_i(t_0, t) + \mathrm{D}_i(t - t_0)\right]$. If the confidence intervals for two learners with indices $i < j$ are more than the h-stable regret bound of $i$ apart, see Equation 2, then deem all learners with index up to $i$ not h-stable and restart the algorithm with a reduced set of base learners.

This kind of elimination condition has already been used in stochastic environments [27, 13, 31], but it requires substantially more care in an adversarial setting. In settings with stochastic rewards (even with corruption, e.g., [31]), no learner can achieve rewards that are significantly higher than the optimal policy. In contrast, in the adversarial setting, it is possible to have negative regret against any fixed policy. Thus, if we were to use the elimination condition in Equation 2 with our base learners as is, then we may eliminate an h-stable base learner $i$ when another base learner $j$ has negative regret against the best fixed policy $\pi_t^\star$. This could in turn lead to undesirable regret in subsequent epochs when only learners with $R(\Pi_i) \gg R(\Pi_{i_\star})$ are left.

**Linking Base Learner Performances.** To address this issue, we link the performance of base learners. Instead of instantiating each learner $\mathbb{A}_i$ with its original policy set $\Pi_i$, we apply it to an extended problem with $M - i$ additional actions $\tilde{a}_{i+1}, \tilde{a}_{i+2}, \ldots, \tilde{a}_M$, and an extended policy set $\tilde{\Pi}_i$ that includes all original policies, along with policies that only choose one of the additional actions $\tilde{a}_i$, that is $\tilde{\Pi}_i \supseteq \Pi_i \cup \{\mathbf{1}\{\tilde{a}_{i+1}\}, \ldots \mathbf{1}\{\tilde{a}_M\}\}$. Whenever a base learner $\mathbb{A}_i$ chooses one of the additional actions $\tilde{a}_j$, then the action proposed by $\mathbb{A}_j$ is followed. Essentially, running each base learner with such an extended action set allows it to choose to follow the actions proposed by any learner with higher index in the hierarchy. The benefit of linking base learners this way is that each base learner now not only competes against the best fixed policy in their set, but also against all learners above them in the hierarchy. Thus, if $\mathbb{A}_i$ is h-stable and extendable (see Definition 3), then it satisfies a regret bound of the form[6]

$$\sum_{\ell=t_0+1}^{t}\left[\mathbb{E}_{a\sim\pi_\ell^j}[r_\ell(a, x_\ell)] - \mathbb{E}_{a\sim\pi_\ell^i}[r_\ell(a, x_\ell)]\right] = \tilde{\mathcal{O}}\left(R(\tilde{\Pi}_i)\sqrt{\frac{t - t_0}{\rho_i}}\right) \tag{3}$$

against any learner $\mathbb{A}_j$ with $j > i$. As a result, since the LHS of Equation 3 is approximately $\widetilde{\operatorname{CRew}}_j(t_0, t) - \widetilde{\operatorname{CRew}}_i(t_0, t) \pm \mathrm{D}_i(t_0, t) \pm \mathrm{D}_j(t_0, t)$, the test in Equation 2 cannot trigger for such $\mathbb{A}_i$, and only base learners that are not h-stable or extendable can be eliminated.

In Appendix C, we show that our regret balancing algorithm achieves the following regret bound:

**Theorem 4.** *Consider a run of Algorithm 1 with Arbe$(\delta, 1, 0)$ and $M$ base algorithms with $1 \leq R(\tilde{\Pi}_1) \leq \cdots \leq R(\tilde{\Pi}_M)$ where $\tilde{\Pi}_i$ is the extended version of policy class $\Pi_i$ with $(M - i)$ additional actions. Then with probability at least $1 - \operatorname{poly}(M)\delta$ the regret $\operatorname{Reg}(t, \Pi_M)$ for all rounds $t \geq i_\star$ is bounded by*

---

[6]We here naturally extend the domain of $r_\ell$ to linked actions as $r_\ell(\tilde{a}_i, x_\ell) = r_\ell(a_\ell^i, x_\ell)$ for all $i \in [M]$.

$$\mathcal{O}\left(\left(\frac{R(\widetilde{\Pi}_{i_\star})}{R(\widetilde{\Pi}_1)}\sqrt{i_\star} + M\right)R(\widetilde{\Pi}_{i_\star})\sqrt{i_\star t \ln \frac{t}{\delta}}\right) , \quad (4)$$

*where $i_\star$ is the smallest index of the base algorithm that is $h$-stable.*

In most settings, the complexity $\frac{R(\widetilde{\Pi}_{i_\star})}{R(\widetilde{\Pi}_1)} \geq \sqrt{M}$ and the first term dominates. This regret recovers the expected regret guarantees of Corral [2] when used with learning rates that do not require knowledge of $i_\star$ but as stronger high-probability bounds. To the best of our knowledge, Theorem 4 is the first high-probability regret bound for adversarial model selection. To illustrate our result, consider to common problem of model selection with nested model classes of dimensions $d_i = 2^i$ for $i \in [M]$, discussed in the introduction. We can use GeometricHedge.P as base learners which are h-stable and extendable if adapted (see Appendix B) and the complexity of extended policy classes $R(\widetilde{\Pi}_i) = R(\Pi_i) + M - i \leq d_i + M$ is not much larger than those of the original policy classes. Arbe with such base learners achieves a regret bound of order $\tilde{O}(d_{i_\star}^2 \sqrt{t})$ up to factors of $M$ and log-factors which are typically small.

## 5 Adversarial Model Selection with Best of Both Worlds Guarantees

In this section, we present our algorithm for adversarial model selection that preserves a logarithmic regret guarantee in case it interacts with a stochastic environment with gap $\Delta > 0$. In order to achieve such a best-of-both-worlds guarantee, we will combine our adversarial regret balancing technique with the algorithmic strategy of Bubeck and Slivkins [11]. The main result for our algorithm is:

**Theorem 5.** *Consider a run of Algorithm 2 with inputs $t_0 = 0$, arbitrary policy policy $\widehat{\pi} \in \Pi_M$ and $M$ base learners $\mathbb{A}_1, \ldots, \mathbb{A}_M$. Then with probability at least $1 - poly(M)\delta$, the following two conditions hold for all $t \geq M^2$ simultaneously. In any adversarial or stochastic environment $\mathbb{B}$, the regret is bounded as*

$$\mathrm{Reg}(t, \Pi_M) = \mathcal{O}\left(\left(M + \ln(t) + \frac{R(\widetilde{\Pi}_{i_\star})}{R(\widetilde{\Pi}_1)}\sqrt{i_\star}\right)R(\widetilde{\Pi}_{i_\star})\sqrt{t(\ln(t) + i_\star)\ln\frac{t}{\delta}}\right) .$$

*If $\mathbb{B}$ is stochastic and there is a unique policy with gap $\Delta > 0$, then the pseudo-regret is bounded as,*

$$\mathrm{PseudoReg}_\mathbb{M}(t, \Pi_M) = \mathcal{O}\left(\frac{R(\Pi_M)^2}{\Delta}\ln(t)\ln\frac{t}{\delta} + \frac{R(\widetilde{\Pi}_{i_\star})^2 R(\widetilde{\Pi}_M)^2}{R(\widetilde{\Pi}_1)^2}\frac{M^2 i_\star}{\Delta}\ln^2\left(\frac{MR(\widetilde{\Pi}_M)}{\Delta\delta}\right)\right).$$
$$(5)$$

The algorithm has the same regret bound as Arbe up to a $\sqrt{\ln t}$ factor. However, in addition, it also maintains poly-logarithmic pseudo-regret if the environment is stochastic and the best policy exhibits a positive gap. Our pseudo-regret bound depends polynomially on $R(\Pi_M)$ in contrast to the $R(\Pi_{i_\star})$ dependency for the adversarial regret rate. This may seem undesirable but, as mentioned in Section 2.1, model selection in stochastic environments with $poly(R(\Pi_{i_\star}), \ln(R(\Pi_M))\frac{\ln(t)}{\Delta}$ regret is impossible (see Appendix A), let alone while maintaining a best-of-both-worlds guarantee with $\sqrt{T}$ regret in adversarial environments. Hence, our algorithm achieves the best kind of guarantee we can hope for, up to improvements in the order of polynomial dependencies.

Our algorithm proceeds in two distinct phases. The first, Arbe-Gap shown in Algorithm 2 is designed to identify a suitable candidate for the optimal policy and to estimate its gap. If such a policy that performs significantly better than any other policy emerges, the algorithm enters the second phase, Arbe-GapExploit which hones in on this policy by playing it most of the time while monitoring its regret in case the environment turns out to be adversarial after all (in which case we simply run Arbe). We now describe both phases.

### 5.1 First Phase: Candidate Policy Identification and Gap Estimation

This goal of this phase is to always maintain the desired adversarial model selection regret guarantee and simultaneously identify the gap between the best policy and the rest if one exists. To achieve the first goal, we employ the regret balancing and elimination technique of Arbe, see Lines 3–11 of Algorithm 2 which are virtually identical to Algorithm 1.

---

**Algorithm 2:** Arbe-Gap$(\delta, t, \widehat{\pi}, n, (\mathbb{A}_i)_{i=s}^M)$

---

**1** **Input:** failure probability $\delta$, timestep $t_0$, focus policy $\widehat{\pi}$, number of (re)starts $n$, learners $(\mathbb{A}_i)_{i=s}^M$

**2** Set $\Pi_{M+1} = \Pi_M \backslash \{\widehat{\pi}\}$ and $\mathbb{A}_{M+1}$ as a copy of $\mathbb{A}_M$ with policy class $\Pi_{M+1}$

**3** Initialize base learners $\mathbb{A}_s, \mathbb{A}_{s+1}, \ldots, \mathbb{A}_{M+1}$ with extended policy classes $\widetilde{\Pi}_s, \widetilde{\Pi}_{s+1}, \ldots, \widetilde{\Pi}_{M+1}$

**4** Set sampling probabilities $\rho_i = \frac{R(\widetilde{\Pi}_i)^{-2}}{\sum_{j=s}^{M+1} R(\widetilde{\Pi}_j)^{-2}}$ for all $i \in \{s, \ldots, M+1\}$, and $\rho_i = 0$ for $i < s$

**5** **for** *round* $t = t_0 + 1, t_0 + 2, \ldots$ **do**

**6**      Sample base learner index $b_t \sim \text{Categorical}(\rho_1, \cdots, \rho_M)$

**7**      Get context $x_t$ and compute $a_t^i \sim \pi_t^i(\cdot|x_t)$, the action each base learner $\mathbb{A}_i$ proposes for $x_t$

**8**      Play action $a_t = a_t^{b_t}$ (resolve linked actions if necessary) and receive reward $r_t(a_t, x_t)$

**9**      Update all base learners $\mathbb{A}_i$ with reward $\frac{\mathbf{1}\{b_t=i\} r_t(a_t, x_t)}{\rho_i}$

**10**      **if** *Equation 2 holds between* $\mathbb{A}_i, \mathbb{A}_j$ *for* $s \leq i < j \leq M+1$ **then**

**11**         restart algorithm by running Arbe-Gap$(\delta, t, \widehat{\pi}, n+1, (\mathbb{A}_{i+1}, \cdots, \mathbb{A}_M))$

     *// Gap Test:* $\mathbb{A}_M$ *better than* $\mathbb{A}_{M+1}$*?*

**12**      Set $\text{W}(t_0, t) = \Theta\left(\sqrt{\frac{R(\widetilde{\Pi}_M)^2}{\rho_M(t-t_0)} \ln \frac{n(t-t_0)}{\delta}} + \frac{\ln \frac{n \ln(t-t_0)}{\delta}}{\rho_M(t-t_0)}\right)$

**13**      Set $\widehat{\Delta}_t = \frac{\widetilde{\text{CRew}}_M(t_0, t) - \widetilde{\text{CRew}}_{M+1}(t_0, t)}{t-t_0} - \text{W}(t_0, t)$

**14**      **if** $2W(t_0, t) \leq \widehat{\Delta}_t \leq R(\widetilde{\Pi}_M)^2$ **then**

**15**         Run Arbe-GapExploit (Appendix D) with inputs $\delta, t_0 = t, \mathbb{A}_M, \widehat{\pi}$ and $\widehat{\Delta} = \widehat{\Delta}_t$

**16**         Run Arbe with inputs $t_0, s, \delta$

     *// New Candidate Policy Test*

**17**      **if** *a policy* $\pi \in \Pi_M \backslash \{\widehat{\pi}\}$ *has been selected in more than* $\frac{3t}{4}$ *of all* $t \geq 9$ *rounds* **then**

**18**         restart algorithm by running Arbe-Gap$(\delta, t, \pi, n+1, (\mathbb{A}_i)_{i=s}^M)$

---

For the second goal, determining the gap, Arbe-Gap maintains a candidate $\widehat{\pi} \in \Pi_M$ for the optimal policy and estimates its gap as follows. The learner hierarchy $\mathbb{A}_1, \ldots, \mathbb{A}_M$ is augmented at the top with an additional learner $\mathbb{A}_{M+1}$ operating on the policy class $\Pi_{M+1} = \Pi_M \backslash \{\widehat{\pi}\}$. Since $\mathbb{A}_{M+1}$ is identical to $\mathbb{A}_M$ except that it does not have access to $\widehat{\pi}$, we can obtain a gap estimate for $\widehat{\pi}$ by monitoring the difference in reward estimates $\widetilde{\text{CRew}}_{M+1}$ and $\widetilde{\text{CRew}}_M$ of $\mathbb{A}_{M+1}$ and $\mathbb{A}_M$. In fact, $\widehat{\Delta}_t$ in Line 13 is a lower confidence bound on the difference between the best policies in $\widehat{\Pi}_M$ and $\widehat{\Pi}_M \backslash \{\widehat{\pi}\}$ and thus, the gap of $\widehat{\pi}$. We test at every round whether $\widehat{\Delta}_t$ exceeds its confidence width $2\text{W}(t_0, t)$. If this test triggers then $\widehat{\pi}$ must have a positive gap $\Delta$ of order $\text{W}(t_0, t)$ and further, $\widehat{\Delta}_t$ must be a multiplicative estimate of $\Delta$, that is, $\widehat{\Delta}_t \leq \Delta \leq 2\widehat{\Delta}_t$. The latter holds, because $\widehat{\Delta}_t + 2\text{W}(t_0, t)$ is an upper-confidence bound on $\Delta$ and the test condition implies $2\widehat{\Delta}_t \geq \widehat{\Delta}_t + 2\text{W}(t_0, t) \geq \Delta$. Since in this case, we have determined that $\widehat{\pi}$ is optimal with a gap of order $\widehat{\Delta}_t$, we can move on to the second phase Arbe-GapExploit discussed later.

Assume the candidate policy $\widehat{\pi}$ is indeed optimal and exhibits a positive gap $\Delta$. Since $\widehat{\Delta}_t$ concentrates around $\Delta$ at a rate of $\text{W}(t_0, t) \approx \frac{\text{poly}(R_M(\widetilde{\Pi}_M))}{\sqrt{t-t_0}}$, the condition of the gap test in Line 14 must trigger after at most $t - t_0 \lesssim \frac{\text{poly}(R_M(\widetilde{\Pi}_M))}{\Delta^2}$ rounds. Finally, since Arbe-Gap always maintains the $\text{poly}(R(\widetilde{\Pi}_{i_\star}))\sqrt{t-t_0}$ adversarial regret rate, the total pseudo-regret incurred until the test triggers is of order $\frac{\text{poly}(R_M(\widetilde{\Pi}_M))}{\Delta}$, leading to the second term in the bound in Equation 5.

With the techniques above, we can reliably detect a positive gap if the candidate policy $\widehat{\pi}$ exhibits one. It remains to identify a suitable candidate $\widehat{\pi}$ of the optimal policy $\pi^\star$ in stochastic environments. To do so, we use the following two observations: Arbe-Gap maintains a $\text{poly}(R(\widetilde{\Pi}_{i_\star}))\sqrt{t}$ adversarial regret rate overall and each policy but $\pi^\star$ will incur on average at least a regret of $\Delta$ per round. Hence, in order to maintain that regret, Arbe-Gap must select $\pi^\star$ in the majority of all rounds when $t = \omega(\frac{\text{poly}(R(\widetilde{\Pi}_{i_\star}))}{\Delta^2})$. Otherwise the regret grows as $\Omega(t\Delta) = \omega(\text{poly}(R(\widetilde{\Pi}_{i_\star}))\sqrt{t})$ violating the

adversarial rate. To leverage this observation and identify a suitable candidate policy $\widehat{\pi}$, Line 17 of Algorithm 2 always checks whether there is a policy other than the current candidate policy that has been selected in at least $3/4$ of all rounds.[7] If so, the algorithm is restarted with this policy as candidate $\widehat{\pi}$. One can show that there are at most $\mathcal{O}(\ln t)$ restarts due to candidate policy switches, only increasing the adversarial regret rate of Arbe-Gap by a factor of $\mathcal{O}(\sqrt{\ln t})$ compared to Arbe's. Our candidate policy selection approach is similar to that by Wei et al. [31] for the corrupted reward setting but they require each adapted base learner to achieve a logarithmic regret rate in the first place. Instead, our approach only requires an adversarial regret rate from the base learner.

### 5.2 Second Phase: Exploitation

Since each base learner only needs to satisfy a $\sqrt{T}$ regret rate even in stochastic environments, we generally cannot hope to recover logarithmic pseudo-regret by only selecting among them. For logarithmic pseudo-regret, we need to ensure that the given policy $\widehat{\pi}$ with gap estimate $\widehat{\Delta}$ is played sufficiently often. However, we also need to monitor its regret against all other policies in case the environments turns out to be adversarial. If at any point, $\widehat{\pi}$ fails to maintain a gap of order $\widehat{\Delta}$, we can conclude that the environment is adversarial. Then Arbe-GapExploit returns and we simply play an instance of Arbe (Line 16 of Algorithm 2).

We will present a brief summary of the main intuition behind our exploitation phase approach here and defer a longer discussion and the detailed pseudo-code to Appendix D. In each round, we play policy $\widehat{\pi}$ with probability approximately $1 - \frac{\text{poly}(R(\Pi_M))}{\widehat{\Delta}^2 t}$, and with the remaining probability $\frac{\text{poly}(R(\Pi_M))}{\widehat{\Delta}^2 t}$ a version of base learner $\mathbb{A}_M$ with policy class $\Pi_M \setminus \{\widehat{\pi}\}$.

If the environment is indeed stochastic, then $\widehat{\pi} = \pi^\star$ does not incur any pseudo-regret and the total regret in other $t' \approx t \cdot \frac{\text{poly}(R(\Pi_M))}{\widehat{\Delta}^2 t} \approx \frac{\text{poly}(R(\Pi_M))}{\widehat{\Delta}^2}$ rounds can be bounded as $t' \cdot \Delta + \text{Reg}_{\mathbb{A}_M}(t', \Pi_M \setminus \{\widehat{\pi}\}) \approx \frac{\text{poly}(R(\Pi_M))}{\Delta} \ln(t)$. The first term is the regret of the best policy in $\Pi_M \setminus \{\widehat{\pi}\}$ and the second term is the regret of $\mathbb{A}_M$ against that policy. Since $\mathbb{A}_M$ is h-stable on $\Pi \setminus \{\widehat{\pi}\}$, its regret against that policy is at most $\text{poly}(R(\Pi_M))\sqrt{t'}$ and we get the desired pseudo-regret.

To ensure good regret in the adversarial case, we need to detect quickly enough when $\widehat{\pi}$ does not exhibit a performance gap anymore and fall back to a fully adversarial algorithm. Similar to $\widehat{\Delta}_t$ in the first phase, we use a lower confidence bound on the average performance gap and continuously test whether it falls below $\frac{\widehat{\Delta}}{2}$. This simple approach would give $\sqrt{t}$ adversarial regret but may exhibit a $\text{poly}(R(\Pi_M))$ dependency. Fortunately, we can avoid this dependency and retain the desired model selection regret rates by extending the intuition above to also test an upper bound on the gap. For details, see Appendix D.

## 6 Conclusions

We have described and analyzed a novel model selection scheme for bandit algorithms that benefits from best-of-both-worlds high probability regret guarantees. Though not restricted to linear bandits, our machinery can be specifically applied to adversarial/stochastic linear bandit tasks, where model selection is performed on the unknown dimensionality of the linear reward function. This has required extending the regret balancing technique of model selection from stochastic to adversarial rewards and a very careful handling of the associated mis-specification tests. The base learners aggregated by our meta-algorithm have to satisfy an anytime high probability regret guarantee in the adversarial case, along with regret stability and action space extendability properties which we have shown are satisfied by (variants of) known algorithm in the bandits literature.

Our best-of-both world model selection regret guarantees cannot in general be improved, specifically in stochastic environments with gaps, where it is generally impossible to obtain $\log t$-like model selection bounds that only depend on the complexity of $\Pi_{i_\star}$. On the other hand, it would be nice to see in Theorem 5 a better polynomial dependence on $M$ and $R(\Pi_M)$. Also, it might be possible to adapt Arbe-Gap to work with adversarially corrupted stochastic rewards while doing model selection on the complexity of the models. We leave this as a future research direction.

---

[7]Due to linking learners, there is slight ambiguity in defining the selected policy per round. We here determine the selected policy as the policy chosen by the learner that eventually picked $a_t$ *after resolving linked actions*.

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
