# Contents

# A Lower Bound on Model Selection for Stochastic Environments

Considering a simple multi-armed bandit problem suffices to prove that in stochastic environments with gap $\Delta$ it is generally impossible to obtain model selection guarantees of the form

$$\frac{R^2(\Pi_{i_\star}) \log T}{\Delta} \ .$$

**Theorem 6.** *Let $K_1, K_2 \in \mathbb{N}$ with $K_1 < K_2$ and let $\Delta \in \mathbb{R}^+$ be fixed. Further, let $c : \mathbb{N} \to \mathbb{R}$ be an arbitrary function over the real numbers. If $c(K_1) < K_2 - K_1$, there is a class of $K_2$ multi-armed bandit problems with one optimal arm and gaps in $[\Delta, 2\Delta]$ and a $T_0 \in \mathbb{N}$ such that the following holds. For any algorithm $\mathbb{A}$ and for all $T \geq T_0$, there is a problem instance such that*

$$\mathbb{E}[\mathrm{Reg}_\mathbb{A}(T)] > \begin{cases} \frac{c(K_1)}{\Delta} \ln T & \text{if the optimal arm } a^\star \in [K_1] \\ \frac{c(K_2)}{\Delta} \ln T & \text{otherwise .} \end{cases}$$

**Remark 7.** *The proof below actually shows the following stronger version*

$$\mathbb{E}[\mathrm{Reg}_\mathbb{A}(T)] > \begin{cases} \frac{c(K_1)}{\Delta} \ln T & \text{if the optimal arm } a^\star \in [K_1] \\ \frac{\Delta}{16} T^{1 - \frac{c(K_1)}{K_2 - K_1}} & \text{otherwise .} \end{cases}$$

*This shows that if we aim to obtain an instance-dependent logarithmic regret bound that only scales with $K_1$ when $a^\star \in [K_1]$, then we cannot recover sublinear regret for $K_2 \gg K_1$ when $a^\star \notin [K_1]$.*

*Proof.* We will show the statement for $K_1 = 1$ but it can be trivially generalized to $K_1 > 1$. The rewards of all arms and in all instances are drawn from a Gaussian distribution with variance 1 but different means. We denote by $\mu_i^j$ the mean reward of arm $j$ in instance $i$. We identify each bandit instance in the family by its mean rewards $\mu_i$. They are given by

$$\mu_i = [\Delta, 0, \ldots, 0] \qquad\qquad \text{for } i = 1 \qquad (6)$$
$$\mu_i = [\Delta, 0, \ldots, \underbrace{2\Delta}_{i\text{'th pos}}, 0, \ldots, 0] \qquad\qquad \text{for } i > 1 \qquad (7)$$

Thus, arm $i$ is optimal in instance $i$. Consider any algorithm $\mathbb{A}$ and assume it violates the lower bound for the case where $a^\star \in [K_1]$, i.e., in the first problem instance,

$$\mathbb{E}\left[\mathrm{Reg}_\mathbb{A}(T, \mu_1)\right] \leq \frac{c(K_1) \log(T)}{\Delta}$$

holds for some $T \geq T_0$ (we will specify $T_0$ later). Otherwise, the statement is already true. We will now show that this algorithm has to satisfy the regret lower bound in one of the other problem instances for $T$. By definition of $\mu_1$, we can write the expected regret as

$$\mathbb{E}\,\mathrm{Reg}_\mathbb{A}(T, \mu_1) \geq 2\Delta \sum_{i=K_1+1}^{K_2} \mathbb{E}_1\left[T_i(T)\right] \ ,$$

where $T_i(T)$ is the (random) number of times $\mathbb{A}$ has pulled arm $i$ up to time $T$. $\mathbb{E}_j$ and $\mathbb{P}_j$ denote the expectation and the probability distribution induced by algorithm $\mathbb{A}$ in instance $j$, respectively. Consider now a problem instance $\widehat{i} \in \mathrm{argmin}_{K_1+1 \leq i \leq K_2} \mathbb{E}_1\left[T_i(T)\right]$. As a consequence of the previous two observations,

$$2\Delta(K_2 - K_1)\mathbb{E}_1\left[T_{\widehat{i}}(T)\right] \leq \frac{c(K_1) \log(T)}{\Delta} \ ,$$

hence

$$\mathbb{E}_1\left[T_{\widehat{i}}(T)\right] \leq \frac{c(K_1) \log(T)}{2(K_2 - K_1)\Delta^2} \ .$$

By the divergence decomposition [20, Lemma 15.1],

$$\mathrm{KL}\left(\mathbb{P}_1, \mathbb{P}_{\widehat{i}}\right) = \mathbb{E}_1\left[T_{\widehat{i}}(T)\right] \frac{(2\Delta)^2}{2} \leq \frac{c(K_1) \log(T)}{K_2 - K_1} \ .$$

Define the event $\mathcal{E} = \{T_1(T) \leq \frac{T}{2}\}$. Notice that,

$$\frac{T}{2}\Delta\mathbb{P}_1(\mathcal{E}) \leq \mathbb{E}_1\operatorname{Reg}_\mathbb{A}(T, \mu_1) \leq \frac{c(K_1)\log(T)}{\Delta}$$

which implies,

$$\mathbb{P}_1(\mathcal{E}) \leq \frac{2c(K_1)\log(T)}{\Delta^2 T} \ . \tag{8}$$

By the Bretagnolle-Huber inequality,

$$\mathbb{P}_1(\mathcal{E}) + \mathbb{P}_{\widehat{i}}(\mathcal{E}^c) \geq \frac{1}{2}\exp\left(-\frac{c(K_1)\log(T)}{K_2 - K_1}\right) = \frac{1}{2}\left(\frac{1}{T}\right)^{\frac{c(K_1)}{K_2 - K_1}} ,$$

and by combining the last two inequalities, we can lower bound $\mathbb{P}_{\widehat{i}}(\mathcal{E}^c)$ as

$$\mathbb{P}_{\widehat{i}}(\mathcal{E}^c) \geq \frac{1}{2}\left(\frac{1}{T}\right)^{\frac{c(K_1)}{K_2 - K_1}} - \frac{2c(K_1)\log(T)}{\Delta^2 T} \ .$$

Since $\mathbb{E}_{\widehat{i}}\operatorname{Reg}_\mathbb{A}(T, \mu_{\widehat{i}}) \geq \frac{T\Delta}{2}\mathbb{P}_{\widehat{i}}(\mathcal{E}^c)$ this implies,

$$\mathbb{E}_{\widehat{i}}\operatorname{Reg}_\mathbb{A}(T, \mu_{\widehat{i}}) \geq \frac{T\Delta}{2}\left(\frac{1}{2}\left(\frac{1}{T}\right)^{\frac{c(K_1)}{K_2 - K_1}} - \frac{2c(K_1)\log(T)}{\Delta T}\right) = \frac{\Delta}{4}T^{1 - \frac{c(K_1)}{K_2 - K_1}} - c(K_1)\log(T) \ .$$

By setting $T_0$ sufficiently large as a function of $\Delta$, $K_1$, $K_2$ and $c$, we can conclude that

$$\mathbb{E}_{\widehat{i}}\operatorname{Reg}_\mathbb{A}(T, \mu_{\widehat{i}}) \geq \frac{\Delta}{4}T^{1 - \frac{c(K_1)}{K_2 - K_1}} - c(K_1)\log(T) \geq \frac{\Delta}{8}T^{1 - \frac{c(K_1)}{K_2 - K_1}} > \frac{c(K_2)\log(T)}{\Delta} \ .$$

Specifically, it suffices to set $T_0$ as the smallest value of $T$ that satisfies the last two inequalities in the display. This shows that the lower bound holds at time $T$ for problem instance $\widehat{i}$. □

# B   Examples of h-Stability and Extendability

This appendix shows examples of h-stability and extendabilty. The first example, contained in Section B.1, is a variant of the Geometric Hedge algorithm from [7]. We sketch a second example in Section B.2, where we deal with a high probability variant of the Exp4 algorithm from [6].

## B.1   Geometric Hedge for Adversarial Linear Bandits

Let us start by introducing a weighted version of the Geometric Hedge Algorithm that satisfies the h-stability condition of Definition 2. We consider the setting where the action $\mathcal{A} \subset \mathbb{R}^d$ is finite (but potentially large). We denote by $\mathbf{a}_t$ and $\boldsymbol{\omega}_t$ the learner's action selection and the adversary's reward vector at time $t$, respectively. The associated reward $r_t = \mathbf{a}_t^\top \boldsymbol{\omega}_t$ is assumed to lie in the interval $[-1, 1]$. Moreover, let $\delta \in (0, 1)$ be a probability parameter and set for brevity $\delta' = \frac{\delta}{|\mathcal{A}|}$.

The algorithm, an anytime variant of the Geometric Hedge algorithm from [7] with John's ellipsoid exploration (e.g., [9]) is detailed in Algorithm 3. We call the algorithm Anytime Weighted Geometric Hedge. The algorithm takes in input the set of actions $\mathcal{A}$, the failure probability $\delta$, and a weighting probability $\rho \in (0, 1]$ which will play the role of an importance weight. As is standard, the algorithm maintains over time a distribution $p_t$ over $\mathcal{A}$, which is itself a mixture of an exponential weight distribution $q_t$ and an exploration distribution $p_E$. The exploration distribution $p_E$ is defined beforehand to be the John's ellipsoid distribution associated with $\mathcal{A}$ (see [9] for details). As in Geometric Hedge [7], the algorithm builds a covariance matrix $\boldsymbol{\Sigma}_t$ by computing the expectation of $\mathbf{a}\mathbf{a}^\top$ where $\mathbf{a}$ is drawn according to the current distribution $p_t$ over actions.[8] Then, the algorithm samples a Bernoulli random variable $b_t$, and computes an (importance-weighted) unbiased estimator $\widehat{\boldsymbol{\omega}}_t$ of $\boldsymbol{\omega}_t$, which is plugged into the (biased) reward estimator $\widetilde{r}_t(\mathbf{a})$ that the algorithm associates with every action $\mathbf{a} \in \mathcal{A}$. The factors $\widetilde{r}_t(\mathbf{a})$ are those that determine the exponential update of distribution $p_t$.

---

[8]In this section, we denote by $\mathcal{F}_{t-1}$ the $\sigma$-algebra generated by all past random variables up to, but excluding, the random draw of $\mathbf{a}_t$ (so that $p_t$ is $\mathcal{F}_{t-1}$-measurable).

**Algorithm 3:** Anytime Weighted Geometric Hedge.

---

**1 Input:** Action set $\mathcal{A}$, failure probability $\delta$, weighting probability $\rho \in (0, 1]$.

**2** Initialize $w_1(\mathbf{a}) = 1$, $W_1 = |\mathcal{A}|$ and $q_1(\mathbf{a}) = \frac{1}{|\mathcal{A}|}$ for all $\mathbf{a} \in \mathcal{A}$

**3 for** $t = 1, 2, \cdots$ **do**

**4** $\quad$ Compute sampling distribution

$$p_t(\mathbf{a}) = (1 - \gamma_t)q_t(\mathbf{a}) + \gamma_t p_E(\mathbf{a}) \qquad \text{where } q_t(\mathbf{a}) = \frac{w_t(\mathbf{a})}{W_t} \tag{9}$$

**5** $\quad$ Adversary generates reward vector $\boldsymbol{\omega}_t$

**6** $\quad$ Sample action $\mathbf{a}_t \sim p_t$

**7** $\quad$ Observe and gather reward $r_t = \mathbf{a}_t^\top \boldsymbol{\omega}_t$

**8** $\quad$ Build covariance matrix $\boldsymbol{\Sigma}_t = \mathbb{E}_{\mathbf{a} \sim p_t} \left[ \mathbf{a}\mathbf{a}^\top | \mathcal{F}_{t-1} \right]$

**9** $\quad$ Sample $b_t \sim \text{Ber}(\rho)$

**10** $\quad$ Compute unbiased reward vector estimator $\widehat{\boldsymbol{\omega}}_t = b_t \frac{r_t \boldsymbol{\Sigma}_t^{-1} \mathbf{a}_t}{\rho}$

**11** $\quad$ Compute the reward upper bounds

$$\widetilde{r}_t(\mathbf{a}) = \mathbf{a}^\top \widehat{\boldsymbol{\omega}}_t + 2\mathbf{a}^\top \boldsymbol{\Sigma}_t^{-1} \mathbf{a} \sqrt{\frac{\ln(12t^2/\delta')}{\rho \, d \, t}} \quad \forall \mathbf{a} \in \mathcal{A}$$

$\quad$ Update distribution

$$w_{t+1}(\mathbf{a}) = \exp\left( \eta_{t+1} \sum_{\ell=1}^{t} \widetilde{r}_\ell(\mathbf{a}) \right) \quad \forall \mathbf{a} \in \mathcal{A} \tag{10}$$

$\quad$ Update normalization factor $W_{t+1} = \sum_{\mathbf{a} \in \mathcal{A}} w_{t+1}(\mathbf{a})$

---

**Remark 8.** *For simplicity, the algorithm is formulated for the case where the action space $\mathcal{A}$ is finite. When $\mathcal{A}$ is infinite, we can still formulate an algorithm that applies to a $\epsilon$-cover of $\mathcal{A}$ that is restarted at exponentially increasing time-steps $t_0 = 1, 2, 4, 8, \ldots,$. At the beginning of each epoch, the covering level $\epsilon$ is set to $\mathcal{O}(1/t_0)$ so as to obtain an anytime algorithm. A very similar bound to the one in Theorem 10 is obtained, where $|\mathcal{A}|$ therein is replaced by $|\mathcal{A}_{1/t}|$, $\mathcal{A}_{1/t}$ being a $1/t$-cover of $\mathcal{A}$ w.r.t. the infinity norm.*

**Remark 9.** *Notice that we are using the fact that $\boldsymbol{\Sigma}_t$ is invertible for all $t$. This is no loss of generality, as we can always assume that $\mathcal{A}$ spans the whole $d$-dimensional space (if this is not the case, we can project each $\mathbf{a}$ onto the space spanned by $\mathcal{A}$ and reduce to this case). Combined with the fact that, for all $t$, distribution $p_t$ assigns a nonzero probability to each action, this implies that the expectation $\boldsymbol{\Sigma}_t = \mathbb{E}_{\mathbf{a} \sim p_t} \left[ \mathbf{a}\mathbf{a}^\top | \mathcal{F}_{t-1} \right]$ must be full rank.*

### B.1.1 h-Stability

Setting the mixture factor $\gamma_t$ and the learning rate $\eta_t$ appropriately, we can prove the following regret guarantee for Algorithm 3.

**Theorem 10.** *Let the Anytime Weighted Geometric Hedge Algorithm be run with*

$$\eta_t = \mathcal{O}\left( \frac{\rho \gamma_t}{d + \sqrt{\frac{d}{t}} \sqrt{\rho \ln |\mathcal{A}| \ln \frac{t}{\delta}}} \right), \qquad \gamma_t = \min\left\{ \sqrt{\frac{d \ln |\mathcal{A}| \ln \frac{t}{\delta}}{\rho t}}, \frac{1}{2} \right\},$$

*and $\rho \in (0, 1]$. Then with probability at least $1 - \delta(3 + 2d^2)$, simultaneously for all $t$, the regret $\text{Reg}(t)$ after $t$ rounds can be bounded as*

$$\text{Reg}(t) = \mathcal{O}\left( \sqrt{\frac{d \, t \ln |\mathcal{A}|}{\rho} \ln \frac{t}{\delta}} + \rho \ln |\mathcal{A}| \ln \frac{t}{\delta} \ln t + \left( \frac{\ln |\mathcal{A}|}{d} \ln \frac{t}{\delta} \right)^{1/4} t^{1/4} \left( \frac{1}{\rho} \right)^{3/4} + \frac{1}{\rho} \ln \frac{t}{\delta} \right).$$

*In the above, the big-oh notation only hides absolute constants.*

Hence we have the following corollary.

**Corollary 11.** *Let the complexity $R(\Pi)$ of the policy space $\Pi$ be defined as $R(\Pi) = \sqrt{d \log |\mathcal{A}|}$. Then with the same assumptions and setting as in Theorem 10, the Anytime Weighted Geometric Hedge algorithm is h-stable in that, for $t \to \infty$ and constant $\rho$ independent of $t$, its regret $\text{Reg}(t)$ satisfies*

$$\text{Reg}(t) = \mathcal{O}\left( R(\Pi) \sqrt{\frac{t}{\rho} \ln \frac{t}{\delta}} \right) ,$$

*with probability at least $1 - \delta$, where the big-oh hides terms in $t$ which are lower order than $\sqrt{t \log t}$ as $t \to \infty$.*

### B.1.2 Extendability

The extendability of the Anytime Weighted Geometric Hedge algorithm is easily obtained by simply observing that, since $\mathcal{A} \subseteq \mathbb{R}^d$, we can extend the dimensionality $d$ to $d + k$. Then we extend each $\mathbf{a} = (a_1, \ldots, a_d) \in \mathcal{A}$ to $\mathbf{a}' = (a_1, \ldots, a_d, \underbrace{0, \ldots, 0}_{k \text{ zeros}})$, and add $k$ new actions $\mathbf{b}_i$ of the form

$$\mathbf{b}_i = (\underbrace{0, \ldots, 0}_{d \text{ zeros}}, \underbrace{0, \ldots, 1, \ldots, 0}_{\text{position } i})$$

for $i = 1, \ldots, k$. Denote the extended policy space by $\mathcal{A}'$. Consistent with this extension, any policy $\pi$ in the original policy space $\Pi$ has to be interpreted as a probability distribution in $\Delta_{\mathcal{A}'}$, whose last $k$ components are zero, while the $k$ extra indicator policies $\mathbf{1}\{b_1\}, \ldots, \mathbf{1}\{b_k\}$ are degenerate probability distributions in $\Delta_{\mathcal{A}'}$, where $\mathbf{1}\{b_i\}$ places all its probability mass on the $(d + i)$-th component. Finally the adversary reward vector $\boldsymbol{\omega}_t \in \mathbb{R}^d$ turns into the $(d + k)$-dimensional vector $\boldsymbol{\omega}'_t \in \mathbb{R}^{d+k}$, where the first $d$ components of the two vectors are the same, and the $(d + i)$-th component of $\boldsymbol{\omega}'_t$ is simply the regret generated for the $i$-th extra action $\mathbf{b}_i$.

### B.1.3 Removing an Individual Policy For Best of Both Worlds Regret

For the best of both worlds regret in Section 5 we adopt the following view. Here, each policy corresponds to a single action $a \in \mathcal{A}$, i.e., the policy space of the learner is $\Pi = \{\mathbf{1}\{a\} : a \in \mathcal{A}\}$. In this case, the random draw of $\mathbf{a}_t \sim p_t$ is interpreted as a random choice of *policy*. Note that this view does not impact the regret guarantee and is consistent with approach for extendability above. This view is necessary for a positive gap to be possible and removing a certain policy can be easily implemented by removing the corresponding action.

### B.1.4 Proofs

We work under the assumption all the rewards $r_t$ have values between $-1$ and $1$:

**Assumption B.1** (Boundedness). *The true rewards $r_t$ are bounded in that $\forall \mathbf{a} \in \mathcal{A}$ and $\forall t \in \mathbb{N}$ we have $|\mathbf{a}^\top \boldsymbol{\omega}_t| \leq 1$.*

For all $\mathbf{a} \in \mathcal{A}$, define

$$\widehat{r}_t(\mathbf{a}) = \mathbf{a}^\top \widehat{\boldsymbol{\omega}}_t \quad \text{where} \quad \widehat{\boldsymbol{\omega}}_t = b_t \frac{r_t (\boldsymbol{\Sigma}_t)^{-1} \mathbf{a}_t}{\rho} .$$

We will use the notation $\mathbb{E}[\cdot|\mathcal{F}_t]$ to denote the conditional expectation where the sigma algebra $\mathcal{F}_t$ is generated by the random variables $(\boldsymbol{\omega}_1, b_1, \mathbf{a}_1, \cdots, \boldsymbol{\omega}_{t-1}, b_{t-1}, \mathbf{a}_{t-1}, \boldsymbol{\omega}_t, b_t, \mathbf{a}_t)$. Let $\mathcal{F}_t^-$ be the sigma algebra generated by $(\boldsymbol{\omega}_1, b_1, \mathbf{a}_1, \cdots, \boldsymbol{\omega}_{t-1}, b_{t-1}, \mathbf{a}_{t-1}, \boldsymbol{\omega}_t)$. Observe that $\widehat{\boldsymbol{\omega}}_t, \widehat{r}_t(\cdot)$ and $\widetilde{r}_t(\cdot)$ are $\mathcal{F}_t$ measurable, $\boldsymbol{\Sigma}_t$ is $\mathcal{F}_{t-1}$ measurable, $\mathbb{E}\left[\widehat{\boldsymbol{\omega}}_t|\mathcal{F}_t^-\right] = \boldsymbol{\omega}_t$ and $\mathbb{E}[b_t|\mathcal{F}_{t-1}] = \mathbb{E}\left[b_t|\mathcal{F}_t^-\right] = \rho$. When considering $\mathbb{E}\left[\cdot|\mathcal{F}_t^-\right]$, the expectation is over $\mathbf{a}_t, b_t$ holding $\boldsymbol{\omega}_t$ fixed. Every time we consider an expectation of the form $\mathbb{E}_{\mathbf{a} \sim p_t}\left[\widehat{r}_t(\mathbf{a})|\mathcal{F}_t^-\right], \mathbb{E}_{\mathbf{a} \sim p_t}\left[\widetilde{r}_t(\mathbf{a})|\mathcal{F}_t^-\right], \mathbb{E}_{\mathbf{a} \sim p_t}\left[\widehat{r}_t(\mathbf{a})|\mathcal{F}_t\right]$ or $\mathbb{E}_{\mathbf{a} \sim p_t}\left[\widetilde{r}_t(\mathbf{a})|\mathcal{F}_t\right]$ the random variable $\mathbf{a}$ is a sample from $p_t$ conditionally independent from $\mathbf{a}_t$ given $\mathcal{F}_t$ or $\mathcal{F}_t^-$. Moreover, for notational simplicity, whenever possible, we will omit the absolute multiplicative constants, and instead resort to a big-oh notation.

**Lemma 12.** *Let*

$$\sup_{\mathbf{a},\mathbf{b}\in\mathcal{A}} \mathbf{a}^\top \mathbf{\Sigma}_t^{-1} \mathbf{b} \leq \frac{c(d)}{\gamma_t} \,, \tag{11}$$

*for some function $c(\cdot)$ whose value will be detailed later on. Then for any fixed $\mathbf{a}\in\mathcal{A}$ and $t\in\mathbb{N}$ the following holds:*

1. *$|\widehat{r}_t(\mathbf{a})| \leq \frac{c(d)}{\rho\gamma_t}$ ;*

2. *$\mathbf{a}^\top \left(\mathbf{\Sigma}_t\right)^{-1}\mathbf{a} \leq \frac{c(d)}{\gamma_t}$ ;*

3. *$\mathbb{E}_{\widetilde{\mathbf{a}}\sim p_t}\left[\widetilde{\mathbf{a}}^\top \left(\mathbf{\Sigma}_t\right)^{-1}\widetilde{\mathbf{a}}\Big|\mathcal{F}_{t-1}\right] = d \qquad and \qquad \mathbb{E}_{\widetilde{\mathbf{a}}\sim p_t}\left[\widetilde{\mathbf{a}}^\top \left(\mathbf{\Sigma}_t\right)^{-1}\widetilde{\mathbf{a}}\Big|\mathcal{F}_t^-\right] = d$ ;*

4. *$\mathbb{E}\left[\widehat{r}_t^2(\mathbf{a})|\mathcal{F}_t^-\right] \leq \frac{\mathbf{a}^\top (\mathbf{\Sigma}_t)^{-1}\mathbf{a}}{\rho}$ .*

*Proof.* Item 1 simply follows by recalling that $\widehat{r}_t(\mathbf{a}) = b_t \frac{r_t \mathbf{a}^\top (\mathbf{\Sigma}_t)^{-1}\mathbf{a}_t}{\rho}$ with $|r_t| \leq 1$. The condition from Equation 11 then implies the result. Item 2 follows from the same condition. Item 3 follows by observing that

$$\mathbb{E}_{\widetilde{\mathbf{a}}\sim p_t}[\widetilde{\mathbf{a}}^\top \left(\mathbf{\Sigma}_t\right)^{-1}\widetilde{\mathbf{a}}|\mathcal{F}_{t-1}] = \mathbb{E}_{\widetilde{\mathbf{a}}\sim p_t}[\text{tr}(\widetilde{\mathbf{a}}\widetilde{\mathbf{a}}^\top \mathbf{\Sigma}_t^{-1})|\mathcal{F}_{t-1}] = \text{tr}\left(\mathbb{E}_{\widetilde{\mathbf{a}}\sim p_t}[\widetilde{\mathbf{a}}\widetilde{\mathbf{a}}^\top]\mathbf{\Sigma}_t^{-1}\right) = d \,.$$

Item 4 follows from the definition of $\widehat{r}_t(\mathbf{a})$. In fact, for any fixed $\mathbf{a}$, we can write

$$\begin{aligned}
\mathbb{E}\left[\widehat{r}_t^2(\mathbf{a}) \,|\, \mathcal{F}_t^-\right] &= \frac{1}{\rho^2}\mathbb{E}_{b_t\sim\text{Ber}(\rho),\mathbf{a}_t\sim p_t}[b_t^2 r_t^2 \mathbf{a}^\top \left(\mathbf{\Sigma}_t\right)^{-1}\mathbf{a}_t\mathbf{a}_t^\top \left(\mathbf{\Sigma}_t\right)^{-1}\mathbf{a} \,|\, \mathcal{F}_t^-] \\
&= \frac{1}{\rho}\mathbb{E}_{\mathbf{a}_t\sim p_t}[r_t^2\mathbf{a}^\top \left(\mathbf{\Sigma}_t\right)^{-1}\mathbf{a}_t\mathbf{a}_t^\top \left(\mathbf{\Sigma}_t\right)^{-1}\mathbf{a} \,|\, \mathcal{F}_t^-] \\
&\overset{(i)}{\leq} \frac{1}{\rho}\mathbb{E}_{\mathbf{a}_t\sim p_t}[\mathbf{a}^\top \left(\mathbf{\Sigma}_t\right)^{-1}\mathbf{a}_t\mathbf{a}_t^\top \left(\mathbf{\Sigma}_t\right)^{-1}\mathbf{a} \,|\, \mathcal{F}_t^-] \\
&= \frac{1}{\rho}\mathbf{a}^\top \left(\mathbf{\Sigma}_t\right)^{-1} \mathbb{E}_t\left[\mathbf{a}_t\mathbf{a}_t^\top\right] \left(\mathbf{\Sigma}_t\right)^{-1}\mathbf{a} \\
&= \frac{1}{\rho}\mathbf{a}^\top \left(\mathbf{\Sigma}_t\right)^{-1}\mathbf{a} \,,
\end{aligned}$$

where $(i)$ holds because $|r_t| \leq 1$. $\qquad\square$

This allows us to prove the following version of Lemma 5 in [7],

**Lemma 13.** *Let $\{\alpha_\ell\}_{\ell=1}^\infty$ be a sequence of deterministic nonnegative weights satisfying $\alpha_\ell \leq 1$ for all $\ell\in\mathbb{N}$. Let $\delta' = \frac{\delta}{|\mathcal{A}|}$. Then with probability at least $1-\delta$, simultaneously for all $\mathbf{a}\in\mathcal{A}$ and all $t\in\mathbb{N}$,*

$$\sum_{\ell=1}^t \alpha_\ell\widetilde{r}_\ell(\mathbf{a}) \geq \sum_{\ell=1}^t \alpha_\ell \mathbf{a}^\top \boldsymbol{\omega}_\ell - \mathcal{O}\left(\sqrt{\frac{dt}{\rho}\ln\frac{t}{\delta'}} + B_t\ln\frac{t}{\delta'}\right) \,, \tag{12}$$

*where $B_t = \max_{\ell\leq t}\frac{c(d)\alpha_\ell}{\rho\gamma_\ell} + \alpha_\ell$.*

*Proof.* Fix $\mathbf{a}\in\mathcal{A}$, and recall the definition of $\widetilde{r}_\ell(\mathbf{a})$ in Algorithm 3. Define $M_t(\mathbf{a}) = \alpha_t\mathbf{a}^\top \boldsymbol{\omega}_t - \alpha_t\widehat{r}_t(\mathbf{a})$, and notice that $\{M_t(\mathbf{a})\}_{t=1,2,\dots}$ is a martingale difference sequence. Using Lemma 12 (Item 1), along with Assumption B.1, we see that

$$|M_t(\mathbf{a})| \leq \frac{c(d)\alpha_t}{\rho\gamma_t} + \alpha_t \,.$$

Let $V_t(\mathbf{a}) = \sum_{\ell=1}^t \text{Var}[M_\ell(\mathbf{a}) \,|\, \mathcal{F}_\ell^-]$ be the sum of conditional variances of variables $M_\ell(\mathbf{a})$. Using Lemma 50 we see that with probability at least $1-\delta'$, simultaneously for all $t$,

$$\sum_{\ell=1}^t \alpha_\ell\widehat{r}_\ell(\mathbf{a}) \geq \sum_{\ell=1}^t \alpha_\ell \mathbf{a}^\top \boldsymbol{\omega}_\ell - \mathcal{O}\left(\sqrt{V_t\ln\frac{t}{\delta'}} + B_t\ln\frac{t}{\delta'}\right) \,. \tag{13}$$

Since $\mathrm{Var}[M_t(\mathbf{a})\,|\,\mathcal{F}_\ell^-] \leq \mathbb{E}[M_t^2(\mathbf{a})\,|\,\mathcal{F}_\ell^-] \leq \alpha_t^2 \mathbb{E}_t\left[\widehat{r}_t^2(\mathbf{a})\right]$, by Lemma 12 (Item 4) we can write

$$\sqrt{V_t(\mathbf{a})} \leq \sqrt{\sum_{\ell=1}^{t} \frac{\alpha_\ell^2 \mathbf{a}^\top (\mathbf{\Sigma}_\ell)^{-1} \mathbf{a}}{\rho}}$$

$$\leq \sqrt{\left(\frac{1}{\sqrt{dt}} \sum_{\ell=1}^{t} \frac{\alpha_\ell^2 \mathbf{a}^\top (\mathbf{\Sigma}_\ell)^{-1} \mathbf{a}}{\sqrt{\rho}}\right) \sqrt{\frac{dt}{\rho}}}$$

$$\leq \frac{1}{2}\left(\frac{1}{\sqrt{dt}} \sum_{\ell=1}^{t} \frac{\alpha_\ell^2 \mathbf{a}^\top (\mathbf{\Sigma}_\ell)^{-1} \mathbf{a}}{\sqrt{\rho}} + \sqrt{\frac{dt}{\rho}}\right),$$

the last inequality being the arithmetic-geometric inequality $\sqrt{ab} \leq \frac{1}{2}(a+b)$. Substituting back into Eq. (13) gives

$$\sum_{\ell=1}^{t} \alpha_\ell \widehat{r}_\ell(\mathbf{a}) \geq \sum_{\ell=1}^{t} \alpha_\ell \mathbf{a}^\top \boldsymbol{\omega}_\ell - \mathcal{O}\left(\left(\frac{\sum_{\ell=1}^{t} \alpha_\ell^2 \mathbf{a}^\top (\mathbf{\Sigma}_\ell)^{-1} \mathbf{a}}{\sqrt{\rho dt}} + \sqrt{\frac{dt}{\rho}}\right) \sqrt{\ln \frac{t}{\delta'}} + B_t \ln \frac{t}{\delta'}\right)$$

with probability at least $1 - \delta'$ for all $t \in \mathbb{N}$. Since the function $g(t) = \frac{\ln \frac{t}{\delta'}}{t}$ is decreasing for all $t \geq 1$ we conclude that $\frac{\ln \frac{\ell}{\delta'}}{d\ell}$ is a decreasing function of $\ell$. Using this last fact together with the condition $\alpha_\ell \leq 1$ we see that

$$\alpha_\ell \frac{\mathbf{a}^\top (\mathbf{\Sigma}_\ell)^{-1} \mathbf{a}}{\sqrt{\rho d\ell}} \sqrt{\ln \frac{\ell}{\delta'}} \geq \alpha_\ell \frac{\mathbf{a}^\top (\mathbf{\Sigma}_\ell)^{-1} \mathbf{a}}{\sqrt{\rho dt}} \sqrt{\ln \frac{t}{\delta'}} \geq \alpha_\ell^2 \frac{\mathbf{a}^\top (\mathbf{\Sigma}_\ell)^{-1} \mathbf{a}}{\sqrt{\rho dt}} \sqrt{\ln \frac{t}{\delta'}},$$

and therefore

$$\sum_{\ell=1}^{t} \alpha_\ell \widehat{r}_\ell(\mathbf{a}) + \alpha_\ell \frac{\mathbf{a}^\top (\mathbf{\Sigma}_\ell)^{-1} \mathbf{a}}{\sqrt{\rho d\ell}} \sqrt{\ln \frac{\ell}{\delta'}} \geq \sum_{\ell=1}^{t} \alpha_\ell \widehat{r}_\ell(\mathbf{a}) + \alpha_\ell \frac{\mathbf{a}^\top (\mathbf{\Sigma}_\ell)^{-1} \mathbf{a}}{\sqrt{\rho dt}} \sqrt{\ln \frac{t}{\delta'}}$$

$$\geq \sum_{\ell=1}^{t} \alpha_\ell \mathbf{a}^\top \boldsymbol{\omega}_\ell - \mathcal{O}\left(\sqrt{\frac{dt}{\rho} \ln \frac{t}{\delta'}} + B_t \ln \frac{t}{\delta'}\right)$$

with probability at least $1 - \delta'$ for all $t \in \mathbb{N}$. The result follows by taking a union bound over all $\mathbf{a} \in \mathcal{A}$. $\qquad\square$

In particular when all weights $\alpha_\ell = 1$ Lemma 13 implies the following.

**Corollary 14.** *With the same notation as in Lemma 13, with probability at least $1 - \delta$ simultaneously for all $\mathbf{a} \in \mathcal{A}$ and all $t \in \mathcal{N}$,*

$$\sum_{\ell=1}^{t} \widetilde{r}_\ell(\mathbf{a}) \geq \sum_{\ell=1}^{t} \mathbf{a}^\top \boldsymbol{\omega}_\ell - \mathcal{O}\left(\sqrt{\frac{dt}{\rho} \ln \frac{t}{\delta'}} + B_t \ln \frac{t}{\delta'}\right),$$

*where $B_t = \max_{\ell \leq t} \frac{c(d)}{\rho \gamma_\ell} + 1$.*

We now proceed to upper bound $|\widetilde{r}_t(\mathbf{a})|$. This will inform our choice for learning rate $\eta_t$.

**Lemma 15.** *Let $\delta' = \frac{\delta}{|\mathcal{A}|}$. For all $\mathbf{a} \in \mathcal{A}$, $|\widetilde{r}_t(\mathbf{a})| = \mathcal{O}\left(\frac{c(d)}{\rho \gamma_t} + \left(\frac{c(d)}{\gamma_t \sqrt{\rho dt}} \sqrt{\ln \frac{t}{\delta'}}\right)\right).$*

*Proof.* For each $\mathbf{a} \in \mathcal{A}$, we can write

$$|\widetilde{r}_t(\mathbf{a})| = \mathcal{O}\left(|\widehat{r}_t(\mathbf{a})| + \frac{\mathbf{a}^\top (\mathbf{\Sigma}_t)^{-1} \mathbf{a}}{\sqrt{\rho dt}} \sqrt{\ln \frac{t}{\delta'}}\right)$$

$$= \mathcal{O}\left(\frac{c(d)}{\rho \gamma_t} + \left(\frac{c(d)}{\gamma_t \sqrt{\rho dt}} \sqrt{\ln \frac{t}{\delta'}}\right)\right),$$

the last inequality holding as a consequence of Lemma 12. $\qquad\square$

For the analysis of exponential weights, we will insure that $|\eta_t \widetilde{r}_t(\mathbf{a})| \leq 1$ for all $\mathbf{a}$ and $t$. This imposes the restriction of the following form

$$\eta_t = \mathcal{O}\left(\frac{\rho}{\frac{c(d)}{\gamma_t} + \left(\frac{c(d)}{\gamma_t\sqrt{dt}}\sqrt{\rho\ln\frac{t}{\delta'}}\right)}\right) = \mathcal{O}\left(\frac{\rho\gamma_t}{c(d) + \frac{c(d)}{\sqrt{dt}}\sqrt{\rho\ln\frac{t}{\delta'}}}\right). \tag{14}$$

We are now ready to tackle the anytime high probability regret guarantees for Algorithm 3.

**Lemma 16.** *Let the condition in Eq. (14) hold with a nonincreasing sequence of learning rates $\eta_t$. Then for all $\bar{\mathbf{a}} \in \mathcal{A}$ and $t \in \mathbb{N}$*

$$\sum_{\ell=1}^{t} \widetilde{r}_\ell(\bar{\mathbf{a}}) \leq 1 + \frac{\ln(\mathcal{A})}{\eta_t} + \sum_{\ell=1}^{t}\frac{1}{1-\gamma_\ell}\left(\mathbb{E}_{\mathbf{a}\sim p_\ell}\left[\widetilde{r}_\ell(\mathbf{a}) + \eta_\ell\left(\widetilde{r}_\ell(\mathbf{a})\right)^2\Big|\mathcal{F}_\ell\right] - \gamma_\ell\mathbb{E}_{\mathbf{a}\sim p_E(\mathbf{a})}\left[\widetilde{r}_\ell(\mathbf{a})\Big|\mathcal{F}_\ell\right]\right).$$

*Proof.* Recall that

$$w_\ell(\mathbf{a}) = \exp\left(\eta_\ell\sum_{\ell'=1}^{\ell-1}\widetilde{r}_{\ell'}(\mathbf{a})\right)$$

and $W_\ell = \sum_{\mathbf{a}\in\mathcal{A}} w_\ell(\mathbf{a})$. Let us also define

$$w_\ell^-(\mathbf{a}) = \exp\left(\eta_{\ell-1}\sum_{\ell'=1}^{\ell-1}\widetilde{r}_{\ell'}(\mathbf{a})\right)$$

and, $W_\ell^- = \sum_{\mathbf{a}\in\mathcal{A}} w_\ell^-(\mathbf{a})$. Moreover, set for brevity $A_\ell = \ln\left(\frac{W_{\ell+1}^-}{W_\ell}\right)$. We can write

$$\exp(A_\ell) = \frac{W_{\ell+1}^-}{W_\ell}$$

$$= \frac{\sum_{\mathbf{a}\in\mathcal{A}}\exp\left(\eta_\ell\sum_{\ell'=1}^{\ell}\widetilde{r}_{\ell'}(\mathbf{a})\right)}{\sum_{\mathbf{a}\in\mathcal{A}}\exp\left(\eta_\ell\sum_{\ell'=1}^{\ell-1}\widetilde{r}_{\ell'}(\mathbf{a})\right)}$$

$$= \sum_{\mathbf{a}\in\mathcal{A}} q_\ell(\mathbf{a})\exp\left(\eta_\ell\widetilde{r}_\ell(\mathbf{a})\right)$$

$$\leq 1 + \sum_{\mathbf{a}\in\mathcal{A}} q_\ell(\mathbf{a})\eta_\ell\widetilde{r}_\ell(\mathbf{a}) + q_\ell(\mathbf{a})\eta_\ell^2\left(\widetilde{r}_\ell(\mathbf{a})\right)^2$$

the last inequality holding because $e^x \leq 1 + x + x^2$ whenever $|x| \leq 1$. Taking logs and using the fact that $\ln(1 + x) \leq x$ yields

$$A_\ell \leq \eta_\ell\sum_{\mathbf{a}\in\mathcal{A}} q_\ell(\mathbf{a})\widetilde{r}_\ell(\mathbf{a}) + q_\ell(\mathbf{a})\eta_\ell\left(\widetilde{r}_\ell(\mathbf{a})\right)^2$$

$$\overset{(i)}{\leq} \eta_\ell\sum_{\mathbf{a}\in\mathcal{A}} q_\ell(\mathbf{a})\widetilde{r}_\ell(\mathbf{a}) + \frac{p_\ell(\mathbf{a})}{1-\gamma_\ell}\eta_\ell\left(\widetilde{r}_\ell(\mathbf{a})\right)^2$$

$$\overset{(ii)}{=} \frac{\eta_\ell}{1-\gamma_\ell}\sum_{\mathbf{a}\in\mathcal{A}} p_\ell(\mathbf{a})\widetilde{r}_\ell(\mathbf{a}) + p_\ell(\mathbf{a})\eta_\ell\left(\widetilde{r}_\ell(\mathbf{a})\right)^2 - \gamma_\ell p_E(\mathbf{a})\widetilde{r}_\ell(\mathbf{a}),$$

where $(i)$ follows because $q_\ell(\mathbf{a}) \leq \frac{p_\ell(\mathbf{a})}{1-\gamma_\ell}$ and $(ii)$ because $p_\ell = (1-\gamma_\ell)q_\ell(\mathbf{a}) + \gamma_\ell p_E(\mathbf{a})$. Hence

$$\sum_{\ell=1}^{t}\frac{A_\ell}{\eta_\ell} = \sum_{\ell=1}^{t}\frac{1}{\eta_\ell}\ln\left(\frac{W_{\ell+1}^-}{W_\ell}\right)$$

$$\leq \sum_{\ell=1}^{t}\frac{1}{1-\gamma_\ell}\sum_{\mathbf{a}\in\mathcal{A}} p_\ell(\mathbf{a})\widetilde{r}_\ell(\mathbf{a}) + p_\ell(\mathbf{a})\eta_\ell\left(\widetilde{r}_\ell(\mathbf{a})\right)^2 - \gamma_\ell p_E(\mathbf{a})\widetilde{r}_\ell(\mathbf{a}). \tag{15}$$

Now, define the following potential

$$\Phi_\ell(\eta) = \frac{1}{\eta} \ln \left( \frac{1}{|\mathcal{A}|} \sum_{\mathbf{a} \in \mathcal{A}} \exp \left( \eta \sum_{\ell'=1}^{\ell-1} \widetilde{r}_{\ell'}(\mathbf{a}) \right) \right) .$$

Notice that by de L'Hopital's rule, this implies $\lim_{\eta \to 0} \Phi_\ell(\eta) = \Phi_\ell(0) = 1$ for all $\ell$. Let $\eta_0 = 0$. We have

$$1 + \sum_{\ell=1}^{t} \frac{1}{\eta_\ell} \ln \left( \frac{W_{\ell+1}^-}{W_\ell} \right) = \Phi_1(\eta_0) + \sum_{\ell=1}^{t} \Phi_{\ell+1}(\eta_\ell) - \Phi_\ell(\eta_\ell)$$

$$= \left( \sum_{\ell=1}^{t} (\Phi_\ell(\eta_{\ell-1}) - \Phi_\ell(\eta_\ell)) \right) + \Phi_{t+1}(\eta_t) .$$

Next, we now show that for all $\ell$ the function $\Phi_\ell(\eta)$ is an increasing function of $\eta$. To this effect, let $p_\ell^\eta(\mathbf{a}) = \frac{\exp\left(\eta \sum_{\ell'=1}^{\ell-1} \widetilde{r}_{\ell'}(\mathbf{a})\right)}{\sum_{\mathbf{a}' \in \mathcal{A} } \exp\left(\eta \sum_{\ell'=1}^{\ell-1} \widetilde{r}_{\ell'}(\mathbf{a}')\right)}$. Observe that the following relationship holds,

$$\Phi_\ell'(\eta) = \frac{-1}{\eta^2} \ln \left( \frac{1}{|\mathcal{A}|} \sum_{\mathbf{a} \in \mathcal{A}} \exp \left( \eta \sum_{\ell'}^{\ell-1} \widetilde{r}_{\ell'}(\mathbf{a}) \right) \right) + \frac{1}{\eta} \frac{\sum_{\mathbf{a} \in \mathcal{A}} \left[ \sum_{\ell'=1}^{\ell-1} \widetilde{r}_{\ell'}(\mathbf{a}) \right] \exp \left( \eta \sum_{\ell'=1}^{\ell-1} \widetilde{r}_{\ell'}(\mathbf{a}) \right)}{\sum_{\mathbf{a} \in \mathcal{A}} \exp \left( \eta \sum_{\ell'=1}^{\ell-1} \widetilde{r}_{\ell'}(\mathbf{a}) \right)}$$

$$= \frac{1}{\eta^2} \sum_{\mathbf{a} \in \mathcal{A}} p_\ell^\eta(\mathbf{a}) \left( \eta \sum_{\ell'=1}^{\ell-1} \widetilde{r}_{\ell'}(\mathbf{a}) - \ln \left( \frac{1}{|\mathcal{A}|} \sum_{\mathbf{a}' \in \mathcal{A}} \exp \left( \eta \sum_{\ell'=1}^{\ell-1} \widetilde{r}_{\ell'}(\mathbf{a}') \right) \right) \right)$$

$$= \frac{1}{\eta^2} \mathrm{KL} \left( p_\ell^\eta, \mathrm{Uniform}(\mathcal{A}) \right)$$

$$\geq 0 ,$$

where $\mathrm{KL}(\cdot,\cdot)$ denotes the Kullback Leibler divergence between the two distributions at arguments.

Since we are assuming $\eta_\ell \leq \eta_{\ell-1}$, this implies that $\Phi_\ell(\eta_{\ell-1}) \geq \Phi_\ell(\eta_\ell)$. Thus,

$$1 + \sum_{\ell=1}^{t} \frac{1}{\eta_\ell} \ln \left( \frac{W_{\ell+1}^-}{W_\ell} \right) = \left( \sum_{\ell=1}^{t} (\Phi_\ell(\eta_{\ell-1}) - \Phi_\ell(\eta_\ell)) \right) + \Phi_{t+1}(\eta_t) \geq \Phi_{t+1}(\eta_t). \qquad (16)$$

Combining (15) with (16) gives

$$\Phi_{t+1}(\eta_t) \leq 1 + \sum_{\ell=1}^{t} \frac{1}{\eta_\ell} \ln \left( \frac{W_{\ell+1}^-}{W_\ell} \right)$$

$$\leq 1 + \sum_{\ell=1}^{t} \frac{1}{1 - \gamma_\ell} \sum_{\mathbf{a} \in \mathcal{A}} p_\ell(\mathbf{a}) \widetilde{r}_\ell(\mathbf{a}) + p_\ell(\mathbf{a}) \eta_\ell \left( \widetilde{r}_\ell(\mathbf{a}) \right)^2 - \gamma_\ell p_\mathrm{E}(\mathbf{a}) \widetilde{r}_\ell(\mathbf{a}) .$$

Since for any $\bar{\mathbf{a}} \in \mathcal{A}$ the potential $\Phi_{t+1}(\eta_t)$ satisfies,

$$\Phi_{t+1}(\eta_t) = \frac{1}{\eta_t} \ln \left( \frac{1}{|\mathcal{A}|} \sum_{\mathbf{a} \in \mathcal{A}} \exp \left( \eta_t \sum_{\ell=1}^{t} \widetilde{r}_\ell(\mathbf{a}) \right) \right) \geq \sum_{\ell=1}^{t} \widetilde{r}_\ell(\bar{\mathbf{a}}) - \frac{\ln(\mathcal{A})}{\eta_t}$$

we have

$$\sum_{\ell=1}^{t} \widetilde{r}_\ell(\bar{\mathbf{a}}) \leq 1 + \frac{\ln(\mathcal{A})}{\eta_t} + \sum_{\ell=1}^{t} \frac{1}{1 - \gamma_\ell} \sum_{\mathbf{a} \in \mathcal{A}} \left( p_\ell(\mathbf{a}) \widetilde{r}_\ell(\mathbf{a}) + p_\ell(\mathbf{a}) \eta_\ell \left( \widetilde{r}_\ell(\mathbf{a}) \right)^2 - \gamma_\ell p_\mathrm{E}(\mathbf{a}) \widetilde{r}_\ell(\mathbf{a}) \right)$$

The claimed result now follows by simply observing that

$$\sum_{\mathbf{a} \in \mathcal{A}} p_\ell(\mathbf{a}) \widetilde{r}_\ell(\mathbf{a}) + p_\ell(\mathbf{a}) \eta_\ell \left( \widetilde{r}_\ell(\mathbf{a}) \right)^2 = \mathbb{E}_{\mathbf{a} \sim p_\ell} \left[ \widetilde{r}_\ell(\mathbf{a}) + \eta_\ell \left( \widetilde{r}_\ell(\mathbf{a}) \right)^2 \Big| \mathcal{F}_\ell \right] .$$

$$\square$$

In the sequel, we shall impose the restriction

$$\gamma_t \in (0, 1/2] \,, \tag{17}$$

holding for all $t$, so that $\frac{1}{1-\gamma_t} \leq 2$ and $\frac{\gamma_t}{1-\gamma_t} \leq 1$.

To get a high probability anytime bound starting from Lemma 16, we are required to prove high probability bounds for each of the terms **I**, **II**, **III** and **IV** defined below:

$$\mathbf{I} = \sum_{\ell=1}^{t} \widetilde{r}_\ell(\bar{\mathbf{a}}).$$

$$\mathbf{II} = \sum_{\ell=1}^{t} \frac{1}{1-\gamma_\ell} \mathbb{E}_{\mathbf{a}\sim p_\ell} \left[ \widetilde{r}_\ell(\mathbf{a}) | \mathcal{F}_\ell \right].$$

$$\mathbf{III} = \sum_{\ell=1}^{t} \frac{\eta_\ell}{1-\gamma_\ell} \mathbb{E}_{\mathbf{a}\sim p_\ell} \left[ \left( \widetilde{r}_\ell(\mathbf{a}) \right)^2 | \mathcal{F}_\ell \right]$$

$$\mathbf{IV} = -\sum_{\ell=1}^{t} \frac{\gamma_\ell}{1-\gamma_\ell} \mathbb{E}_{\mathbf{a}\sim p_E} \left[ \widetilde{r}_\ell(\mathbf{a}) | \mathcal{F}_\ell \right]$$

We proceed by (upper or lower) bounding the four terms above in turn.

**Bounding term I.** By Corollary 14 with probability at least $1 - \delta$ for all $\bar{\mathbf{a}} \in \mathcal{A}$ and all $t \in \mathbb{N}$ simultaneously,

$$\mathbf{I} \geq \sum_{\ell=1}^{t} \bar{\mathbf{a}}^\top \boldsymbol{\omega}_\ell - \mathcal{O}\left( \sqrt{\frac{dt}{\rho} \ln \frac{t}{\delta'}} - \left( \max_{\ell \leq t} \frac{c(d)}{\rho\gamma_\ell} + 1 \right) \ln \frac{t}{\delta'} \right) \,.$$

Let us denote the event where this bound holds by $\mathcal{E}_{\mathbf{I}}$. The preceding discussion implies $\mathbb{P}\left( \mathcal{E}_{\mathbf{I}} \right) \geq 1 - \delta$.

**Bounding term II.** Recalling the definition of $\widetilde{r}_\ell(\mathbf{a})$, we can write

$$
\begin{aligned}
\mathbf{II} &= \sum_{\ell=1}^{t} \frac{1}{1-\gamma_\ell} \mathbb{E}_{\mathbf{a}\sim p_\ell} \left[ \widetilde{r}_\ell(\mathbf{a}) | \mathcal{F}_\ell \right] \\
&= \sum_{\ell=1}^{t} \frac{1}{1-\gamma_\ell} \mathbb{E}_{\mathbf{a}\sim p_\ell} \left[ \widehat{r}_\ell(\mathbf{a}) + \mathcal{O}\left( \frac{\mathbf{a}^\top \Sigma_\ell^{-1} \mathbf{a}}{\sqrt{\rho d\ell}} \sqrt{\ln\left( \frac{\ell}{\delta'} \right)} \right) | \mathcal{F}_\ell \right] \\
&\stackrel{(i)}{=} \sum_{\ell=1}^{t} \frac{1}{1-\gamma_\ell} \mathbb{E}_{\mathbf{a}\sim p_\ell} \left[ \widehat{r}_\ell(\mathbf{a}) | \mathcal{F}_\ell \right] + \mathcal{O}\left( \frac{1}{(1-\gamma_\ell)} \sqrt{\frac{d}{\rho\ell}} \sqrt{\ln\left( \frac{\ell}{\delta'} \right)} \right) \\
&\stackrel{(ii)}{\leq} \sum_{\ell=1}^{t} \frac{1}{1-\gamma_\ell} \mathbb{E}_{\mathbf{a}\sim p_\ell} \left[ \widehat{r}_\ell(\mathbf{a}) | \mathcal{F}_\ell \right] + \mathcal{O}\left( \sqrt{\frac{dt}{\rho} \ln\left( \frac{t}{\delta'} \right)} \right) \,, \tag{18}
\end{aligned}
$$

where $(i)$ follows from Item 3 in Lemma 12, and in $(ii)$ we have used $\frac{1}{1-\gamma_\ell} \leq 2$, $\ln\left( \frac{\ell}{\delta'} \right) \leq \ln\left( \frac{t}{\delta'} \right)$, along with $\sum_{\ell=1}^{t} \sqrt{\frac{d}{\ell}} \leq 2\sqrt{dt}$.

We are left to prove a high probability upper bound for $\sum_{\ell=1}^{t} \frac{1}{1-\gamma_\ell} \mathbb{E}_{\mathbf{a}\sim p_\ell} \left[ \widehat{r}_\ell(\mathbf{a}) | \mathcal{F}_\ell \right]$ which we achieve through the following Lemma.

**Lemma 17.** *With probability at least $1 - \delta$ for all $t \in \mathbb{N}$,*

$$\sum_{\ell=1}^{t} \frac{1}{1-\gamma_\ell} \mathbb{E}_{\mathbf{a}\sim p_\ell} \left[ \widehat{r}_\ell(\mathbf{a}) | \mathcal{F}_\ell \right] \leq \sum_{\ell=1}^{t} \frac{r_\ell}{1-\gamma_\ell} + \mathcal{O}\left( \sqrt{\frac{dt}{\rho} \ln \frac{t}{\delta}} + \max_{\ell \leq t} \left( \frac{c(d)}{\rho\gamma_\ell} + 1 \right) \ln \frac{t}{\delta} \right) \,.$$

*Proof.* Let $\bar{\mathbf{a}}_\ell = \mathbb{E}_{\mathbf{a}\sim p_\ell}[\mathbf{a}|\mathcal{F}_\ell]$ where the samples $\mathbf{a} \sim p_\ell$ are conditionally independent from $\mathbf{a}_\ell$. Observe that

$$\sum_{\ell=1}^{t} \frac{1}{1-\gamma_\ell} \mathbb{E}_{\mathbf{a}\sim p_\ell}[\widehat{r}_\ell(\mathbf{a})|\mathcal{F}_\ell] = \sum_{\ell=1}^{t} \frac{1}{1-\gamma_\ell} \mathbb{E}_{\mathbf{a}\sim p_\ell}[\widehat{\boldsymbol{\omega}}_\ell^\top \mathbf{a}|\mathcal{F}_\ell] = \sum_{\ell=1}^{t} \frac{1}{1-\gamma_\ell} \widehat{\boldsymbol{\omega}}_\ell^\top \bar{\mathbf{a}}_\ell .$$

The proof of this lemma follows closely the proof of Lemma 6 in [7]. Consider the martingale difference sequence $Y_\ell = \frac{\widehat{\boldsymbol{\omega}}_\ell^\top \bar{\mathbf{a}}_\ell - r_\ell}{1-\gamma_\ell}$ with respect to the filtration $\{\mathcal{F}_\ell^-\}_{\ell=1}^{\infty}$, where we recall that $\widehat{\boldsymbol{\omega}}_\ell = b_\ell \frac{r_\ell(\boldsymbol{\Sigma}_\ell)^{-1}\mathbf{a}_\ell}{\rho}$. The process $\{Y_\ell\}_{\ell=1}^{\infty}$ is a martingale difference sequence w.r.t. the filtration $\{\mathcal{F}_\ell^-\}_{\ell=1}^{\infty}$ since $\mathbb{E}[\widehat{\boldsymbol{\omega}}_\ell^\top \bar{\mathbf{a}}_\ell|\mathcal{F}_\ell^-] = \boldsymbol{\omega}_\ell^\top \bar{\mathbf{a}}_\ell = \mathbb{E}[r_\ell|\mathcal{F}_\ell^-]$, and therefore $\mathbb{E}[\widehat{\boldsymbol{\omega}}_\ell^\top \bar{\mathbf{a}}_\ell - r_\ell|\mathcal{F}_\ell^-] = 0$.

The conditional variance of $Y_\ell$ can be bounded as follows:

$$\begin{aligned}
\mathrm{Var}[Y_\ell|\mathcal{F}_\ell^-] &= \mathbb{E}\left[(Y_\ell)^2|\mathcal{F}_\ell^-\right] \\
&= \frac{\mathbb{E}\left[(\widehat{\boldsymbol{\omega}}_\ell^\top \bar{\mathbf{a}}_\ell - r_\ell)^2|\mathcal{F}_\ell^-\right]}{(1-\gamma_\ell)^2} \\
&\overset{(i)}{\leq} 4\mathbb{E}\left[(\widehat{\boldsymbol{\omega}}_\ell^\top \bar{\mathbf{a}}_\ell)^2|\mathcal{F}_\ell^-\right] \\
&\overset{(ii)}{\leq} \frac{4\bar{\mathbf{a}}_\ell^\top \Sigma_\ell^{-1} \bar{\mathbf{a}}_\ell}{\rho} \\
&\overset{(iii)}{\leq} \frac{4\mathbb{E}_{\mathbf{a}\sim p_\ell}[\mathbf{a}^\top \Sigma_\ell^{-1}\mathbf{a}|\mathcal{F}_{\ell-1}]}{\rho} \\
&\overset{(iv)}{=} \frac{4d}{\rho} ,
\end{aligned}$$

where: $(i)$ holds because $\mathbb{E}\left[(\widehat{\boldsymbol{\omega}}_\ell^\top \bar{\mathbf{a}}_\ell - r_\ell)^2|\mathcal{F}_\ell^-\right] \leq \mathbb{E}\left[(\widehat{\boldsymbol{\omega}}_\ell^\top \bar{\mathbf{a}}_\ell)^2|\mathcal{F}_\ell^-\right]$ and $\frac{1}{1-\gamma_\ell} \leq 2$; $(ii)$ is a consequence of Item 4 of Lemma 12 (treating $\bar{\mathbf{a}}_\ell$ as a fixed vector); $(iii)$ holds by Jensen's inequality; $(iv)$ holds by Item 3 of Lemma 12. Thus, $\mathrm{Var}[Y_\ell|\mathcal{F}_\ell^-] \leq \frac{4d}{\rho}$

As a consequence, $\sum_{\ell=1}^{t} \mathrm{Var}[Y_\ell|\mathcal{F}_\ell^-] \leq \frac{4td}{\rho}$. Furthermore, $|Y_\ell| \leq \frac{2c(d)}{\rho\gamma_\ell} + 2$ by Item 1 in Lemma 12 and because $\frac{1}{1-\gamma_\ell} \leq 2$.

We are in a position to apply Lemma 51 (setting therein $V_t = \frac{4td}{\rho}$ and $B_t = \frac{2c(d)}{\rho\gamma_t} + 2$ ) to the martingale differences sequence $\{Y_\ell\}_{\ell=1}^{\infty}$. Rearranging terms this gives the claimed bound. $\quad\square$

Denote by $\mathcal{E}_{\mathrm{II}}$ the event where the bound of Lemma 17 holds. By the previous result we have $\mathbb{P}(\mathcal{E}_{\mathrm{II}}) \geq 1 - \delta$. Lemma 17 along with $|r_\ell| \leq 1$ for all $\ell$ (see Assumption B.1) together imply the following.

**Corollary 18.** *If $\mathcal{E}_{\mathrm{II}}$ holds then*

$$\sum_{\ell=1}^{t} \frac{1}{1-\gamma_\ell} \mathbb{E}_{\mathbf{a}\sim p_\ell}[\widehat{r}_\ell(\mathbf{a})|\mathcal{F}_\ell] \leq \sum_{\ell=1}^{t} r_\ell + 2\gamma_\ell + \mathcal{O}\left(\sqrt{\frac{dt}{\rho}\ln\frac{t}{\delta}} + \max_{\ell\leq t}\left(\frac{c(d)}{\rho\gamma_\ell} + 1\right)\ln\frac{t}{\delta}\right) .$$

*Proof.* Since $\frac{1}{1-\gamma_\ell} - 1 = \frac{\gamma_\ell}{1-\gamma_\ell}$ we immediately see that

$$\sum_{\ell=1}^{t} \frac{r_\ell}{1-\gamma_\ell} = \sum_{\ell=1}^{t} r_\ell + \frac{\gamma_\ell r_\ell}{1-\gamma_\ell}$$

And from $|r_\ell| \leq 1$ and $\frac{1}{1-\gamma_\ell} \leq 2$,

$$\sum_{\ell=1}^{t} \frac{r_\ell}{1-\gamma_\ell} = \sum_{\ell=1}^{t} r_\ell + 2\gamma_\ell ,$$

thereby concluding the proof. $\quad\square$

Finally, (18) and Corollary 18 imply that, in case $\mathcal{E}_{\mathbf{II}}$ holds,

$$\mathbf{II} \leq \sum_{\ell=1}^{t} r_\ell + 2\gamma_\ell + \mathcal{O}\left(\sqrt{\frac{dt}{\rho}\ln\frac{t}{\delta}} + \max_{\ell \leq t}\left(\frac{c(d)}{\rho\gamma_\ell} + 1\right)\ln\frac{t}{\delta} + \sqrt{\frac{dt}{\rho}\ln\left(\frac{t}{\delta'}\right)}\right) .$$

**Bounding term III.** By definition of $\widetilde{r}_\ell(\mathbf{a})$, the fact that $\frac{1}{1-\gamma_\ell} \leq 2$, along with the inequality $(a+b)^2 \leq 2a^2 + 2b^2$, we can write

$$\sum_{\ell=1}^{t}\frac{\eta_\ell}{1-\gamma_\ell}\mathbb{E}_{\mathbf{a}\sim p_\ell}\left[\widetilde{r}_\ell^2(\mathbf{a})|\mathcal{F}_\ell\right] \leq \sum_{\ell=1}^{t}4\eta_\ell\mathbb{E}_{\mathbf{a}\sim p_\ell}\left[(\widehat{r}_\ell(\mathbf{a}))^2 + \frac{4\ln\left(\frac{12\ell^2}{\delta'}\right)}{\rho d\ell}\left(\mathbf{a}^\top\Sigma_\ell^{-1}\mathbf{a}\right)^2\Big|\mathcal{F}_\ell\right]$$

$$= \underbrace{\sum_{\ell=1}^{t}4\eta_\ell\mathbb{E}_{\mathbf{a}\sim p_\ell}\left[(\widehat{r}_\ell(\mathbf{a}))^2\Big|\mathcal{F}_\ell\right]}_{\mathbf{A}}$$

$$+ \underbrace{\mathcal{O}\left(\sum_{\ell=1}^{t}\eta_\ell\mathbb{E}_{\mathbf{a}\sim p_\ell}\left[\frac{\left(\mathbf{a}^\top\Sigma_\ell^{-1}\mathbf{a}\right)^2\ln\left(\frac{\ell}{\delta'}\right)}{\rho d\ell}\Big|\mathcal{F}_\ell\right]\right)}_{\mathbf{B}} .$$

We proceed to upper bound $\mathbf{A}$ and $\mathbf{B}$ separately. Let us start from term $\mathbf{B}$. We have

$$\mathbf{B} \overset{(i)}{=} \mathcal{O}\left(\sum_{\ell=1}^{t}\frac{\eta_\ell c(d)\ln\left(\frac{\ell}{\delta'}\right)}{\rho\ell\gamma_\ell}\mathbb{E}_{\mathbf{a}\sim p_\ell}\left[\mathbf{a}^\top\Sigma_\ell^{-1}\mathbf{a}\right]\right) \overset{(ii)}{=} \sum_{\ell=1}^{t}\mathcal{O}\left(\frac{\eta_\ell c(d)d\ln\left(\frac{\ell}{\delta'}\right)}{\rho\ell\gamma_\ell}\right) , \qquad (19)$$

where $(i)$ follows from Item 2. of Lemma 12, and $(ii)$ follows from Item 3. of the same lemma. Notice that this upper bound holds deterministically. Let us now turn to handling term $\mathbf{A}$ now. We use a similar argument as Lemma 8 in [7].

**Lemma 19.** *With probability at least $1 - \delta$ simultaneously for all $t \in \mathbb{N}$,*

$$\mathbf{A} = \mathcal{O}\left(\sum_{\ell=1}^{t}\frac{\eta_\ell d}{\rho} + \sqrt{\ln\left(\frac{t}{\delta}\right)\sum_{\ell=1}^{t}\frac{\eta_\ell^2 c(d)d}{\gamma_\ell\rho^3}} + \max_{\ell \leq t}\frac{c(d)\eta_\ell}{\rho^2\gamma_\ell}\ln\frac{t}{\delta}\right) . \qquad (20)$$

*Proof.* Recalling that $\widehat{\boldsymbol{\omega}}_\ell = b_\ell\frac{r_\ell(\Sigma_\ell)^{-1}\mathbf{a}_\ell}{\rho}$, we first observe that

$$\mathbb{E}_{\mathbf{a}\sim p_\ell}\left[(\widehat{r}_\ell(\mathbf{a}))^2\Big|\mathcal{F}_\ell\right] = \sum_{\mathbf{a}\in\mathcal{A}}p_\ell(\mathbf{a})\widehat{\boldsymbol{\omega}}_\ell^\top\mathbf{a}\mathbf{a}^\top\widehat{\boldsymbol{\omega}}_\ell$$

$$= \widehat{\boldsymbol{\omega}}_\ell^\top\left(\sum_{\mathbf{a}\in\mathcal{A}}p_\ell(\mathbf{a})\mathbf{a}\mathbf{a}^\top\right)\widehat{\boldsymbol{\omega}}_\ell$$

$$= \frac{r_\ell^2 b_\ell^2}{\rho^2}\mathbf{a}_\ell^\top\Sigma_\ell^{-1}\Sigma_\ell\Sigma_\ell^{-1}\mathbf{a}_\ell$$

$$= \frac{r_\ell^2 b_\ell}{\rho^2}\mathbf{a}_\ell^\top\Sigma_\ell^{-1}\Sigma_\ell\Sigma_\ell^{-1}\mathbf{a}_\ell$$

$$\leq \frac{b_\ell\mathbf{a}_\ell^\top\Sigma_\ell^{-1}\mathbf{a}_\ell}{\rho^2} .$$

Summing over $\ell$ and multiplying by $4\eta_\ell$ yields

$$\mathbf{A} \leq \sum_{\ell=1}^{t}\frac{4\eta_\ell}{\rho^2}b_\ell\mathbf{a}_\ell^\top\Sigma_\ell^{-1}\mathbf{a}_\ell .$$

Now, Item 2 of Lemma 12 implies the magnitude of each of the terms $\frac{4\eta_\ell}{\rho^2}b_\ell\mathbf{a}_\ell^\top\Sigma_\ell^{-1}\mathbf{a}_\ell$ is at most $4\frac{c(d)\eta_\ell}{\rho^2\gamma_\ell}$. Moreover, Item 3 of Lemma 12 implies that for each term the conditional expectation

$\mathbb{E}\left[\frac{4\eta_\ell}{\rho^2}b_\ell\mathbf{a}_\ell^\top\Sigma_\ell^{-1}\mathbf{a}_\ell\Big|\mathcal{F}_\ell^-\right]$ equals $\frac{4\eta_\ell d}{\rho}$. As for the conditional variance, we can write

$$\text{Var}\left[\frac{4\eta_\ell}{\rho^2}b_\ell\mathbf{a}_\ell^\top\Sigma_\ell^{-1}\mathbf{a}_\ell - \frac{4\eta_\ell d}{\rho}\Big|\mathcal{F}_\ell^-\right] \leq \frac{16\eta_\ell^2}{\rho^4}\mathbb{E}\left[b_\ell^2\left(\mathbf{a}_\ell^\top\Sigma_\ell^{-1}\mathbf{a}_\ell\right)^2\Big|\mathcal{F}_\ell^-\right]$$

$$\overset{(i)}{\leq} \frac{16\eta_\ell^2 c(d)}{\gamma_\ell\rho^4}\mathbb{E}\left[b_\ell\mathbf{a}_\ell^\top\Sigma_\ell^{-1}\mathbf{a}_\ell|\mathcal{F}_\ell^-\right]$$

$$\overset{(ii)}{=} \frac{16\eta_\ell^2 c(d)d}{\gamma_\ell\rho^3}$$

where $(i)$ follows from Item 2 in Lemma 12, and $(ii)$ is from Item 3. An application of Lemma 51 concludes the proof $\qquad\square$

We denote by $\mathcal{E}_{\mathbf{III}}$ the event where the bound of Lemma 19 holds. By the previous result, $\mathbb{P}(\mathcal{E}_{\mathbf{III}}) \geq 1 - \delta$. Thus if $\mathcal{E}_{\mathbf{III}}$ holds, Equations 19 and 20 imply,

$$\mathbf{III} = \sum_{\ell=1}^t \frac{\eta_\ell}{1-\gamma_\ell}\mathbb{E}_{\mathbf{a}\sim p_\ell}\left[\widetilde{r}_\ell^2(\mathbf{a})\right]$$

$$= \mathcal{O}\left(\sum_{\ell=1}^t \frac{\eta_\ell d}{\rho} + \sqrt{\ln\left(\frac{t}{\delta}\right)\sum_{\ell=1}^t\frac{\eta_\ell^2 c(d)d}{\gamma_\ell\rho^3}} + \max_{\ell\leq t}\frac{c(d)\eta_\ell}{\rho^2\gamma_\ell}\ln\frac{t}{\delta} + \sum_{\ell=1}^t\frac{\eta_\ell c(d)d\ln\left(\frac{\ell}{\delta'}\right)}{\rho\ell\gamma_\ell}\right)\ .$$

**Bounding Term IV.** Define $\text{supp}(p_E) = \{\mathbf{a}\in\mathcal{A} : p_E(\mathbf{a}) > 0\}$, and recall that

$$\mathbf{IV} = -\sum_{\ell=1}^t\frac{\gamma_\ell}{1-\gamma_\ell}\sum_{\mathbf{a}\in\mathcal{A}}p_{\mathrm{E}}(\mathbf{a})\widetilde{r}_\ell(\mathbf{a})\ .$$

Let $\mathbf{a}$ be any action in $\text{supp}(p_E)$, and set in Lemma 13 $\alpha_\ell = \frac{\gamma_\ell}{1-\gamma_\ell}$. This implies that with probability at least $1 - \delta$,

$$\sum_{\ell=1}^t\frac{\gamma_\ell}{1-\gamma_\ell}\widetilde{r}_\ell(\mathbf{a}) \geq \sum_{\ell=1}^t\frac{\gamma_\ell}{1-\gamma_\ell}\mathbf{a}^\top\omega_\ell - \mathcal{O}\left(\sqrt{\frac{dt}{\rho}\ln\frac{t}{\delta'}} + \left(\frac{c(d)}{\rho}+1\right)\ln\frac{t}{\delta'}\right)$$

$$\geq -2\sum_{\ell=1}^t\gamma_\ell - \mathcal{O}\left(\sqrt{\frac{dt}{\rho}\ln\frac{t}{\delta'}} + \left(\frac{c(d)}{\rho}+1\right)\ln\frac{t}{\delta'}\right)$$

the last inequality following from $|\mathbf{a}^\top\omega_\ell| \leq 1$ and $\frac{1}{1-\gamma_\ell} \leq 2$.

A simple union bound along with the fact that $\sum_{\mathbf{a}\in\text{supp}(p_E)}p_E(\mathbf{a}) = 1$ implies that with probability at least $1 - |\text{supp}(p_E)|\delta$

$$\mathbf{IV} \leq 2\sum_{\ell=1}^t\gamma_\ell + \mathcal{O}\left(\sqrt{\frac{dt}{\rho}\ln\frac{t}{\delta'}} + \left(\frac{c(d)}{\rho}+1\right)\ln\frac{t}{\delta'}\right)\ .$$

Similar to before, we denote by $\mathcal{E}_{\mathbf{IV}}$ the event where this bound holds. By the previous result $\mathbb{P}(\mathcal{E}_{\mathbf{IV}}) \geq 1 - |\text{supp}(p_E)|\delta$.

**Putting it all together.** We plug the bounds so obtained on $\mathbf{I} - \mathbf{IV}$ back into Lemma 16, collect common terms, and overapproximate. We obtain that, when $\mathcal{E}_{\mathbf{I}}\cap\mathcal{E}_{\mathbf{II}}\cap\mathcal{E}_{\mathbf{III}}\cap\mathcal{E}_{\mathbf{IV}}$ holds,

$$\sum_{\ell=1}^t\left(\bar{\mathbf{a}}^\top\omega_\ell - r_\ell\right) = \mathcal{O}\left(\frac{\ln(\mathcal{A})}{\eta_t} + \sqrt{\frac{dt}{\rho}\ln\frac{t}{\delta'}} + \max_{\ell\leq t}\left(\frac{c(d)}{\rho\gamma_\ell}+1\right)\ln\frac{t}{\delta'} + \sum_{\ell=1}^t\gamma_\ell + \sum_{\ell=1}^t\frac{\eta_\ell c(d)d\ln\left(\frac{\ell}{\delta'}\right)}{\rho\ell\gamma_\ell}\right.$$

$$\left. + \sum_{\ell=1}^t\frac{\eta_\ell d}{\rho} + \sqrt{\ln\left(\frac{t}{\delta}\right)\sum_{\ell=1}^t\frac{\eta_\ell^2 c(d)d}{\gamma_\ell\rho^3}} + \max_{\ell\leq t}\frac{c(d)\eta_\ell}{\rho^2\gamma_\ell}\ln\frac{t}{\delta}\right)\ ,$$

with $\delta' = \frac{\delta}{|\mathcal{A}|}$.

Now, recall the restriction on $\eta_t$ as in (14). In order to fulfill this requirement, we set

$$\eta_t = \mathcal{O}\left(\frac{\rho\gamma_t}{c(d) + \frac{c(d)}{\sqrt{dt}}\sqrt{\rho\ln|\mathcal{A}|\ln\frac{t}{\delta}}}\right).$$

This gives

$$\sum_{\ell=1}^{t}\left(\bar{\mathbf{a}}^\top\boldsymbol{\omega}_\ell - r_\ell\right) = \mathcal{O}\Bigg(\frac{c(d)\ln|\mathcal{A}|}{\rho\gamma_t}\left(1 + 2\sqrt{\frac{\rho\ln|\mathcal{A}|}{dt}\ln\frac{t}{\delta}}\right) + \max_{\ell\le t}\left(\frac{c(d)}{\rho\gamma_\ell} + 1\right)\ln|\mathcal{A}|\ln\frac{t}{\delta}$$

$$+ \sum_{\ell=1}^{t}\gamma_\ell + \sum_{\ell=1}^{t}\frac{\ln|\mathcal{A}|\ln\left(\frac{\ell}{\delta}\right)}{\ell} + \sum_{\ell=1}^{t}\frac{\gamma_\ell d}{c(d)} + \sqrt{\frac{dt\ln|\mathcal{A}|}{\rho}\ln\frac{12t^2}{\delta}}$$

$$+ \sqrt{\ln\frac{t}{\delta}\sum_{\ell=1}^{t}\frac{\gamma_\ell d}{c(d)\rho} + \frac{1}{\rho}\ln\frac{t}{\delta}}\Bigg).$$

We now set $\gamma_\ell$ so as to satisfy (17):

$$\gamma_\ell = \min\left\{\sqrt{\frac{c(d)\ln|\mathcal{A}|\ln\frac{\ell}{\delta}}{\rho\ell}}, \frac{1}{2}\right\}.$$

Under the assumption that $c(d) \ge d$ (see below) this gets

$$\sum_{\ell=1}^{t}\left(\bar{\mathbf{a}}^\top\boldsymbol{\omega}_\ell - r_\ell\right) = \mathcal{O}\Bigg(\sqrt{\frac{c(d)t\ln|\mathcal{A}|}{\rho}}\ln\frac{t}{\delta} + \rho\ln|\mathcal{A}|\ln\frac{t}{\delta}\ln t + \sqrt{\frac{dt\ln|\mathcal{A}|}{\rho}}\ln\frac{t}{\delta}$$

$$+ \left(\frac{\ln|\mathcal{A}|}{c(d)}\ln\frac{t}{\delta}\right)^{1/4}t^{1/4}\left(\frac{1}{\rho}\right)^{3/4} + \frac{1}{\rho}\ln\frac{t}{\delta}\Bigg).$$

Now, from [9] (Ch. 5 therein), it is known that with John's exploration, the smallest eigenvalue of $\Sigma_t$ is at least $\frac{\gamma_t}{d}$, so that the function $c(d)$ in Lemma 12 is $\le d$. Moreover, the support of John's exploration distribution has size at most $d(d+1)/2 + 1 \le 2d^2$. Combining with the last displayed equation, and taking a final union bound so as to make the events $\mathcal{E}_{\mathbf{I}}, \mathcal{E}_{\mathbf{II}}, \mathcal{E}_{\mathbf{III}}$, and $\mathcal{E}_{\mathbf{IV}}$ jointly hold concludes the proof of Theorem 10.

## B.2 Exp4 algorithm for finite policy classes

Consider now the case of the Exp4 algorithm from [6]. The algorithm operates with a finite set of policies $\Pi$. An anytime high probability regret guarantee for a biased version of Exp4 can be derived by following a similar pattern as in Section B.1, but it can also be derived, e.g., by modifying the high-probability analysis for Exp3 contained in [25]. The proof is omitted.

**Corollary 20.** *Let the complexity $R(\Pi)$ of the policy space $\Pi$ be defined as $R(\Pi) = \sqrt{|\mathcal{A}|\log|\Pi|}$. Then a version of the Exp4 algorithm from [6] exists that is h-stable in that, for $t \to \infty$ and constant $\rho$ independent of $t$, its regret $\mathrm{Reg}(t)$ satisfies*

$$\mathrm{Reg}(t) = \mathcal{O}\left(R(\Pi)\sqrt{\frac{t}{\rho}\ln\frac{t}{\delta}}\right),$$

*with probability at least $1 - \delta$, where the big-oh hides terms in $t$ which are lower order than $\sqrt{t\log t}$ as $t \to \infty$.*

Regarding extendability and the ability to handle policy removals, this is fairly immediate for Exp4, and we omit the trivial details.

# C  Adversarial Regret Balancing and Elimination

In this section, we provide the proof of Arbe's regret bound for adversarial environments. For convenience, we restate the main theorem here:

**Theorem 4.** *Consider a run of Algorithm 1 with* Arbe$(\delta, 1, 0)$ *and $M$ base algorithms with $1 \leq R(\widetilde{\Pi}_1) \leq \cdots \leq R(\widetilde{\Pi}_M)$ where $\tilde{\Pi}_i$ is the extended version of policy class $\Pi_i$ with $(M - i)$ additional actions. Then with probability at least $1 - \mathrm{poly}(M)\delta$ the regret $\mathrm{Reg}(t, \Pi_M)$ for all rounds $t \geq i_\star$ is bounded by*

$$\mathcal{O}\left( \left( \frac{R(\widetilde{\Pi}_{i_\star})}{R(\widetilde{\Pi}_1)} \sqrt{i_\star} + M \right) R(\widetilde{\Pi}_{i_\star}) \sqrt{i_\star t \ln \frac{t}{\delta}} \right) , \tag{4}$$

*where $i_\star$ is the smallest index of the base algorithm that is $h$-stable.*

The proof of this regret bound relies on the following regret bound of Arbe in between restarts:

**Lemma 21** (Regret per Epoch of Arbe). *Consider a run of Algorithm 1 with* Arbe$(\delta, s, t_0)$ *let $T \in \mathbb{N} \cup \{\infty\}$ be the round when the algorithm restarts ($T = \infty$ if there is no restart). Then the regret against $\Pi_M$ is bounded with probability at least $1 - \mathrm{poly}(M)\delta$ for all $t \in [T]$ simultaneously as*

$$\mathrm{Reg}_{\mathbb{M}}([t_0 + 1, t], \Pi_M) = \mathcal{O}\left( \left( MR(\widetilde{\Pi}_{i_\star}) + \frac{R(\widetilde{\Pi}_{i_\star})^2}{R(\widetilde{\Pi}_1)} \sqrt{i_\star} \right) \sqrt{(t - t_0) \ln \frac{t}{\delta}} + M \ln \frac{\ln t}{\delta} \right).$$

*Further, if $s \geq i^\star$, then the algorithm does not restart, i.e., $T = \infty$.*

With this result, Theorem 4 can be proven quickly:

*Proof of Theorem 4.* Denote by $t_0 = 0$ and $t_i$ the round of the $i$-th restart and $\infty$ if it does not exist for $i \in [i_\star]$. By Lemma 21, there can be at most $i_\star - 1$ restarts and thus $t_{i_\star} = \infty$. The total regret of Arbe$(\delta, 1, 0)$ can be decomposed into the regret between two restarts

$$\mathrm{Reg}_{\mathbb{M}}(t, \Pi_M) = \sum_{i=1}^{i_\star} \mathrm{Reg}([t_{i-1} + 1, \min\{t_i, t\}], \Pi_M) .$$

We can now plug in the bound from Lemma 21 for each term on the RHS which gives

$$\mathrm{Reg}_{\mathbb{M}}(t, \Pi_M) = \sum_{s=1}^{i_\star} \mathcal{O}\left( \left( MR(\widetilde{\Pi}_{i_\star}) + \frac{R(\widetilde{\Pi}_{i_\star})^2}{R(\widetilde{\Pi}_1)} \sqrt{i_\star} \right) \sqrt{\bar{t}_s \ln \frac{t}{\delta}} + M \ln \frac{\ln t}{\delta} \right) . \tag{21}$$

where $\bar{t}_s = \max\{\min\{t_s, t\} - t_{s-1}, 0\}$. We can bound this further using Jensen's inequality and the fact that $\sum_{s=1}^{i_\star} \bar{t}_s = t$ as

$$\mathrm{Reg}_{\mathbb{M}}(t, \Pi_M) = \mathcal{O}\left( \left( \frac{R(\widetilde{\Pi}_{i_\star})}{R(\widetilde{\Pi}_1)} \sqrt{i_\star} + M \right) R(\widetilde{\Pi}_{i_\star}) \sqrt{i_\star t \ln \frac{t}{\delta}} + M i_\star \ln \frac{\ln t}{\delta} \right) \tag{22}$$

When $t \geq i_\star$ the last term is dominated by the others and hence, the proof is complete. $\qquad \square$

## C.1  Proof of Lemma 21

We first show that there are at most $i_\star$ restarts. This is because, due to extendability and h-stability of learners above $i_\star$, the elimination test can never trigger for them. The following lemma makes this argument formal:

**Lemma 22.** *With probability at least $1 - \mathrm{poly}(M)\delta$, the elimination test in Equation 2 never triggers for $i, j$ with $i_\star \leq i < j \leq M$.*

*Proof.* Let $i, j \in [i_\star, M] = \{i_\star, i_\star + 1, \ldots, M\}$ and $t \in \mathbb{N}$ and denote by $t_0$ the round of the last restart before $t$. By definition of $i_\star$ in Section 2.1, base learner $i$ is h-stable and extendable. We consider the event where the regret bound of $i$ holds and where the statements in Lemma 23 hold.

This happens for all $i \geq i_\star$ with probability at least $1 - \text{poly}(M)\delta$. Since $i \geq i_\star$ is h-stable and extendable, we have

$$\max_{\pi \in \widetilde{\Pi}_i} \sum_{\ell=t_0+1}^{t} \mathbb{E}_{a \sim \pi}[r_\ell(a, x_\ell)] - r_\ell(a_\ell^i, x_\ell) \leq cR(\widetilde{\Pi}_i)\sqrt{\frac{t-t_0}{\rho_i} \ln \frac{t}{\delta}}$$

where $c$ is an absolute constant. Since the extended policy class $\widetilde{\Pi}_i$ includes an action $\widetilde{a}_j$ that always follows base learner $\mathbb{A}_j$, i.e., $r_\ell(\widetilde{a}_j, x_\ell) = r_\ell(a_\ell^j, x_\ell)$, we have in particular

$$\sum_{\ell=t_0+1}^{t} \left[ r_\ell(a_\ell^j, x_\ell) - r_\ell(a_\ell^i, x_\ell) \right] \leq cR(\widetilde{\Pi}_i)\sqrt{\frac{t-t_0}{\rho_i} \ln \frac{t}{\delta}} \ .$$

Using now Equation 25 from Lemma 23, we have

$$\widetilde{\text{CRew}}_j(t_0, t) - \widetilde{\text{CRew}}_i(t_0, t) \leq \text{D}_j(t_0, t) + \text{D}_i(t_0, t) + \sum_{\ell=t_0+1}^{t} \left[ r_\ell(a_\ell^j, x_\ell) - r_\ell(a_\ell^i, x_\ell) \right]$$

$$\leq \text{D}_j(t_0, t) + \text{D}_i(t_0, t) + cR(\widetilde{\Pi}_i)\sqrt{\frac{t-t_0}{\rho_i} \ln \frac{t}{\delta}}$$

and thus, the test in Equation 2 does not trigger for $i$ and $j$ in round $t$. $\qquad\square$

We are now ready to do the proof of Lemma 21.

*Proof of Lemma 21.* By Lemma 22 there are at most $i_\star$ restarts and $s \leq i_\star$ at all times. The regret of Arbe in rounds $[t_0 + 1, t]$ against any policy $\pi' \in \Pi_M$ can be written as

$$\sum_{\ell=t_0+1}^{t} \left[ \mathbb{E}_{a \sim \pi'}[r_\ell(a, x_\ell)] - r_\ell(a_\ell, x_\ell) \right] = \sum_{i=s}^{M} \sum_{\ell=t_0+1}^{t} \left[ \rho_i \mathbb{E}_{a \sim \pi'}[r_\ell(a, x_\ell)] - \mathbf{1}\{b_\ell = i\} r_\ell(a_\ell, x_\ell) \right] \ .$$

For $i \geq i_\star$, we can bound the summand on the RHS directly using h-stability and extendability of $i$ as

$$\sum_{\ell=t_0+1}^{t} \left[ \rho_i \mathbb{E}_{a \sim \pi'}[r_\ell(a, x_\ell)] - \mathbf{1}\{b_\ell = i\} r_\ell(a_\ell, x_\ell) \right]$$

$$= \rho_i \sum_{\ell=t_0+1}^{t} \left[ \mathbb{E}_{a \sim \pi'}[r_\ell(a, x_\ell)] - \frac{\mathbf{1}\{b_\ell = i\} r_\ell(a_\ell^i, x_\ell)}{\rho_i} \right]$$

$$\leq \rho_i \sum_{\ell=t_0+1}^{t} \left[ \mathbb{E}_{a \sim \pi'}[r_\ell(a, x_\ell)] - r_\ell(a_\ell^i, x_\ell) \right] + \rho_i \text{D}_i(t_0, t) \qquad\qquad \text{(Lemma 23)}$$

$$\leq cR(\widetilde{\Pi}_i)\sqrt{(t-t_0)\rho_i \ln \frac{t}{\delta}} + \rho_i \text{D}_i(t_0, t) \qquad\qquad \text{(h-stability of } \mathbb{A}_i\text{)}$$

$$= \mathcal{O}\left( R(\widetilde{\Pi}_i)\sqrt{(t-t_0)\rho_i \ln \frac{t}{\delta}} + \sqrt{(t-t_0)\rho_i \ln \frac{\ln t}{\delta}} + \ln \frac{\ln t}{\delta} \right) \qquad \text{(definition of } \text{D}_i\text{)}$$

$$= \mathcal{O}\left( R(\widetilde{\Pi}_i)\sqrt{(t-t_0)\rho_i \ln \frac{t}{\delta}} + \ln \frac{\ln t}{\delta} \right) \qquad\qquad \text{(since } R(\widetilde{\Pi}_i) \geq 1\text{)}$$

$$= \mathcal{O}\left( \sqrt{\frac{t-t_0}{\sum_{j=s}^{M} R(\widetilde{\Pi}_j)^{-2}} \ln \frac{t}{\delta}} + \ln \frac{\ln t}{\delta} \right) \qquad\qquad \text{(definition of } \rho_i\text{)}$$

$$= \mathcal{O}\left( \sqrt{\frac{t-t_0}{R(\widetilde{\Pi}_{i_\star})^{-2}} \ln \frac{t}{\delta}} + \ln \frac{\ln t}{\delta} \right) \qquad\qquad \text{(since } s \leq i_\star\text{)}$$

$$= \mathcal{O}\left( R(\widetilde{\Pi}_{i_\star})\sqrt{(t-t_0) \ln \frac{t}{\delta}} + \ln \frac{\ln t}{\delta} \right) \ . \qquad\qquad (23)$$

For $i < i_\star$, we cannot rely on h-stability and extendability and instead have to use the fact that the misspecification test in Equation 2 did not trigger until the last round where there can be at most a regret of 1. This allows us to bound

$$\sum_{\ell=t_0+1}^{t} \left[\rho_i \mathbb{E}_{a\sim\pi'}[r_\ell(a,x_\ell)] - \mathbf{1}\left\{b_\ell = i\right\} r_\ell(a_\ell, x_\ell)\right]$$

$$= \rho_i \sum_{\ell=t_0+1}^{t} \left[\mathbb{E}_{a\sim\pi'}[r_\ell(a,x_\ell)] - \frac{\mathbf{1}\left\{b_\ell = i\right\} r_\ell(a_\ell^i, x_\ell)}{\rho_i}\right]$$

$$\leq \rho_i \sum_{\ell=t_0+1}^{t} \left[\mathbb{E}_{a\sim\pi'}[r_\ell(a,x_\ell)] - \frac{\mathbf{1}\left\{b_\ell = i_\star\right\} r_\ell(a_\ell^{i_\star}, x_\ell)}{\rho_{i_\star}}\right]$$

$$\qquad + \rho_i \mathrm{D}_i(t_0, t) + \rho_i \mathrm{D}_{i_\star}(t_0, t) + R(\widetilde{\Pi}_i)\sqrt{\rho_i(t-t_0)\ln\frac{t}{\delta}} + 1 \qquad \text{(misspecification test failed)}$$

$$\leq \rho_i \sum_{\ell=t_0+1}^{t} \left[\mathbb{E}_{a\sim\pi'}[r_\ell(a,x_\ell)] - r_\ell(a_\ell^{i_\star}, x_\ell)\right] \qquad\qquad\qquad\text{(Lemma 23)}$$

$$\qquad + 2\rho_i \mathrm{D}_{i_\star}(t_0, t) + R(\widetilde{\Pi}_i)\sqrt{\rho_i(t-t_0)\ln\frac{t}{\delta}} + 1 \qquad\qquad (\mathrm{D}_{i_\star}(t_0,t) \geq \mathrm{D}_i(t_0,t))$$

$$\leq \rho_i c R(\widetilde{\Pi}_{i_\star})\sqrt{\frac{t-t_0}{\rho_{i_\star}}\ln\frac{t}{\delta}} + 2\rho_i \mathrm{D}_{i_\star}(t_0,t) + R(\widetilde{\Pi}_i)\sqrt{\rho_i(t-t_0)\ln\frac{t}{\delta}} + 1 \qquad \text{(h-stability of } \mathbb{A}_{i_\star})$$

$$= \mathcal{O}\left(\rho_i R(\widetilde{\Pi}_{i_\star})\sqrt{\frac{t-t_0}{\rho_{i_\star}}\ln\frac{t}{\delta}} + \sqrt{\rho_i}\sqrt{\frac{\rho_i}{\rho_{i_\star}}(t-t_0)\ln\frac{t}{\delta}} + R(\widetilde{\Pi}_i)\sqrt{\rho_i}\sqrt{(t-t_0)\ln\frac{t}{\delta}} + \ln\frac{\ln t}{\delta}\right)$$

$$\text{(definition of } \mathrm{D}_{i_\star})$$

$$= \mathcal{O}\left(\left(\frac{\rho_i R(\widetilde{\Pi}_{i_\star})}{\sqrt{\rho_{i_\star}}} + R(\widetilde{\Pi}_i)\sqrt{\rho_i}\right)\sqrt{(t-t_0)\ln\frac{t}{\delta}} + \ln\frac{\ln t}{\delta}\right)$$

$$= \mathcal{O}\left(\left(\frac{R(\widetilde{\Pi}_{i_\star})^2}{R(\widetilde{\Pi}_i)} + R(\widetilde{\Pi}_i)\right)\sqrt{\rho_i(t-t_0)\ln\frac{t}{\delta}} + \ln\frac{\ln t}{\delta}\right)$$

$$= \mathcal{O}\left(\frac{R(\widetilde{\Pi}_{i_\star})^2}{R(\widetilde{\Pi}_i)}\sqrt{\rho_i(t-t_0)\ln\frac{t}{\delta}} + \ln\frac{\ln t}{\delta}\right) \qquad\qquad (R(\widetilde{\Pi}_i) \leq R(\widetilde{\Pi}_{i_\star}))$$

Finally, combining both bounds yields

$$\mathrm{Reg}_{\mathbb{M}}([t_0+1, t], \Pi_M) = \mathcal{O}\left(\left(\sum_{i=i_\star}^{M} R(\widetilde{\Pi}_{i_\star}) + \sum_{i=s}^{i_\star-1} \frac{R(\widetilde{\Pi}_{i_\star})^2}{R(\widetilde{\Pi}_i)}\sqrt{\rho_i}\right)\sqrt{(t-t_0)\ln\frac{t}{\delta}} + M\ln\frac{\ln t}{\delta}\right)$$

$$= \mathcal{O}\left(\left(M R(\widetilde{\Pi}_{i_\star}) + \frac{R(\widetilde{\Pi}_{i_\star})^2}{R(\widetilde{\Pi}_1)}\sqrt{i_\star}\right)\sqrt{(t-t_0)\ln\frac{t}{\delta}} + M\ln\frac{\ln t}{\delta}\right).$$

$\qquad\qquad\qquad\qquad\qquad\qquad\qquad\qquad\qquad\qquad\qquad\qquad\qquad\qquad\qquad\qquad\qquad\qquad\qquad\qquad\square$

## C.2 Concentration Bounds on Reward Sequences

**Lemma 23.** *With probability at least $1 - poly(M)\delta$, the following inequalities hold for all base learners $i \in [M]$ for all time steps $t \in \mathbb{N}$ where $\mathbb{A}_i$ was not eliminated yet*

$$\left|\sum_{\ell=t_0+1}^{t}\left[\mathbb{E}_{a\sim\pi_\ell^i}[r_\ell(a,x_\ell)] - r_\ell(a_\ell^i, x_\ell)\right]\right| = \mathcal{O}\left(\sqrt{(t-t_0)\ln\frac{\ln(t-t_0)}{\delta}}\right) \qquad (24)$$

$$\left|\sum_{\ell=t_0+1}^{t}\left[\frac{\mathbf{1}\left\{b_\ell = i\right\} r_\ell(a_\ell, x_\ell)}{\rho_i} - r_\ell(a_\ell^i, x_\ell)\right]\right| = \mathcal{O}\left(\sqrt{\frac{t-t_0}{\rho_i}\ln\frac{\ln(t-t_0)}{\delta}} + \frac{1}{\rho_i}\ln\frac{\ln(t-t_0)}{\delta}\right) \qquad (25)$$

*where $t_0$ is the time of the last restart of Algorithm 1 before $t$ and $\rho_i$ is the probability with which learner $i$ is chosen in round $t$.*

*Proof.* There can be at most $M$ restarts of Algorithm 1. Thus, we can prove the concentration bounds for a single restart and a single base learner $i$ and obtain the statement for all restarts with a union bound over $M^2$.

Consider first Equation 24 and let

$$\mathcal{F}_\ell = \sigma\left(\{r_j, x_j, \pi_1^j, \{a_j^i\}_{i\in[M]}, b_j\}_{j\in[t_0+1,\ell-1]} \cup \{r_\ell, x_\ell, \pi_\ell^i, \{a_\ell^k\}_{k\in[M]\setminus\{i\}}\}\right)$$

be the sigma field of all previous reward functions, contexts and actions as well as the context and action in the current round. Further let $\tau$ be the stopping time w.r.t. $\{\mathcal{F}_\ell\}$ of when the algorithm restarts, and denote

$$X_\ell = \mathbf{1}\left\{\ell \geq \tau\right\}\left(\mathbb{E}_{a\sim\pi_\ell^i}\left[r_\ell(a, x_\ell)\right] - r_\ell(a_\ell^i, x_\ell)\right) .$$

The sequence $\{X_\ell\}_{\ell>t_0}$ is a martingale difference sequence w.r.t. $\{\mathcal{F}_\ell\}_{\ell>t_0}$ and $X_\ell \in [-1, 1]$ almost surely for all $\ell$. Then by Lemma 46 (setting $m = 1$ and $a_t = -1, b_t = 1$) with probability at least $1 - \delta$, we have for all $t > t_0$

$$\sum_{\ell=t_0+1}^{t} X_\ell \leq 1.44\sqrt{(t-t_0)\left(1.4\ln\ln\left(4\left(t-t_0\right)\right) + \ln\frac{5.2}{\delta}\right)} = \mathcal{O}\left(\sqrt{(t-t_0)\ln\frac{\ln(t-t_0)}{\delta}}\right)$$

We can apply the same argument to $-\sum_{\ell=t_0+1}^{t} X_\ell$ which proves Equation 24.

Consider now Equation 25 and let

$$\mathcal{F}_\ell = \sigma\left(\{r_j, x_j, \pi_1^j, \{a_j^i\}_{i\in[M]}, b_j\}_{j\in[t_0+1,\ell-1]} \cup \{r_\ell, x_\ell, \pi_\ell^i, \{a_\ell^k\}_{k\in[M]}\}\right) .$$

Again, let $\tau$ be the stopping time w.r.t. $\{\mathcal{F}_\ell\}_\ell$ of when the algorithm restarts and denote

$$X_\ell = \mathbf{1}\left\{\ell \geq \tau\right\}\left[\frac{\mathbf{1}\left\{b_\ell = i\right\}r_\ell(a_\ell^i, x_\ell)}{\rho_i} - r_\ell(a_\ell^i, x_\ell)\right]$$

which is a martingale difference sequence w.r.t. $\{\mathcal{F}_\ell\}_{\ell>t_0}$. We have $X_\ell \leq \frac{1}{\rho_i}$ almost surely and $\mathbb{E}[X_\ell^2|\mathcal{F}_\ell] \leq \frac{1}{\rho_i}$. By Lemma 47 (with $m = 1/\rho_i$), this implies that with probability at least $1 - \delta$ for all $t > t_0$

$$\sum_{\ell=t_0+1}^{t} X_\ell \leq \underbrace{1.44\sqrt{\frac{t-t_0}{\rho_i}\left(1.4\ln\ln\left(4(t-t_0)\right) + \ln\frac{5.2}{\delta}\right)} + \frac{0.41}{\rho_i}\left(1.4\ln\ln\left(4(t-t_0)\right) + \ln\frac{5.2}{\delta}\right)}_{=:D_i(t_0,t)} .$$

We define the RHS as the precise definition of $D_i(t_0, t)$ used in Algorithm 1. Note that

$$D_i(t_0, t) = \mathcal{O}\left(\sqrt{\frac{t-t_0}{\rho_i}\ln\frac{\ln(t-t_0)}{\delta}} + \frac{1}{\rho_i}\ln\frac{\ln(t-t_0)}{\delta}\right) \tag{26}$$

are required. Finally, we can apply the exact same argument to $-\sum_{\ell=t_0+1}^{t} X_\ell$ which finishes the proof. $\qquad\square$

## D   Adversarial Regret Balancing and Elimination with Best of Both Worlds Regret

In this section, we provide the proofs of the main regret bound for Arbe-Gap in Theorem 5 and describe the Arbe-GapExploit subroutine in detail. We restate the theorem her for convenience:

**Theorem 5.** *Consider a run of Algorithm 2 with inputs $t_0 = 0$, arbitrary policy policy $\widehat{\pi} \in \Pi_M$ and M base learners $\mathbb{A}_1, \ldots, \mathbb{A}_M$. Then with probability at least $1 - poly(M)\delta$, the following two conditions hold for all $t \geq M^2$ simultaneously. In any adversarial or stochastic environment $\mathbb{B}$, the regret is bounded as*

$$\mathrm{Reg}(t, \Pi_M) = \mathcal{O}\left( \left( M + \ln(t) + \frac{R(\widetilde{\Pi}_{i_\star})}{R(\widetilde{\Pi}_1)} \sqrt{i_\star} \right) R(\widetilde{\Pi}_{i_\star}) \sqrt{t(\ln(t) + i_\star) \ln \frac{t}{\delta}} \right) .$$

*If $\mathbb{B}$ is stochastic and there is a unique policy with gap $\Delta > 0$, then the pseudo-regret is bounded as,*

$$\mathrm{PseudoReg}_{\mathbb{M}}(t, \Pi_M) = \mathcal{O}\left( \frac{R(\Pi_M)^2}{\Delta} \ln(t) \ln \frac{t}{\delta} + \frac{R(\widetilde{\Pi}_{i_\star})^2 R(\widetilde{\Pi}_M)^2}{R(\widetilde{\Pi}_1)^2} \frac{M^2 i_\star}{\Delta} \ln^2 \left( \frac{MR(\widetilde{\Pi}_M)}{\Delta\delta} \right) \right).$$
$$(5)$$

We break the proof of this statement into several parts based on the phases of the algorithm. The first phase end when Arbe-Gap calls Arbe-GapExploit and the second phase are all rounds played by Arbe-GapExploit. Finally, in case Arbe-GapExploit was called but terminated at some point, we have a final phase where we simply execute Arbe. For the regret in this final phase, we can directly use the guarantees of Arbe. The behavior in the first two phases is analyzed below. The following lemma characterizes the regret and pseudo-regret in the first phase until Arbe-GapExploit is called. It also ensures that if the environment is stochastic, the inputs of Arbe-GapExploit are correct, i.e., $\widehat{\pi}$ is the optimal policy and $\widehat{\Delta}$ is an accurate estimate of its gap.

**Lemma 24** (Guarantee for First Phase). *Consider a run of Algorithm 2 with inputs $t_0 = 0$, arbitrary policy policy $\widehat{\pi} \in \Pi_M$ and M base learners $\mathbb{A}_1, \ldots, \mathbb{A}_M$. Further, let $t_{\mathrm{gap}} \in \mathbb{N} \cup \{\infty\}$ be the round where the Arbe-GapExploit subroutine is called. Then with probability at least $1 - poly(M)\delta$, the following conditions hold for all rounds $t \in [2i_\star, t_{\mathrm{gap}}]$. The regret is bounded as*

$$\mathrm{Reg}_{\mathbb{M}}(t, \Pi_M) = \mathcal{O}\left( \left( M + \frac{R(\widetilde{\Pi}_{i_\star})}{R(\widetilde{\Pi}_1)} \sqrt{i_\star} \right) R(\widetilde{\Pi}_{i_\star}) \sqrt{t(\ln(t) + i_\star) \ln \frac{t}{\delta}} \right).$$

*If $\mathbb{B}$ is stochastic and there is a unique policy $\pi_\star$ with gap $\Delta > 0$, then the gap estimator $\widehat{\Delta}$ and policy $\widehat{\pi}$ passed onto Arbe-GapExploit satisfy $\frac{\Delta}{2} \leq \widehat{\Delta} \leq \Delta$ and $\widehat{\pi} = \pi_\star$. Further, the pseudo-regret is bounded as*

$$\mathrm{PseudoReg}_{\mathbb{M}}(t, \Pi_M) = \mathcal{O}\left( \frac{R(\widetilde{\Pi}_{i_\star})^2 R(\widetilde{\Pi}_M)^2}{R(\widetilde{\Pi}_1)^2} \frac{M^2 i_\star}{\Delta} \ln \left( \frac{MR(\widetilde{\Pi}_M)}{\Delta\delta} \right) \ln \frac{t}{\delta} \right).$$

The proof of this result can be found in Appendix D.2. To characterize the regret and pseudo-regret of the second phase, we use the following main properties of the Arbe-GapExploit routine in Algorithm 4. It ensures that the regret and pseudoregret are well controlled and that the routine never terminates if the environment was indeed stochastic with a gap.

**Lemma 25** (Guarantee for Second Phase). *Let $\mathbb{A}$ be an h-stable learner with policy class $\Pi_{\mathbb{A}}$. Then the regret of Algorithm 4 against $\Pi_{\mathbb{A}} \cup \{\widehat{\pi}\}$ is bounded with probability at least $1 - \mathcal{O}(\delta)$ for all rounds $t > t_0$ that the algorithm has not terminated yet as*

$$\mathrm{Reg}_{\mathbb{M}}([t_0 + 1, t], \Pi_{\mathbb{A}} \cup \{\widehat{\pi}\}) = \mathcal{O}\left( \frac{R(\Pi_{\mathbb{A}})^2}{\widehat{\Delta}} \left( \ln(t) \ln \frac{t}{\delta} + \ln \frac{R(\Pi_{\mathbb{A}})}{\widehat{\Delta}\delta} \right) + \sqrt{(t - t_0) \ln(t) \ln \frac{t}{\delta}} \right).$$

*Further, if the environment is stochastic with an optimal policy $\pi_\star$ that has a gap $\Delta$ compared to the best policy in $\Pi_{\mathbb{A}}$ and the inputs satisfy $\widehat{\pi} = \pi_\star$ and $\widehat{\Delta} \leq \Delta \leq 2\widehat{\Delta}$, then with probability at least $1 - \mathcal{O}(\delta)$ the pseudo-regret of Algorithm 4 is bounded in all rounds $t > t_0$ as*

$$\mathrm{PseudoReg}_{\mathbb{M}}([t_0 + 1, t], \{\widehat{\pi}\} \cup \Pi_{\mathbb{A}}) = \mathcal{O}\left( \frac{R(\Pi_{\mathbb{A}})^2}{\Delta} \left( \ln(t) \ln \frac{t}{\delta} + \ln \frac{R(\Pi_{\mathbb{A}})}{\Delta\delta} \right) \right)$$

*and the algorithm never terminates.*

The proof of this statement is available in Appendix D.3. While combining Lemma 24 and Lemma 25 gives the desired pseudo-regret guarantee in Theorem 5 above for stochastic environments fairly directly, the bound on the regret in Theorem 5 requires more work. Arbe-GapExploit guarantees only guarantees that the regret is of order $\tilde{\mathcal{O}}(R(\Pi_M \setminus \{\widehat{\pi}\})^2/\widehat{\Delta} + \sqrt{t})$ while we would like a bound that does not scale with $R(\Pi_M \setminus \{\widehat{\pi}\}$ in our final result. To achieve that, we will use the following lemma which states that the length of the initial phase $t_{gap}$ has to be sufficiently large as a function of the gap estimate $\widehat{\Delta}$. This will allow us to absorb the $R(\Pi_M \setminus \{\widehat{\pi}\})^2/\widehat{\Delta}$ term into the regret of the first phase.

**Lemma 26.** *Consider a run of Algorithm 2 with inputs $t_0 = 0, n = 1$, arbitrary policy $\widehat{\pi} \in \Pi_M$ and $M$ base learners $\mathbb{A}_1, \ldots, \mathbb{A}_M$ with $1 \leq R(\widetilde{\Pi}_1) \leq R(\widetilde{\Pi}_2) \leq \cdots \leq R(\widetilde{\Pi}_M)$ where $\widetilde{\Pi}_i$ is the extended version of policy class $\Pi_i$ with $(M + 1 - i)$ additional actions. Let $t_{gap}$ be the round where ArbeGap-Exploit was called with gap estimate $\widehat{\Delta}$. Then with probability at least $1 - poly(M)\delta$*

$$\widehat{\Delta} = \Omega\left(\frac{R(\widetilde{\Pi}_M)^2}{R(\widetilde{\Pi}_{i_\star})}\sqrt{\frac{\ln\frac{t_{gap}}{\delta}}{t_{gap}}}\right).$$

*Proof.* ArbeGap calls ArbeGap-Exploit as soon as $2W(t_0, t) \leq \widehat{\Delta}_t$. Hence, when the call happened in round $t_{gap}$, we must have $2W(t_0, t_{gap}) \leq \widehat{\Delta}$ or, plugging in the definition of $W(t_0, t_{gap})$ with an appropriate absolute constant $c$

$$\frac{cR(\widetilde{\Pi}_M)}{\sqrt{\rho_M}}\sqrt{\frac{\ln\frac{n(k)}{\delta}}{k}} + \frac{c}{\rho_M}\frac{\ln\frac{n\ln(k)}{\delta}}{k} \leq \widehat{\Delta},$$

where $k = t_{gap} - t_0$. We can further lower-bound the LHS as

$$\frac{cR(\widetilde{\Pi}_M)}{\sqrt{\rho_M}}\sqrt{\frac{\ln\frac{nk}{\delta}}{k}} + \frac{c}{\rho_M}\frac{\ln\frac{n\ln(k)}{\delta}}{k} \geq \frac{cR(\widetilde{\Pi}_M)}{\sqrt{\rho_M}}\sqrt{\frac{\ln\frac{k}{\delta}}{k}}$$

$$\geq cR(\widetilde{\Pi}_M)^2\sqrt{\sum_{i=s}^{M}R(\widetilde{\Pi}_i)^{-2}}\sqrt{\frac{\ln\frac{k}{\delta}}{k}}$$

$$\geq c\frac{R(\widetilde{\Pi}_M)^2}{R(\widetilde{\Pi}_{i_\star})^2}\sqrt{\frac{\ln\frac{k}{\delta}}{k}}$$

where the last equation holds because $i_\star$ is never eliminated with high probability Lemma 28. Finally, since the function on the RHS is monotonically decreasing in $k$, we can further lower-bound this quantity by replacing $k$ with $t_{gap} \geq k$. Hence, we have $\widehat{\Delta} \geq c\frac{R(\widetilde{\Pi}_M)^2}{R(\widetilde{\Pi}_{i_\star})^2}\sqrt{\frac{\ln\frac{t_{gap}}{\delta}}{t_{gap}}}$. Reordering terms gives the desires statement. $\qquad\square$

We are now ready to state the proof of Theorem 5:

*Proof of Theorem 5.* We first consider stochastic environments with a gap and apply Lemma 24. We denote by $t_{gap}$ the round where Arbe-Gap calls Arbe-GapExploit. For all $t \leq t_{gap}$, the pseudo-regret is bounded as

$$\text{PseudoReg}_{\mathbb{M}}(t, \Pi_M) = \mathcal{O}\left(\frac{R(\widetilde{\Pi}_{i_\star})^2 R(\widetilde{\Pi}_M)^2}{R(\widetilde{\Pi}_1)^2}\frac{M^2 i_\star}{\Delta}\ln\left(\frac{MR(\widetilde{\Pi}_M)}{\Delta\delta}\right)\ln\frac{t}{\delta}\right).$$

Further, Arbe-GapExploit can only be called with $\Pi_{\mathbb{A}} = \Pi_M \setminus \{\pi_\star\}$, $\widehat{\pi} = \pi_\star$ and $\widehat{\Delta}$ that satisfies $\widehat{\Delta} \leq \Delta \leq 2\widehat{\Delta}$. This allows us to apply Lemma 25 to bound the pseudo-regret of any round played by Arbe-GapExploit as

$$\text{PseudoReg}_{\mathbb{M}}([t_{gap} + 1, t], \Pi_M) = \mathcal{O}\left(\frac{R(\Pi_M)^2}{\Delta}\left(\ln(t)\ln\frac{t}{\delta} + \ln\frac{R(\Pi_M)}{\Delta\delta}\right)\right).$$

It further tells us that, with high probability, Arbe-GapExploit will never return. Hence, we can bound the pseudo-regret of both phases to get a bound on the total pseudo-regret after any number of rounds

$$
\text{PseudoReg}_{\mathbb{M}}(t, \Pi_M) = \mathcal{O}\left( \frac{R(\widetilde{\Pi}_{i_\star})^2 R(\widetilde{\Pi}_M)^2}{R(\widetilde{\Pi}_1)^2} \frac{M^2 i_\star}{\Delta} \ln\left( \frac{M R(\widetilde{\Pi}_M)}{\Delta \delta} \right) \ln \frac{t_{gap}}{\delta} \right.
$$

$$
\left. + \frac{R(\Pi_M)^2}{\Delta} \left( \ln(t) \ln \frac{t}{\delta} + \ln \frac{R(\Pi_M)}{\Delta \delta} \right) \right)
$$

$$
= \mathcal{O}\left( \frac{R(\Pi_M)^2}{\Delta} \ln(t) \ln \frac{t}{\delta} + \frac{R(\widetilde{\Pi}_{i_\star})^2 R(\widetilde{\Pi}_M)^2}{R(\widetilde{\Pi}_1)^2} \frac{M^2 i_\star}{\Delta} \ln^2\left( \frac{M R(\widetilde{\Pi}_M)}{\Delta \delta} \right) \right)
$$

where we upper-bounded $t_{gap}$ using a crude upper-bound $\mathcal{O}\left( \frac{R(\widetilde{\Pi}_M)^6 M^5}{\Delta^4 \delta^2} \right)$ of the bound in Lemma 34 which gives that $\ln(t_{gap}) = \mathcal{O}(\ln(M R(\widetilde{\Pi}_M)/\Delta\delta))$.

We now move on to the regret bound in any environment. Again, Lemma 24 gives us a regret bound that holds with high probability for any round $t \le t_{gap}$ of

$$
\text{Reg}_{\mathbb{M}}(t, \Pi_M) = \mathcal{O}\left( \left( M + \frac{R(\widetilde{\Pi}_{i_\star})}{R(\widetilde{\Pi}_1)} \sqrt{i_\star} \right) R(\widetilde{\Pi}_{i_\star}) \sqrt{t(\ln(t) + i_\star) \ln \frac{t}{\delta}} \right).
$$

If $t$ falls into a round that is played by the routine Arbe-Gap, then Lemma 25 the regret after $t_{gap}$ and before $t$ is bounded as

$$
\text{Reg}_{\mathbb{M}}([t_{gap} + 1, t], \Pi_M) = \mathcal{O}\left( \frac{R(\Pi_M)^2}{\widehat{\Delta}} \left( \ln(t) \ln \frac{t}{\delta} + \ln \frac{R(\Pi_M)}{\widehat{\Delta} \delta} \right) + \sqrt{(t - t_{gap}) \ln(t) \ln \frac{t}{\delta}} \right).
$$

Using Lemma 26, we can further bound $\frac{1}{\widehat{\Delta}} = \mathcal{O}\left( \frac{R(\widetilde{\Pi}_{i_\star})}{R(\widetilde{\Pi}_M)^2} \sqrt{\frac{t_{gap}}{\ln(1/\delta)}} \right)$ and plugging this into the bound above gives

$$
\text{Reg}_{\mathbb{M}}([t_{gap} + 1, t], \Pi_M)
$$

$$
= \mathcal{O}\left( R(\widetilde{\Pi}_{i_\star}) \sqrt{t_{gap}} \left( \ln^{3/2}(t) \ln^{1/2} \frac{t}{\delta} + \ln(t_{gap} R(\Pi_M)) \right) + \sqrt{(t - t_{gap}) \ln(t) \ln \frac{t}{\delta}} \right)
$$

$$
= \mathcal{O}\left( R(\widetilde{\Pi}_{i_\star}) \sqrt{t} \ln^{3/2}(t) \ln^{1/2} \frac{t}{\delta} + \sqrt{t \ln(t) \ln \frac{t}{\delta}} \right).
$$

Here, we also used the fact that $t_{gap} = \Omega(R(\Pi_M))$ since the test in Line 14 of Algorithm 2 can only trigger when $W(t_0, t) \le R(\Pi_M)^2$ which is only possible after at least $\Omega(R(\Pi_M))$ rounds. Finally, if $t$ falls into a round after Arbe-GapExploit returned (in round $t_{adv}$, then the regret since the return can be bounded using the regret bound of Arbe in Theorem 4 as

$$
\text{Reg}_{\mathbb{M}}([t_{adv} + 1, t], \Pi_M) = \mathcal{O}\left( \left( \frac{R(\widetilde{\Pi}_{i_\star})}{R(\widetilde{\Pi}_1)} \sqrt{i_\star} + M \right) R(\widetilde{\Pi}_{i_\star}) \sqrt{i_\star t \ln \frac{t}{\delta}} \right),
$$

Combining the bounds from all three possible phases gives the following bound that holds for all $t > M^2$ as

$$
\text{Reg}_{\mathbb{M}}(t, \Pi_M) = \mathcal{O}\left( \left( M + \frac{R(\widetilde{\Pi}_{i_\star})}{R(\widetilde{\Pi}_1)} \sqrt{i_\star} \right) R(\widetilde{\Pi}_{i_\star}) \sqrt{t(\ln(t) + i_\star) \ln \frac{t}{\delta}} \right.
$$

$$
\left. + R(\widetilde{\Pi}_{i_\star}) \sqrt{t} \ln^{3/2}(t) \ln^{1/2} \frac{t}{\delta} \right)
$$

$$
= \mathcal{O}\left( \left( \frac{R(\widetilde{\Pi}_{i_\star})}{R(\widetilde{\Pi}_1)} \sqrt{i_\star} + M + \ln(t) \right) R(\widetilde{\Pi}_{i_\star}) \sqrt{t(\ln(t) + i_\star) \ln \frac{t}{\delta}} \right)
$$

$\square$

---

**Algorithm 4:** Arbe-GapExploit

---

1 **Input:** current round $t_0$, learner $\mathbb{A}$ with policy class $\Pi_{\mathbb{A}}$, candidate policy $\widehat{\pi}$, gap estimate $\widehat{\Delta}$, failure probability $\delta$

2 Initialize $k_0 = \Theta\left(\frac{R^2(\Pi_{\mathbb{A}})}{\widehat{\Delta}^2} \ln \frac{R(\Pi_{\mathbb{A}})}{\widehat{\Delta}\delta}\right)$

3 **for** *epoch* $e = 0, 1, 2 \dots$ **do**

4      Set next epoch length $k_{e+1} = 2k_e$ and final round $t_{e+1} = t_e + k_e$

5      Set learner probability $\rho^e = \Theta\left(\frac{R^2(\Pi_{\mathbb{A}})}{k_e \widehat{\Delta}^2} \ln \frac{k_e}{\delta_e}\right)$ where $\delta_e = \frac{\delta}{(e+1)^2}$

6      Restart $\mathbb{A}$ with failure probability $\delta_e$

7      **for** *round* $t = t_e + 1, t_e + 2, \dots, t_{e+1}$ **do**

8          Set $\pi_t^1$ as the current policy of $\mathbb{A}$ and $\pi_t^0 = \widehat{\pi}$

9          Sample $b_t \sim \text{Bernoulli}(\rho^e)$

10         Get context $x_t$ and compute $a_t^i \sim \pi_t^i(\cdot|x_t)$ for $i \in \{0, 1\}$

11         Play action $a_t = a_t^{b_t}$ and receive reward $r_t(a_t, x_t)$

12         Update learner $\mathbb{A}$ with reward $\frac{b_t r_t(a_t, x_t)}{\rho^e}$

13         Set $V(t) = \Theta\left(R(\Pi_{\mathbb{A}})\sqrt{\frac{\ln \frac{t-t_e}{\delta_e}}{\rho^e(t-t_e)}} + \frac{\ln \frac{\ln(t-t_e)}{\delta_e}}{t-t_e}\right)$

14         **if** $\frac{\widetilde{\text{CRew}}_0(t_e+1,t) - \widetilde{\text{CRew}}_1(t_e+1,t)}{t-t_e} < \widehat{\Delta} - V(t)$ **then**

15            **return** // environment is adversarial

16         **if** $\frac{\widetilde{\text{CRew}}_0(t_e+1,t) - \widetilde{\text{CRew}}_1(t_e+1,t)}{t-t_e} > 4\widehat{\Delta} + V(t)$ **then**

17            **return** // environment is adversarial

---

### D.1 Description of the Second Phase: Gap Exploitation

We present the Arbe-GapExploit algorithm of the Gap Exploitation phase as a general procedure that takes a focus policy $\widehat{\pi}$, a policy class $\Pi_{\mathbb{A}}$ and a gap estimate $\widehat{\Delta}$ and is tasked with testing the hypothesis '$\widehat{\pi}$ is the optimal policy, and has a gap of order $\Theta(\widehat{\Delta})$', incurring in small regret while doing so.

We use Arbe-GapExploit with input policy class $\Pi_{\mathbb{A}} = \Pi_M$. If it ever returns, the learner can conclude the environment is adversarial and thus, start playing Arbe with the policy classes $\Pi_s, \dots, \Pi_M$ not yet eliminated by the misspecification tests during Arbe-Gap. We develop results for the more general case when the input algorithm and policy class equal $\mathbb{A}$ and $\Pi_{\mathbb{A}}$. In case the environment is stochastic with gap $\Delta$ and $\widehat{\Delta} = \Theta(\Delta)$, we show Arbe-GapExploit has a pseudo regret of order $\mathcal{O}\left(\frac{R(\Pi_{\mathbb{A}})^2}{\Delta}\left(\ln(t)\ln\frac{t}{\delta} + \ln\frac{R(\Pi_{\mathbb{A}})}{\Delta\delta}\right)\right)$ at time $t$ (see Lemma 38). Similarly we show that Arbe-GapExploit has an adversarial regret rate of order $\mathcal{O}\left(\frac{R(\Pi_{\mathbb{A}})^2}{\widehat{\Delta}}\left(\ln(t)\ln\frac{t}{\delta} + \ln\frac{R(\Pi_{\mathbb{A}})}{\widehat{\Delta}\delta}\right) + \sqrt{(t-t_0)\ln(t)\ln\frac{t}{\delta}}\right)$ at time $t$ (see Lemma 41). The adversarial rate consists of a poly-logarithmic factor with an upfront multiplier of order $\frac{R^2(\Pi_{\mathbb{A}})}{\widehat{\Delta}}$ plus a factor scaling with $\sqrt{t - t_0}$. Since in our case $\widehat{\Delta} = \Omega\left(\frac{R(\widetilde{\Pi}_M)^2}{R(\widetilde{\Pi}_{i_\star})}\sqrt{\frac{M \ln \frac{t_{gap}}{\delta}}{t_{gap}}}\right)$ (see Lemma 26) and $\Pi_{\mathbb{A}} = \Pi_M$, the adversarial regret has an upper bound of the form $\widetilde{\mathcal{O}}\left(R(\widetilde{\Pi}_{i_\star})\sqrt{t_{\text{gap}}} + \sqrt{t - t_{\text{gap}}}\right)$ (where $\widetilde{\mathcal{O}}$ hides polylogarithmic factors) thus satisfying a model selection guarantee.

In Arbe-GapExploit we divide time into epochs indexed from $e = 0, 1, \dots$ of length $k_e = k_0 \cdot 2^e$ where $k_0 = \Theta\left(\frac{R^2(\widetilde{\Pi}_{\mathbb{A}}) \log \frac{R(\widetilde{\Pi}_{\mathbb{A}})}{\widehat{\Delta}\delta}}{\widehat{\Delta}^2}\right)$. During epoch $e$, learner $\mathbb{A}$ is sampled with probability $\rho^e = \Theta\left(\frac{R^2(\widetilde{\Pi}_{\mathbb{A}}) \log\left(\frac{k_e}{\delta_e}\right)}{k_e \widehat{\Delta}^2}\right)$. We define $k_0$ and $\rho^e$ so that for all epochs $1 - \rho^e \geq \frac{1}{2}$. Thus

for all $t \in \{t_e + 1, \cdots, t_{e+1}\}$, it holds that $\left| \widetilde{\mathrm{CRew}}_0(t_e + 1, t) - \sum_{\ell=t_e+1}^{t} \mathbb{E}_{a \sim \widehat{\pi}} \left[ r_\ell(a, x_\ell) \right] \right| = \mathcal{O}\left( \sqrt{(t - t_e) \ln \frac{t - t_e}{\delta_e}} \right)$. By the h-stability of $\mathbb{A}$, and using a concentration argument, we prove that $\widetilde{\mathrm{CRew}}_1(t_e + 1, t)$ can be used to estimate $\max_{\pi \in \Pi_{\mathbb{A}} \setminus \{\widehat{\pi}\}} \sum_{\ell=t_e+1}^{t} \mathbb{E}_{a \sim \pi} \left[ r_\ell(a, x_\ell) \right]$ up to

$$V(t) = \Theta\left( R(\Pi_{\mathbb{A}}) \sqrt{\frac{\ln \frac{t - t_e}{\delta_e}}{\rho^e (t - t_e)}} + \frac{\ln \frac{\ln(t - t_e)}{\delta_e}}{t - t_e} \right) \text{ accuracy.}$$

When the environment is stochastic, $\widehat{\pi}$ is the optimal policy and $\Delta/2 \leq \widehat{\Delta} \leq \Delta$ it follows that

$$(t - t_e)\widehat{\Delta} \leq \sum_{\ell=t_e+1}^{t} \mathbb{E}_{a \sim \widehat{\pi}, x \sim \mathcal{D}} \left[ r(a, x) \right] - \max_{\pi \in \Pi_{\mathbb{A}} \setminus \{\widehat{\pi}\}} \sum_{\ell=t_e+1}^{t} \mathbb{E}_{a \sim \pi, x \sim \mathcal{D}} \left[ r(a, x) \right] \leq 4\widehat{\Delta} .$$

Therefore when the condition in line 14 or 16 of Algorithm 4 trigger, we would have found evidence that

$$\sum_{\ell=t_e+1}^{t} \mathbb{E}_{a \sim \widehat{\pi}, x \sim \mathcal{D}} \left[ r(a, x) \right] - \max_{\pi \in \Pi_{\mathbb{A}} \setminus \{\widehat{\pi}\}} \sum_{\ell=t_e+1}^{t} \mathbb{E}_{a \sim \pi, x \sim \mathcal{D}} \left[ r(a, x) \right] < (t - t_e)\widehat{\Delta}$$

or

$$\sum_{\ell=t_e+1}^{t} \mathbb{E}_{a \sim \widehat{\pi}, x \sim \mathcal{D}} \left[ r(a, x) \right] - \max_{\pi \in \Pi_{\mathbb{A}} \setminus \{\widehat{\pi}\}} \sum_{\ell=t_e+1}^{t} \mathbb{E}_{a \sim \pi, x \sim \mathcal{D}} \left[ r(a, x) \right] > 4(t - t_e)\widehat{\Delta}$$

thus indicating the environment cannot be stochastic.

The observations above imply that in case the environment is stochastic the tests of lines 14 and 16 in Algorithm 4 do not trigger. Let us jump to the task of analyzing the regret of Arbe-GapExploit in stochastic environments. We will assume $t$ lies in epoch $e$. By the h-stability of $\mathbb{A}$, the sum of its pseudo-rewards in a stochastic environment satisfies

$$\sum_{\ell=t_e+1}^{t} \mathbb{E}_{a \sim \pi_\ell^1, x \sim \mathcal{D}} \left[ r(a, x) \right] + \underbrace{\mathcal{O}\left( R(\Pi_{\mathbb{A}} \setminus \{\widehat{\pi}\}) \sqrt{\frac{(t - t_e)}{\rho^e} \ln \frac{t - t_e}{\delta_e}} \right)}_{\mathbf{I}} \geq \max_{\pi \in \Pi_{\mathbb{A}} \setminus \{\widehat{\pi}\}} \sum_{\ell=t_e+1}^{t} \mathbb{E}_{a \sim \pi, x \sim \mathcal{D}} \left[ r(a, x) \right] .$$

In this case, pseudo-regret is only incurred when $b_t = 1$. From $t_e + 1$ to $t$ the variable $b_t$ equals 1 an average of $\rho^e(t - t_e)$ times. Let us use the notation $\pi_\star' = \max_{\pi \in \Pi_{\mathbb{A}} \setminus \{\widehat{\pi}\}} \mathbb{E}_{a \sim \pi, x \sim \mathcal{D}} \left[ r(x, a) \right]$. The regret collected during these rounds can be upper bounded by

$$\mathcal{O}\left( \underbrace{\Delta \cdot \rho^e(t - t_e)}_{\text{Regret of } \pi_\star' \text{ w.r.t. } \widehat{\pi}} + \underbrace{\rho^e \times \mathbf{I}}_{\text{Regret of } \mathbb{A} \text{ w.r.t. } \pi_\star'} \right) = \mathcal{O}\left( \Delta \rho^e(t - t_e) + R(\Pi_{\mathbb{A}} \setminus \{\widehat{\pi}\}) \sqrt{\rho^e(t - t_e) \ln \frac{t - t_e}{\delta_e}} \right) .$$

Substituting in the value of $\rho^e$ and using the fact that $\widehat{\Delta} = \Theta(\Delta)$ when the environment is stochastic allow us to write

$$\Delta \cdot \rho^e(t - t_e) + \rho^e \cdot \mathbf{I} = \mathcal{O}\left( \frac{R^2(\Pi_{\mathbb{A}})}{k_e \Delta} (t - t_e) \ln \left( \frac{k_e}{\delta_e} \right) + \frac{R^2(\Pi_{\mathbb{A}})}{\Delta} \sqrt{\frac{t - t_e}{k_e} \ln \left( \frac{t - t_e}{\delta_e} \right) \ln \left( \frac{k_e}{\delta_e} \right)} \right)$$

$$= \mathcal{O}\left( \frac{R^2(\Pi_{\mathbb{A}})}{\Delta} \ln \left( \frac{k_e}{\delta_e} \right) \right) .$$

Summing over all epochs $e' \leq e$, and using $\sum_{e'=0}^{e} \ln \left( \frac{k_{e'}}{\delta_{e'}} \right) = \mathcal{O}\left( \ln(t) \ln \left( \frac{t}{\delta} \right) + \ln \left( \frac{R(\Pi_{\mathbb{A}})}{\Delta \delta} \right) \right)$ we conclude that

$$\mathrm{PseudoReg}_{\mathbb{M}}([t_0 + 1, t], \{\widehat{\pi}\} \cup \Pi_{\mathbb{A}}) = \mathcal{O}\left( \frac{R(\Pi_{\mathbb{A}})^2}{\Delta} \left( \ln \frac{R(\Pi_{\mathbb{A}})}{\Delta \delta} + \ln(t) \ln \frac{t}{\delta} \right) \right)$$

and thus the proof sketch of Lemma 38.

We now bound the adversarial regret of Arbe-GapExploit during the timesteps before Lines 14 or 16 of Algorithm 4 trigger. In this case, the h-stability of $\mathbb{A}$ implies,

$$\sum_{\ell=t_e+1}^{t} \mathbb{E}_{a\sim\pi_\ell^1}\left[r_\ell(a,x_\ell)\right] + \underbrace{\mathcal{O}\left(R(\Pi_{\mathbb{A}}\setminus\{\widehat{\pi}\})\sqrt{\frac{(t-t_e)}{\rho^e}\ln\frac{t-t_e}{\delta_e}}\right)}_{\mathbf{II}(t,t_e)} \geq \max_{\pi\in\Pi_{\mathbb{A}}\setminus\{\widehat{\pi}\}}\sum_{\ell=t_e+1}^{t}\mathbb{E}_{a\sim\pi}\left[r_\ell(a,x_\ell)\right].$$

(27)

While Lines 14 or 16 of Algorithm 4 have not triggered, we can certify that with high probability

$$\widehat{\Delta}(t-t_e)-V_e(t)(t-t_e) \leq \sum_{\ell=t_e+1}^{t}\mathbb{E}_{a\sim\widehat{\pi}}\left[r_\ell(a,x_\ell)\right] - \sum_{\ell=t_e+1}^{t}\mathbb{E}_{a\sim\pi_\ell^1}\left[r_\ell(a,x_\ell)\right] \leq 4\widehat{\Delta}(t-t_e)+V_e(t)(t-t_e).$$

(28)

In the above, we used the notation $V_e(t)$ to denote the $V(t)$ of epoch $e$. Hence the regret collected during rounds $\{t_e+1,\ldots,t\}$ can be upper bounded by the sum of three terms

$$\max\underbrace{\left(\sum_{\ell=t_e+1}^{t}\mathbb{E}_{a\sim\widehat{\pi}}\left[r_\ell(a,x_\ell)\right] - \sum_{\ell=t_e+1}^{t}\mathbb{E}_{a\sim\pi_\ell}\left[r_\ell(a,x_\ell)\right],0\right)}_{\mathbf{B}}$$

$$+\max\underbrace{\left(\max_{\pi\in\Pi_{\mathbb{A}}\setminus\{\widehat{\pi}\}}\sum_{\ell=t_e+1}^{t}\mathbb{E}_{a\sim\pi}\left[r_\ell(a,x_\ell)\right] - \sum_{\ell=t_e+1}^{t}\mathbb{E}_{a\sim\widehat{\pi}}\left[r_\ell(a,x_\ell)\right],0\right)}_{\mathbf{C}}$$

$$+\underbrace{\sum_{\ell=t_e+1}^{t}\mathbb{E}_{a\sim\widehat{\pi}}\left[r_\ell(a,x_\ell)\right] - r_\ell(a_\ell,x_\ell)}_{\mathbf{D}}.$$

From $t_e+1$ to $t$ the variable $b_t$ equals 1 an average of $\rho^e(t-t_e)$ times. Note that $V(t)(t-t_e)\approx \mathcal{O}\left(\widehat{\Delta}\sqrt{k_e(t-t_e)}\right)$. We can bound $\mathbf{B},\mathbf{C},\mathbf{D}$ individually as follows:

- $\mathbf{B}$ is the Pseudo-Regret of $\mathbb{A}$ w.r.t $\widehat{\pi}$, and can be upper bounded by $\mathcal{O}\left(\widehat{\Delta}\cdot\rho^e(t-t_e)+\widehat{\Delta}\cdot\rho^e\sqrt{(t-t_e)k_e}\right) = \mathcal{O}\left(\widehat{\Delta}\cdot\rho^e\sqrt{(t-t_e)k_e}\right)$ as a consequence of multiplying the right hand side of Equation 28 by $\rho^e$.

- $\mathbf{C}$ is the Pseudo-Regret of $\widehat{\pi}$ w.r.t. $\Pi_{\mathbb{A}}\setminus\{\widehat{\pi}\}$, and can be upper bounded by $\mathbf{II}(t,t_e)-\widehat{\Delta}(t-t_e)+V_e(t)(t-t_e)$ as a consequence of combining Equations 27 and the left hand side of 28.

- $\mathbf{D}$ is the difference between sample rewards vs. Pseudo-Rewards and can be bound by Hoeffding's inequality.

Let $t_e'=\min(t,t_e)$. Summing the upper bound $\mathbf{C}$ over all epochs $e'\leq e$, we choose the multiplier $c$ in $\rho^e=\frac{cR^2(\Pi_{\mathbb{A}})}{k_e\widehat{\Delta}^2}\ln\frac{k_e}{\delta_e}$ such that[9]

$$\sum_{e'\leq e}\mathbf{II}(t_{e'}',t_{e'})-\widehat{\Delta}(t_{e'}'-t_{e'})+V_{e'}(t_{e'}')(t_{e'}'-t_{e'}) = \sum_{e'\leq e}\mathcal{O}\left(\widehat{\Delta}\sqrt{\frac{k_e(t_{e'}'-t_{e'})}{c}}\right)-\widehat{\Delta}(t_{e'}'-t_{e'})$$

$$\leq \mathcal{O}\left(\widehat{\Delta}k_0\right).$$

(29)

Combining the bounds for $\mathbf{B}$ and $\mathbf{D}$,

$$\mathcal{O}\left(\underbrace{\widehat{\Delta}\cdot\rho^e(t-t_e)}_{\mathbf{B}}+\underbrace{\sqrt{(t-t_e)\ln\left(\frac{t-t_e}{\delta_e}\right)}}_{\mathbf{D}}\right) = \mathcal{O}\left(\frac{R^2(\Pi_{\mathbb{A}})}{\widehat{\Delta}}\ln\left(\frac{k_e}{\delta_e}\right)+\sqrt{(t-t_e)\ln\left(\frac{t-t_e}{\delta_e}\right)}\right).$$

[9]Increasing the value of $c$ implies we have to set $k_0$ to be larger. This has the only effect of increasing the constant on the RHS of Eq. 29.

Summing over all epochs and using the upper bounds

$$\sum_{e'=0}^{e} \ln\left(\frac{k_{e'}}{\delta_{e'}}\right) = \mathcal{O}\left(\ln(t)\ln\left(\frac{t}{\delta}\right) + \ln\left(\frac{R(\Pi_{\mathbb{A}})}{\widehat{\Delta}\delta}\right)\right)$$

$$\sum_{e'=0}^{e} \sqrt{(t'_{e'} - t_{e'})\ln\left(\frac{t'_{e'} - t_{e'}}{\delta_{e'}}\right)} = \mathcal{O}\left(\sqrt{(t - t_0)\ln(t)\ln\frac{t}{\delta}}\right)$$

and the bound from Eq. 29 along with the observation $\widehat{\Delta}k_0 = \mathcal{O}\left(\frac{R^2(\Pi_{\mathbb{A}})}{\widehat{\Delta}}\ln\frac{R(\Pi_{\mathbb{A}})}{\widehat{\Delta}\delta}\right)$ allows us to conclude,

$$\text{Reg}_{\mathbb{M}}([t_0 + 1, t], \Pi_{\mathbb{A}} \cup \{\widehat{\pi}\}) = \mathcal{O}\left(\frac{R(\Pi_{\mathbb{A}})^2}{\widehat{\Delta}}\left(\ln(t)\ln\frac{t}{\delta} + \ln\frac{R(\Pi_{\mathbb{A}})}{\widehat{\Delta}\delta}\right) + \sqrt{(t - t_0)\ln(t)\ln\frac{t}{\delta}}\right),$$

and with it we finish the proof sketch of Lemma 41. Combining these results finalizes the proof of Lemma 25.

## D.2 Analysis of the First Phase

We will use the notation $t_{\text{gap}}$ to denote the (random) time when the **Arbe-Gap** estimation phase ends (see Algorithm 2).

**Lemma 27** (Guarantee for First Phase). *Consider a run of Algorithm 2 with inputs $t_0 = 0$, arbitrary policy policy $\widehat{\pi} \in \Pi_M$ and $M$ base learners $\mathbb{A}_1, \ldots, \mathbb{A}_M$. Further, let $t_{\text{gap}} \in \mathbb{N} \cup \{\infty\}$ be the round where the **Arbe-GapExploit** subroutine is called. Then with probability at least $1 - \text{poly}(M)\delta$, the following conditions hold for all rounds $t \in [2i_\star, t_{\text{gap}}]$. The regret is bounded as*

$$\text{Reg}_{\mathbb{M}}(t, \Pi_M) = \mathcal{O}\left(\left(M + \frac{R(\widetilde{\Pi}_{i_\star})}{R(\widetilde{\Pi}_1)}\sqrt{i_\star}\right)R(\widetilde{\Pi}_{i_\star})\sqrt{t(\ln(t) + i_\star)\ln\frac{t}{\delta}}\right).$$

*If $\mathbb{B}$ is stochastic and there is a unique policy $\pi_\star$ with gap $\Delta > 0$, then the gap estimator $\widehat{\Delta}$ and policy $\widehat{\pi}$ passed onto **Arbe-GapExploit** satisfy $\frac{\Delta}{2} \leq \widehat{\Delta} \leq \Delta$ and $\widehat{\pi} = \pi_\star$. Further, the pseudo-regret is bounded as*

$$\text{PseudoReg}_{\mathbb{M}}(t, \Pi_M) = \mathcal{O}\left(\frac{R(\widetilde{\Pi}_{i_\star})^2 R(\widetilde{\Pi}_M)^2}{R(\widetilde{\Pi}_1)^2}\frac{M^2 i_\star}{\Delta}\ln\left(\frac{MR(\widetilde{\Pi}_M)}{\Delta\delta}\right)\ln\frac{t}{\delta}\right).$$

### D.2.1 Adversarial Guarantees

We first start by bounding the number of restarts of the algorithm:

**Lemma 28** (Number of Restarts of **Arbe-Gap**). *Consider a run of Algorithm 2 with inputs $t_0 = 0$, arbitrary policy policy $\widehat{\pi} \in \Pi_M$ and $M$ base learners $\mathbb{A}_1, \ldots, \mathbb{A}_M$. Then for any total number of rounds $t$, there are at most $\ln(t)$ restarts due to a change in candidate policy (Line 17 in Algorithm 2) up to that round $t$. Further, with probability at least $1 - \text{poly}(M)\delta$, there are at most $i_\star - 1$ restarts due an elimination of a base learner (Line 11 in Algorithm 2).*

*Proof.* We first show the bound on the number of restarts due to changes in the candidate policy $\widehat{\pi}$. Let $t_1, t_2, \ldots$ be the rounds at which a restart is triggered in Line 17 of Algorithm 2 and $\widehat{\pi}_1, \widehat{\pi}_2, \ldots$ be the candidate policies selected at those restarts. For each restart $i$, we know that $\widehat{\pi}_i$ was selected in at least $\frac{3t_i}{4}$ of the first $t_i$ rounds and therefore also $t_{i+1}$ rounds. However, since the policy changed from $\widehat{\pi}_i$ to $\widehat{\pi}_{i+1}$ at round $t_{i+1}$, we also know that $\widehat{\pi}_i$ can only be selected at most $\frac{t_{i+1}}{4}$ of the first $t_{i+1}$ rounds. Combining both bounds yields

$$\frac{3t_i}{4} \leq \frac{t_{i+1}}{4}$$

and thus $t_{i+1} \geq 3t_i$ holds for all $i$. Since also $t_1 \geq 9$ by the condition in the algorithm, up to round $t$, there can only be $\log_3(t) - 1 \leq \ln(t) - 1 \leq \ln(t)$ restarts.

Finally, the number of restarts due to base learner elimination is bounded by $i_\star - 1$ since this condition can never trigger for $i \geq i_\star$ by Lemma 22 (which also holds for **Arbe-Gap**). $\square$

The following lemma now bounds the regret within each restart:

**Lemma 29.** *Consider a run of Algorithm 2 with Arbe-Gap$(\delta, s, t_0, \widehat{\pi})$ where $s \leq i_\star$ and let $T \in \mathbb{N} \cup \{\infty\}$ be the round when the algorithm restarts or calls Arbe-GapExploit ($T = \infty$ if there is no restart or transition to the second phase). Then the regret against $\Pi_M$ is bounded with probability at least $1 - poly(M)\delta$ for all $t \in [t_0 + 1, T] = \{t_0 + 1, t_0 + 2, \ldots, T\}$ simultaneously as*

$$\mathrm{Reg}_{\mathbb{M}}([t_0 + 1, t], \Pi_M) = \mathcal{O}\left(\left(MR(\widetilde{\Pi}_{i_\star}) + \frac{R(\widetilde{\Pi}_{i_\star})^2}{R(\widetilde{\Pi}_1)}\sqrt{i_\star}\right)\sqrt{(t - t_0)\ln\frac{t}{\delta}} + M\ln\frac{\ln t}{\delta}\right).$$

*Proof.* The regret of Arbe-Gap in rounds $[t_0 + 1, t]$ against any policy $\pi' \in \Pi_M$ can be written as

$$\sum_{\ell=t_0+1}^{t} [\mathbb{E}_{a\sim\pi'}[r_\ell(a, x_\ell)] - r_\ell(a_\ell, x_\ell)] = \sum_{i=s}^{M+1}\sum_{\ell=t_0+1}^{t} [\rho_i\mathbb{E}_{a\sim\pi'}[r_\ell(a, x_\ell)] - \mathbf{1}\{b_\ell = i\}\, r_\ell(a_\ell, x_\ell)] .$$

For $i \in [s, M]$, we can follow the analysis of Arbe and apply the arguments in the proof of Lemma 21 verbatim. This yields with probability at least $1 - poly(M)\delta$

$$\sum_{i=s}^{M}\sum_{\ell=t_0+1}^{t} [\rho_i\mathbb{E}_{a\sim\pi'}[r_\ell(a, x_\ell)] - \mathbf{1}\{b_\ell = i\}\, r_\ell(a_\ell, x_\ell)]$$

$$= \mathcal{O}\left(\left(MR(\widetilde{\Pi}_{i_\star}) + \frac{R(\widetilde{\Pi}_{i_\star})^2}{R(\widetilde{\Pi}_1)}\sqrt{i_\star}\right)\sqrt{(t - t_0)\ln\frac{t}{\delta}} + M\ln\frac{\ln t}{\delta}\right). \qquad (30)$$

It only remains to bound the regret contribution of the special base learner $\mathbb{A}_{M+1}$, which does not exist in Arbe. To do so, we will use the fact that the gap test in Line 14 can only trigger in the last round before a restart happens. This allows us to relate the regret of $\mathbb{A}_{M+1}$ to that of $\mathbb{A}_M$. For the regret of $\mathbb{A}_M$, we again use the arguments in the proof of Lemma 21 (Equation 23 specifically) verbatim to show

$$\sum_{\ell=t_0+1}^{t} [\rho_M\mathbb{E}_{a\sim\pi'}[r_\ell(a, x_\ell)] - \mathbf{1}\{b_\ell = M\}\, r_\ell(a_\ell, x_\ell)] = \mathcal{O}\left(R(\widetilde{\Pi}_{i_\star})\sqrt{(t - t_0)\ln\frac{t}{\delta}} + \ln\frac{\ln t}{\delta}\right). \tag{31}$$

The regret contribution of $\mathbb{A}_{M+1}$ can now be bounded as

$$\sum_{\ell=t_0+1}^{t} [\rho_{M+1}\mathbb{E}_{a\sim\pi'}[r_\ell(a, x_\ell)] - \mathbf{1}\{b_\ell = M + 1\}\, r_\ell(a_\ell, x_\ell)]$$

$$= \rho_{M+1}\left(\left[\sum_{\ell=t_0+1}^{t}\mathbb{E}_{a\sim\pi'}[r_\ell(a, x_\ell)]\right] - \widetilde{\mathrm{CRew}}_{M+1}(t_0 + 1, t)\right) \qquad \text{(definition of } \widetilde{\mathrm{CRew}}_{M+1})$$

$$= \rho_{M+1}\left(\left[\sum_{\ell=t_0+1}^{t}\mathbb{E}_{a\sim\pi'}[r_\ell(a, x_\ell)]\right] - \widetilde{\mathrm{CRew}}_M(t_0 + 1, t)\right)$$

$$\quad + \rho_{M+1}\left(\widetilde{\mathrm{CRew}}_M(t_0 + 1, t) - \widetilde{\mathrm{CRew}}_{M+1}(t_0 + 1, t)\right)$$

$$\leq \rho_{M+1}\left(\left[\sum_{\ell=t_0+1}^{t}\mathbb{E}_{a\sim\pi'}[r_\ell(a, x_\ell)]\right] - \widetilde{\mathrm{CRew}}_M(t_0 + 1, t)\right)$$

$$\quad + 7\rho_{M+1}W(t_0, t)(t - t_0) + 1 \qquad \text{(gap test not triggered at } t - 1)$$

$$\leq \frac{\rho_{M+1}}{\rho_M}\mathcal{O}\left(R(\widetilde{\Pi}_{i_\star})\sqrt{(t - t_0)\ln\frac{t}{\delta}} + \ln\frac{\ln t}{\delta}\right) + 7\rho_{M+1}W(t_0, t)(t - t_0) + 1 \qquad \text{(Equation 31)}$$

$$\leq \mathcal{O}\left(R(\widetilde{\Pi}_{i_\star})\sqrt{(t - t_0)\ln\frac{t}{\delta}} + \ln\frac{\ln t}{\delta}\right) + 7\rho_M W(t_0, t)(t - t_0) \qquad (\rho_M \geq \rho_{M+1})$$

$$= \mathcal{O}\left(R(\widetilde{\Pi}_{i_\star})\sqrt{(t-t_0)\ln\frac{t}{\delta}} + \ln\frac{\ln t}{\delta}\right) + \mathcal{O}\left(R(\widetilde{\Pi}_M)\sqrt{\rho_M(t-t_0)\ln\frac{t}{\delta}}\right) \quad \text{(definition of } W\text{)}$$

$$= \mathcal{O}\left(R(\widetilde{\Pi}_{i_\star})\sqrt{(t-t_0)\ln\frac{t}{\delta}} + \ln\frac{\ln t}{\delta} + \sqrt{\frac{t-t_0}{\sum_{i=s}^{M+1}R(\widetilde{\Pi}_i)^{-2}}\ln\frac{t}{\delta}}\right) \quad \text{(definition of } \rho_M\text{)}$$

$$= \mathcal{O}\left(R(\widetilde{\Pi}_{i_\star})\sqrt{(t-t_0)\ln\frac{t}{\delta}} + \ln\frac{\ln t}{\delta}\right) \quad (s \le i_\star)$$

Note that $\rho_M \ge \rho_{M+1}$ holds without loss of generality since $\widetilde{\Pi}_{M+1}$ contains 2 fewer policies than $\widetilde{\Pi}_M$. Finally, the desired statement follows by combining the previous display with Equation 30. $\quad\square$

Equipped with the previous two lemmas, we can now prove the regret bound of the first phase for adversarial environments:

**Lemma 30.** *Consider a run of Algorithm 2 with inputs $t_0 = 0$, arbitrary policy policy $\widehat{\pi} \in \Pi_M$ and $M$ base learners $\mathbb{A}_1,\ldots,\mathbb{A}_M$. Further, let $t_{\mathrm{gap}} \in \mathbb{N}\cup\{\infty\}$ be the round where the **Arbe-GapExploit** subroutine is called. Then with probability at least $1 - poly(M)\delta$, the following conditions hold for all rounds $t \in [2i_\star, t_{\mathrm{gap}}]$. The regret is bounded as*

$$\mathrm{Reg}_{\mathbb{M}}(t, \Pi_M) = \mathcal{O}\left(\left(MR(\widetilde{\Pi}_{i_\star}) + \frac{R(\widetilde{\Pi}_{i_\star})^2}{R(\widetilde{\Pi}_1)}\sqrt{i_\star}\right)\sqrt{t(\ln(t) + i_\star)\ln\frac{t}{\delta}}\right).$$

*Proof.* Let $\tau_0, \tau_1, \ldots$ be the rounds where **Arbe-Gap** restarts or eventually calls **Arbe-GapExploit**. By convention, we set $\tau_0 = 0$ and $\tau_i = \infty$ when there are less than $i$ total calls to **Arbe-Gap**. By Lemma 28, there are at most $\ln(t) + i_\star$ calls of **Arbe-Gap** up to round $t$ with probability at least $1 - poly(M)\delta$ for all $t \in \mathbb{N}$ jointly. Further, by Lemma 29, the regret in each of these calls is bounded with probability $1 - poly(M)\delta$ as well. If we were to apply a naive union bound over all $\ln(t) + i_\star$, then our failure probability would increase at a rate of $\ln(t)$. However, we can easily avoid this by choosing the absolute constants in the definition of $\mathrm{D}_i$ appropriately. A factor of 3 larger is sufficient. This ensures that each of these terms is effectively at least as large as if we had invoked them with $\frac{\delta}{n^2}$ instead of $\delta$ in the $n$-th restart of **Arbe-Gap**. We now illustrate why this is true. Let $c'$ be the absolute constant such that

$$\mathrm{D}_i(t_0, t) = c'\sqrt{\frac{t-t_0}{\rho_i}\ln\frac{\ln t}{\delta}} + \frac{c'}{\rho_i}\ln\frac{\ln t}{\delta}$$

satisfies the necessary concentration bounds (see Section C.2) for a single restart of **Arbe-Gap**. Now, we have

$$3c'\sqrt{\frac{t-t_0}{\rho_i}\ln\frac{\ln t}{\delta}} + \frac{3c'}{\rho_i}\ln\frac{\ln t}{\delta} \ge c'\sqrt{\frac{t-t_0}{\rho_i}\ln\frac{(\ln t)^9}{\delta^9}} + \frac{c'}{\rho_i}\ln\frac{(\ln t)^3}{\delta^3}$$

$$\ge c'\sqrt{\frac{t-t_0}{\rho_i}\ln\frac{(\ln t)^3}{\delta}} + \frac{c'}{\rho_i}\ln\frac{(\ln t)^3}{\delta}$$

$$\ge c'\sqrt{\frac{t-t_0}{\rho_i}\ln\frac{\ln t}{\frac{\delta}{\ln^2 t}}} + \frac{c'}{\rho_i}\ln\frac{\ln t}{\frac{\delta}{\ln^2 t}}$$

$$\ge c'\sqrt{\frac{t-t_0}{\rho_i}\ln\frac{\ln t}{\frac{\delta}{n^2}}} + \frac{c'}{\rho_i}\ln\frac{\ln t}{\frac{\delta}{n^2}}$$

where the last step holds because the number of calls $n$ to **Arbe-Gap** due to a change in candidate policy at round $t$ is bounded as $\ln(t)$. Hence, with this choice of constant, we can ensure that the statement of Lemma 29 holds with probability at least $1 - poly(M)\delta$ jointly for all calls of **Arbe-Gap** (for the remaining $i_\star \le M$ restarts possible due to elimination of a base learner, we apply a standard union bound).

Now, just as in the proof of Theorem 4, we write the regret of Arbe-Gap using $\bar{t}_s = \max\{\min\{\tau_s, t\} - \tau_{s-1}, 0\}$ as

$$\text{Reg}_{\mathbb{M}}(t, \Pi_M)$$

$$= \sum_{i=1}^{\ln(t)+i_\star} \text{Reg}([\tau_{i-1}+1, \min\{\tau_i, t\}], \Pi_M)$$

$$= \sum_{i=1}^{\ln(t)+i_\star} \mathcal{O}\left(\left(MR(\widetilde{\Pi}_{i_\star}) + \frac{R(\widetilde{\Pi}_{i_\star})^2}{R(\widetilde{\Pi}_1)}\sqrt{i_\star}\right)\sqrt{\bar{t}_i \ln \frac{t}{\delta}} + M\ln\frac{\ln t}{\delta}\right) \qquad \text{(Lemma 29)}$$

$$= \mathcal{O}\left(\left(MR(\widetilde{\Pi}_{i_\star}) + \frac{R(\widetilde{\Pi}_{i_\star})^2}{R(\widetilde{\Pi}_1)}\sqrt{i_\star}\right)\sqrt{(\ln(t)+i_\star)\sum_{i=1}^{\ln(t)+i_\star}\bar{t}_i \ln\frac{t}{\delta}}\right)$$

$$\quad + \mathcal{O}\left(M(\ln(t)+i_\star)\ln\frac{\ln t}{\delta}\right) \qquad \text{(Jensen's inequality)}$$

$$= \mathcal{O}\left(\left(MR(\widetilde{\Pi}_{i_\star}) + \frac{R(\widetilde{\Pi}_{i_\star})^2}{R(\widetilde{\Pi}_1)}\sqrt{i_\star}\right)\sqrt{t(\ln(t)+i_\star)\ln\frac{t}{\delta}}\right)$$

$$\quad + \mathcal{O}\left(M(\ln(t)+i_\star)\ln\frac{\ln t}{\delta}\right) \qquad (\sum_s \bar{t}_s \le t)$$

$$= \mathcal{O}\left(\left(MR(\widetilde{\Pi}_{i_\star}) + \frac{R(\widetilde{\Pi}_{i_\star})^2}{R(\widetilde{\Pi}_1)}\sqrt{i_\star}\right)\sqrt{t(\ln(t)+i_\star)\ln\frac{t}{\delta}}\right) \qquad (t \ge 2i_\star \text{ by assumption})$$

This concludes the proof. $\qquad\qquad\square$

### D.2.2 Stochastic Guarantees

As a first step, we show that Arbe-Gap always maintains valid confidence bounds on the gap of the candidate policy in a stochastic environment:

**Lemma 31** (Confidence bounds on the gap). *Consider a run of Algorithm 2 with inputs $n = 1, t_0 = 0$, arbitrary policy policy $\widehat{\pi} \in \Pi_M$ and $M$ base learners $\mathbb{A}_1, \ldots, \mathbb{A}_M$ with $1 \le R(\widetilde{\Pi}_1) \le R(\widetilde{\Pi}_2) \le \cdots \le R(\widetilde{\Pi}_M)$ where $\widetilde{\Pi}_i$ is the extended version of policy class $\Pi_i$ with $(M + 1 - i)$ additional actions. Assume the environment $\mathbb{B}$ is stochastic and there is a policy $\pi_\star$ with gap $\Delta > 0$. Then with probability at least $1 - poly(M)\delta$ in all rounds $t \in \mathbb{N}$*

$$\widehat{\Delta}_t \le \Delta_{\widehat{\pi}} \le \widehat{\Delta}_t + 2W(t_0, t)$$

*where $W(t_0, t)$ is the term used in the definition of $\widehat{\Delta}_t$ in the algorithm with*

$$W(t_0, t) = \Theta\left(\frac{R(\widetilde{\Pi}_M)}{\sqrt{\rho_M}}\sqrt{\frac{\ln\frac{n(t-t_0)}{\delta}}{t-t_0}} + \frac{1}{\rho_M}\frac{\ln\frac{n\ln(t-t_0)}{\delta}}{t-t_0}\right),$$

*and $t_0$ and $n$ are the time and number of the last restart and before $t$ and $\Delta_{\widehat{\pi}} = \mathbf{1}\{\widehat{\pi} = \pi_\star\}\Delta$ is the gap of the candidate policy $\widehat{\pi}$ in round $t$.*

*Proof.* First, consider a single restart of Arbe-Gap. We note that both base learner $\mathbb{A}_M$ and $\mathbb{A}_{M+1}$ are h-stable on their respective policy classes $\widetilde{\Pi}_M$ and $\widetilde{\Pi}_{M+1}$ by assumption (removing a single policy usually does not impede $h$-stability, see e.g. Appendix B). Further, denote by $\pi_\star^i = \text{argmin}_{\pi \in \widetilde{\Pi}_i} \mathbb{E}_{a \sim \pi, x \sim \mathcal{D}}[r(a, x)]$ a best policy in policy class $\widetilde{\Pi}_i$. We can use these insights and definition to derive the following lower-bound on $\widetilde{\text{CRew}}_i(t_0, t)$ for $i \in \{M, M + 1\}$ that holds

uniformly with probability at least $1 - \text{poly}(M)\delta$

$$\widetilde{\text{CRew}}_i(t_0, t)$$

$$\geq \sum_{\ell=t_0+1}^{t} r_\ell(a_\ell^i, x_\ell) - \mathcal{O}\left(\sqrt{\frac{t-t_0}{\rho_i} \ln \frac{\ln(t-t_0)}{\delta}} + \frac{1}{\rho_i} \ln \frac{\ln(t-t_0)}{\delta}\right) \qquad \text{(Lemma 23)}$$

$$\geq \max_{\pi' \in \widetilde{\Pi}_i} \sum_{\ell=t_0+1}^{t} \mathbb{E}_{a \sim \pi'}[r_\ell(a, x_\ell)] - \mathcal{O}\left(R(\widetilde{\Pi}_i)\sqrt{\frac{t-t_0}{\rho_i} \ln \frac{t-t_0}{\delta}} + \frac{1}{\rho_i} \ln \frac{\ln(t-t_0)}{\delta}\right)$$

$$\text{(h-stability of } \mathbb{A}_i)$$

$$\geq \sum_{\ell=t_0+1}^{t} \mathbb{E}_{a \sim \pi_\star^i}[r_\ell(a, x_\ell)] - \mathcal{O}\left(R(\widetilde{\Pi}_i)\sqrt{\frac{t-t_0}{\rho_i} \ln \frac{t-t_0}{\delta}} + \frac{1}{\rho_i} \ln \frac{\ln(t-t_0)}{\delta}\right) \qquad (\pi_\star^i \in \widetilde{\Pi}_i)$$

$$\geq \sum_{\ell=t_0+1}^{t} \mathbb{E}_{a \sim \pi_\star^i, x \sim \mathcal{D}}[r(a, x)] - \mathcal{O}\left(R(\widetilde{\Pi}_i)\sqrt{\frac{t-t_0}{\rho_i} \ln \frac{t-t_0}{\delta}} + \frac{1}{\rho_i} \ln \frac{\ln(t-t_0)}{\delta}\right)$$

$$- \mathcal{O}\left(\sqrt{(t-t_0) \ln \frac{\ln t}{\delta}}\right) \qquad \text{(Lemma 43)}$$

$$= \sum_{\ell=t_0+1}^{t} \mathbb{E}_{a \sim \pi_\star^i, x \sim \mathcal{D}}[r(a, x)] - \mathcal{O}\left(R(\widetilde{\Pi}_i)\sqrt{\frac{t-t_0}{\rho_i} \ln \frac{t-t_0}{\delta}} + \frac{1}{\rho_i} \ln \frac{\ln(t-t_0)}{\delta}\right).$$

Conversely, using similar concentration arguments, we can derive the following upper-bound for $\widetilde{\text{CRew}}_i(t_0, t)$ for $i \in \{M, M+1\}$ that holds uniformly with probability at least $1 - \text{poly}(M)\delta$:

$$\widetilde{\text{CRew}}_i(t_0, t)$$

$$\leq \sum_{\ell=t_0+1}^{t} r_\ell(a_\ell^i, x_\ell) + \mathcal{O}\left(\sqrt{\frac{t-t_0}{\rho_i} \ln \frac{\ln(t-t_0)}{\delta}} + \frac{1}{\rho_i} \ln \frac{\ln(t-t_0)}{\delta}\right) \qquad \text{(Lemma 23)}$$

$$\leq \sum_{\ell=t_0+1}^{t} \mathbb{E}_{a \sim \pi_\ell^i}[r_\ell(a, x_\ell)] + \mathcal{O}\left(\sqrt{\frac{t-t_0}{\rho_i} \ln \frac{\ln(t-t_0)}{\delta}} + \frac{1}{\rho_i} \ln \frac{\ln(t-t_0)}{\delta}\right) \qquad \text{(Lemma 23)}$$

$$\leq \sum_{\ell=t_0+1}^{t} \mathbb{E}_{a \sim \pi_\ell^i, x \sim \mathcal{D}}[r(a, x)] + \mathcal{O}\left(\sqrt{\frac{t-t_0}{\rho_i} \ln \frac{\ln(t-t_0)}{\delta}} + \frac{1}{\rho_i} \ln \frac{\ln(t-t_0)}{\delta}\right) \qquad \text{(Lemma 43)}$$

$$\leq \sum_{\ell=t_0+1}^{t} \mathbb{E}_{a \sim \pi_\star^i, x \sim \mathcal{D}}[r(a, x)] + \mathcal{O}\left(\sqrt{\frac{t-t_0}{\rho_i} \ln \frac{\ln(t-t_0)}{\delta}} + \frac{1}{\rho_i} \ln \frac{\ln(t-t_0)}{\delta}\right) \qquad \text{(def. of } \pi_\star^i)$$

Note that $\Delta_{\widehat{\pi}} = \mathbb{E}_{a \sim \pi_\star^M, x \sim \mathcal{D}}[r(a, x)] - \mathbb{E}_{a \sim \pi_\star^{M+1}, x \sim \mathcal{D}}[r(a, x)]$ which is either $\Delta$ if $\widehat{\pi} = \pi_\star$ or $0$ if $\widehat{\pi} \neq \pi^\star$. Combining the bounds on $\widetilde{\text{CRew}}_i$ with a union bound, we can derive the following deviation bound

$$- \mathcal{O}\left(\left(\frac{R(\widetilde{\Pi}_M)}{\sqrt{\rho_M}} + \frac{1}{\sqrt{\rho_{M+1}}}\right)\sqrt{\frac{\ln\frac{t-t_0}{\delta}}{t-t_0}} + \left(\frac{1}{\rho_M} + \frac{1}{\rho_{M+1}}\right)\frac{\ln\frac{\ln(t-t_0)}{\delta}}{t-t_0}\right)$$

$$\leq \frac{\widetilde{\text{CRew}}_M(t_0, t) - \widetilde{\text{CRew}}_{M+1}(t_0, t)}{t-t_0} - \Delta_{\widehat{\pi}} \leq$$

$$+ \mathcal{O}\left(\left(\frac{R(\widetilde{\Pi}_{M+1})}{\sqrt{\rho_{M+1}}} + \frac{1}{\sqrt{\rho_M}}\right)\sqrt{\frac{\ln\frac{t-t_0}{\delta}}{t-t_0}} + \left(\frac{1}{\rho_M} + \frac{1}{\rho_{M+1}}\right)\frac{\ln\frac{\ln(t-t_0)}{\delta}}{t-t_0}\right).$$

We can further simplify those bounds by noting that $R(\widetilde{\Pi}_{M+1}) \leq R(\widetilde{\Pi}_M)$ and thus also $\rho_{M+1} \geq \rho_M$ since $\widetilde{\Pi}_M$ is identical to $\widetilde{\Pi}_{M+1}$ but contains two more policies. Thus, we can bound the magnitude

of the upper and lower bound further by

$$\mathcal{O}\left(\frac{R(\widetilde{\Pi}_M)}{\sqrt{\rho_M}}\sqrt{\frac{\ln\frac{t-t_0}{\delta}}{t-t_0}} + \frac{1}{\rho_M}\frac{\ln\frac{\ln(t-t_0)}{\delta}}{t-t_0}\right).$$

We now rebind $\delta$ by $\frac{\delta}{n^2}$ and apply a union bound over all restarts of Arbe-Gap. Thus, we can choose a constant in the definition of

$$\mathbf{W}(t_0, t) = \Theta\left(\frac{R(\widetilde{\Pi}_M)}{\sqrt{\rho_M}}\sqrt{\frac{\ln\frac{n(t-t_0)}{\delta}}{t-t_0}} + \frac{1}{\rho_M}\frac{\ln\frac{n\ln(t-t_0)}{\delta}}{t-t_0}\right)$$

large enough so that

$$-\mathbf{W}(t_0, t) \le \frac{\widetilde{\mathrm{CRew}}_M(t_0, t) - \widetilde{\mathrm{CRew}}_{M+1}(t_0, t)}{t-t_0} - \Delta_{\widehat{\pi}} \le \mathbf{W}(t_0, t)$$

holds for all $t$ in all possible restarts of Arbe-Gap with the desired $1 - \mathrm{poly}(M)\delta$ probability. $\qquad\square$

The lemma above immediately implies the correctness of the first phase, in the sense that if the algorithm moves on to the second phase in a stochastic environment, the candidate policy has to be optimal and the gap estimate is accurate up to a multiplicative factor:

**Corollary 32.** *Consider a run of Algorithm 2 with inputs $n = 1, t_0 = 0$, arbitrary policy policy $\widehat{\pi} \in \Pi_M$ and $M$ base learners $\mathbb{A}_1, \ldots, \mathbb{A}_M$ with $1 \le R(\widetilde{\Pi}_1) \le R(\widetilde{\Pi}_2) \le \cdots \le R(\widetilde{\Pi}_M)$ where $\widetilde{\Pi}_i$ is the extended version of policy class $\Pi_i$ with $(M + 1 - i)$ additional actions. Assume the environment $\mathbb{B}$ is stochastic and there is a policy $\pi_\star$ with gap $\Delta > 0$. Then with probability at least $1 - \mathrm{poly}(M)\delta$ the policy $\widehat{\pi}$ and gap estimate $\widehat{\Delta}$ passed to Arbe-GapExploit satisfy*

$$\widehat{\pi} = \pi_\star \qquad and \qquad \widehat{\Delta} \le \Delta \le 2\widehat{\Delta}.$$

*Proof.* The statement follows from Lemma 31 and the condition in Line 14 of Algorithm 2. First, note that the test cannot trigger when $\widehat{\pi} \ne \pi_\star$ since $\widehat{\Delta}_t \le 0$ in this case. Second, since $\widehat{\Delta}$ satisfies $2W(t_0, t) \le \widehat{\Delta}$ and $\widehat{\Delta} \le \Delta \le \widehat{\Delta} + 2W(t_0, t)$ when the test triggers, we have

$$\widehat{\Delta} \le \Delta \le \widehat{\Delta} + 2W(t_0, t) \le 2\widehat{\Delta},$$

as claimed. $\qquad\square$

We now move on to show that if there is a policy with a gap, the alorithm has to identify it within a certain number of rounds:

**Lemma 33** (Arbe-Gap selects the right candidate policy). *Consider a run of Algorithm 2 with inputs $t_0 = 0$, arbitrary policy policy $\widehat{\pi} \in \Pi_M$ and $M$ base learners $\mathbb{A}_1, \ldots, \mathbb{A}_M$ with $1 \le R(\widetilde{\Pi}_1) \le R(\widetilde{\Pi}_2) \le \cdots \le R(\widetilde{\Pi}_M)$ where $\widetilde{\Pi}_i$ is the extended version of policy class $\Pi_i$ with $(M + 1 - i)$ additional actions. Assume the environment $\mathbb{B}$ is stochastic and there is a policy $\pi_\star$ with gap $\Delta > 0$. Then with probability at least $1 - \mathrm{poly}(M)\delta$ the number of rounds until $\pi_\star$ is always chosen as the candidate policy $\widehat{\pi}$ is bounded as*

$$\mathcal{O}\left(\left(M^2 + \frac{R(\widetilde{\Pi}_{i_\star})^2}{R(\widetilde{\Pi}_1)^2}i_\star\right)\frac{R(\widetilde{\Pi}_{i_\star})^2 i_\star}{\Delta^2}\ln^2\frac{MR(\widetilde{\Pi}_{i_\star})}{\Delta\delta}\right).$$

*Proof.* By Lemma 30, with probability at least $1 - \mathrm{poly}(M)\delta$ the regret of Arbe-Gap is bounded for all rounds $t \in [2i_\star, t_{\mathrm{gap}}]$ as

$$\mathrm{Reg}_{\mathbb{M}}(t, \Pi_M) = \mathcal{O}\left(\left(MR(\widetilde{\Pi}_{i_\star}) + \frac{R(\widetilde{\Pi}_{i_\star})^2}{R(\widetilde{\Pi}_1)}\sqrt{i_\star}\right)\sqrt{t(\ln(t) + i_\star)\ln\frac{t}{\delta}}\right)$$

$$= \mathcal{O}\left(\left(MR(\widetilde{\Pi}_{i_\star})\sqrt{i_\star} + \frac{R(\widetilde{\Pi}_{i_\star})^2}{R(\widetilde{\Pi}_1)}i_\star\right)\sqrt{t}\ln\frac{t}{\delta}\right).$$

By the concentration argument in Lemma 42, the same bound can be established for the pseudo-regret

$$\text{PseudoReg}_{\mathbb{M}}(t, \Pi_M) \leq c \left( M R(\widetilde{\Pi}_{i_\star}) \sqrt{i_\star} + \frac{R(\widetilde{\Pi}_{i_\star})^2}{R(\widetilde{\Pi}_1)} i_\star \right) \sqrt{t} \ln \frac{t}{\delta} \tag{32}$$

for some sufficiently large absolute constant $c$. Now denote $\gamma = c \left( M R(\widetilde{\Pi}_{i_\star}) \sqrt{i_\star} + \frac{R(\widetilde{\Pi}_{i_\star})^2}{R(\widetilde{\Pi}_1)} i_\star \right)$ and consider the value

$$t' = \frac{16^2 \gamma^2}{\Delta^2} \ln^2 \frac{8\gamma}{\Delta\delta} \ .$$

Then by the properties of $\frac{\ln(t)}{\sqrt{t}}$ investigated in Lemma 52, we can bound for $t \geq t'$

$$\begin{aligned}
\text{PseudoReg}_{\mathbb{M}}(t, \Pi_M) &\leq \gamma \sqrt{t} \ln \frac{t}{\delta} && \text{(Equation 32)} \\
&= \gamma t \frac{\ln(t/\delta)}{\sqrt{t}} \leq \gamma t \frac{\Delta}{4\gamma} && \text{(Lemma 52)} \\
&= \frac{t}{4} \Delta \ .
\end{aligned}$$

We have shown that the adversarial regret rate implies that the pseudo-regret for rounds $t \geq t'$ has to be bounded by $\frac{t}{4}\Delta$. Since each policy but $\pi^\star$ incurs a pseudo-regret at least $\Delta$ per round, Arbe-Gap has to select $\pi^\star$ in at least $\frac{3}{4}t$ among all $t$ rounds to satisfy this pseudo-regret bound. As a result, a switch of the candidate policy to $\pi^\star$ would be triggered if it is not already the candidate policy. Further, no other policy can be selected more than a quarter of the times, thus the candidate policy has to be $\pi^\star$ in all rounds $t \geq t'$. $\qquad \square$

**Lemma 34** (Number of Rounds in the First Phase). *Consider a run of Algorithm 2 with inputs $t_0 = 0, n = 1$, arbitrary policy $\widehat{\pi} \in \Pi_M$ and $M$ base learners $\mathbb{A}_1, \ldots, \mathbb{A}_M$ with $1 \leq R(\widetilde{\Pi}_1) \leq R(\widetilde{\Pi}_2) \leq \cdots \leq R(\widetilde{\Pi}_M)$ where $\widetilde{\Pi}_i$ is the extended version of policy class $\Pi_i$ with $(M + 1 - i)$ additional actions. Assume the environment $\mathbb{B}$ is stochastic and there is a policy $\pi_\star$ with gap $\Delta > 0$. Then with probability at least $1 - poly(M)\delta$ the number of rounds until the algorithm enters the second phase by calling ArbeGap-Exploit is bounded as*

$$\mathcal{O}\left( \left( \frac{R(\widetilde{\Pi}_M)^4}{R(\widetilde{\Pi}_1)^2} + M R(\widetilde{\Pi}_{i_\star})^2 \right) \frac{M i_\star}{\Delta^2} \ln^2 \frac{M R(\widetilde{\Pi}_{i_\star})}{\Delta\delta} \right) \ .$$

*Proof.* By Lemma 33, after a certain number of rounds $t_{pol}$, the candidate policy has to be $\pi_\star$ at all rounds. Hence, there can be no restarts due to candidate policy switches anymore. According to Lemma 28, there can only be up to $i_\star$ restarts after round $t_{pol}$ due to elimination of a base learner. We will in the following show that if ArbeGap has been (re)started with candidate policy $\pi_\star$ and there are no other restarts in the meantime, it has to switch to the second phase within a certain number of rounds $k$. The total number of rounds in the first phase, is then bounded by

$$t_{pol} + i_\star \cdot k \ .$$

We will now show a bound on $k$. By Lemma 31, we have at all times that $\widehat{\Delta}_t \leq \Delta$ (since $\widehat{\pi} = \pi_\star$ by assumption) and the algorithm moves on to the second phase as soon as $2W(t_0, t) \leq \widehat{\Delta}_t$. Note that the condition of $\widehat{\Delta}_t \leq \Delta \leq 1 \leq R(\widetilde{\Pi}_M)^2$ is always satisfies in stochastic environments. Hence, the algorithm cannot stay in the first phase if $2W(t_0, t) \leq \Delta$ or, plugging in the definition of $W(t_0, t)$ with an appropriate absolute constant $c$

$$\frac{c R(\widetilde{\Pi}_M)}{\sqrt{\rho_M}} \sqrt{\frac{\ln \frac{n(t-t_0)}{\delta}}{t - t_0}} + \frac{c}{\rho_M} \frac{\ln \frac{n \ln(t-t_0)}{\delta}}{t - t_0} \leq \Delta.$$

Hence, we can obtain an value for the bound $k$ by identifying a value that satisfies

$$\frac{\ln \frac{k}{\delta/n}}{k} \leq \frac{\Delta \rho_M}{2c} \quad \text{and} \quad \frac{\ln \frac{k}{\delta/n}}{k} \leq \frac{\Delta^2 \rho_M}{4c^2 R(\widetilde{\Pi}_M)^2} \ .$$

Since $\Delta \in (0,1]$ and $c, R(\widetilde{\Pi}_M) \geq 1$ without loss of generality, it is sufficient to only consider the condition on the right. By Lemma 53, we can set $k$ as

$$k = \frac{16c^2 R(\widetilde{\Pi}_M)^2}{\Delta^2 \rho_M} \ln\left(\frac{2n}{\delta} \frac{4c^2 R(\widetilde{\Pi}_M)^2}{\Delta^2 \rho_M}\right) = \mathcal{O}\left(\frac{MR(\widetilde{\Pi}_M)^4}{R(\widetilde{\Pi}_1)^2 \Delta^2} \ln \frac{R(\widetilde{\Pi}_M)}{\delta \Delta}\right),$$

where we used the fact that $n \leq \ln t_{pol} + i_\star = \mathcal{O}\left(\frac{R(\widetilde{\Pi}_M)}{\delta \Delta}\right)$. Hence, the total length of the first phase can be at most

$$
\begin{aligned}
t_{pol} + i_\star \cdot k &= \mathcal{O}\left(\frac{i_\star MR(\widetilde{\Pi}_M)^4}{R(\widetilde{\Pi}_1)^2 \Delta^2} \ln \frac{R(\widetilde{\Pi}_M)}{\Delta \delta} + \left(M^2 + \frac{R(\widetilde{\Pi}_{i_\star})^2}{R(\widetilde{\Pi}_1)^2} i_\star\right) \frac{R(\widetilde{\Pi}_{i_\star})^2 i_\star}{\Delta^2} \ln^2 \frac{MR(\widetilde{\Pi}_{i_\star})}{\Delta \delta}\right) \\
&= \mathcal{O}\left(\left(\frac{R(\widetilde{\Pi}_M)^4}{R(\widetilde{\Pi}_1)^2} + MR(\widetilde{\Pi}_{i_\star})^2\right) \frac{Mi_\star}{\Delta^2} \ln^2 \frac{MR(\widetilde{\Pi}_{i_\star})}{\Delta \delta}\right) \\
&= \mathcal{O}\left(\frac{M^2 i_\star R(\widetilde{\Pi}_M)^4}{R(\widetilde{\Pi}_1)^2 \Delta^2} \ln^2 \frac{MR(\widetilde{\Pi}_M)}{\Delta \delta}\right),
\end{aligned}
$$

as claimed. $\qquad\square$

**Lemma 35** (Pseudo-Regret of the First Phase). *Consider a run of Algorithm 2 with inputs $t_0 = 0, n = 1$, arbitrary policy $\widehat{\pi} \in \Pi_M$ and $M$ base learners $\mathbb{A}_1, \ldots, \mathbb{A}_M$ with $1 \leq R(\widetilde{\Pi}_1) \leq R(\widetilde{\Pi}_2) \leq \cdots \leq R(\widetilde{\Pi}_M)$ where $\widetilde{\Pi}_i$ is the extended version of policy class $\Pi_i$ with $(M+1-i)$ additional actions. Assume the environment $\mathbb{B}$ is stochastic and there is a policy $\pi_\star$ with gap $\Delta > 0$. Let $t_{gap}$ be the round where ArbeGap-Exploit was called. Then with probability at least $1 - poly(M)\delta$, the pseudo-regret in all rounds $t \in [2i_\star, t_{gap}]$ is bounded as*

$$\mathrm{PseudoReg}_{\mathbb{M}}(t, \Pi_M) = \mathcal{O}\left(\frac{R(\widetilde{\Pi}_{i_\star})^2 R(\widetilde{\Pi}_M)^2}{R(\widetilde{\Pi}_1)^2} \frac{M^2 i_\star}{\Delta} \ln\left(\frac{MR(\widetilde{\Pi}_M)}{\Delta \delta}\right) \ln \frac{t}{\delta}\right).$$

*Proof.* First, we bound the pseudo-regret by regret through a simple concentration argument in Lemma 42

$$\mathrm{PseudoReg}_{\mathbb{M}}(t, \Pi_M) \leq \mathrm{Reg}_{\mathbb{M}}(t, \Pi_M) + \mathcal{O}\left(\sqrt{t \ln \frac{\ln t}{\delta}}\right).$$

Next, we bound the regret by Lemma 30 which holds for any environment (adversarial but also stochastic). This yields

$$\mathrm{PseudoReg}_{\mathbb{M}}(t, \Pi_M) = \mathcal{O}\left(\left(MR(\widetilde{\Pi}_{i_\star}) + \frac{R(\widetilde{\Pi}_{i_\star})^2}{R(\widetilde{\Pi}_1)} \sqrt{i_\star}\right) \sqrt{t(\ln(t) + i_\star) \ln \frac{t}{\delta}}\right)$$

and finally, we use the bound on $t \leq t_{gap}$, the length of the first phase from Lemma 34 to replace $t(\ln(t) + i_\star)$ above which gives

$\mathrm{PseudoReg}_{\mathbb{M}}(t, \Pi_M)$

$$
\begin{aligned}
&= \mathcal{O}\left(\left(MR(\widetilde{\Pi}_{i_\star}) + \frac{R(\widetilde{\Pi}_{i_\star})^2}{R(\widetilde{\Pi}_1)} \sqrt{i_\star}\right) \frac{R(\widetilde{\Pi}_M)^2}{R(\widetilde{\Pi}_1)} \frac{M\sqrt{i_\star}}{\Delta} \ln\left(\frac{MR(\widetilde{\Pi}_M)}{\Delta \delta}\right) \sqrt{(\ln(t) + i_\star) \ln \frac{t}{\delta}}\right) \\
&= \mathcal{O}\left(\left(M + \frac{R(\widetilde{\Pi}_{i_\star})}{R(\widetilde{\Pi}_1)} \sqrt{i_\star}\right) \frac{Mi_\star R(\widetilde{\Pi}_{i_\star}) R(\widetilde{\Pi}_M)^2}{R(\widetilde{\Pi}_1) \Delta} \ln\left(\frac{MR(\widetilde{\Pi}_M)}{\Delta \delta}\right) \ln \frac{t}{\delta}\right) \\
&= \mathcal{O}\left(\frac{M^2 i_\star R(\widetilde{\Pi}_{i_\star})^2 R(\widetilde{\Pi}_M)^2}{R(\widetilde{\Pi}_1)^2 \Delta} \ln\left(\frac{MR(\widetilde{\Pi}_M)}{\Delta \delta}\right) \ln \frac{t}{\delta}\right),
\end{aligned}
$$

as claimed. $\qquad\square$

## D.3 Analysis of **Arbe-GapExploit**

In this section we prove the following result:

**Lemma 36** (Guarantee for Second Phase). *Let $\mathbb{A}$ be an h-stable learner with policy class $\Pi_{\mathbb{A}}$. Then the regret of Algorithm 4 against $\Pi_{\mathbb{A}} \cup \{\widehat{\pi}\}$ is bounded with probability at least $1 - \mathcal{O}(\delta)$ for all rounds $t > t_0$ that the algorithm has not terminated yet as*

$$\text{Reg}_{\mathbb{M}}([t_0 + 1, t], \Pi_{\mathbb{A}} \cup \{\widehat{\pi}\}) = \mathcal{O}\left(\frac{R(\Pi_{\mathbb{A}})^2}{\widehat{\Delta}}\left(\ln(t)\ln\frac{t}{\delta} + \ln\frac{R(\Pi_{\mathbb{A}})}{\widehat{\Delta}\delta}\right) + \sqrt{(t - t_0)\ln(t)\ln\frac{t}{\delta}}\right).$$

*Further, if the environment is stochastic with an optimal policy $\pi_\star$ that has a gap $\Delta$ compared to the best policy in $\Pi_{\mathbb{A}}$ and the inputs satisfy $\widehat{\pi} = \pi_\star$ and $\widehat{\Delta} \leq \Delta \leq 2\widehat{\Delta}$, then with probability at least $1 - \mathcal{O}(\delta)$ the pseudo-regret of Algorithm 4 is bounded in all rounds $t > t_0$ as*

$$\text{PseudoReg}_{\mathbb{M}}([t_0 + 1, t], \{\widehat{\pi}\} \cup \Pi_{\mathbb{A}}) = \mathcal{O}\left(\frac{R(\Pi_{\mathbb{A}})^2}{\Delta}\left(\ln(t)\ln\frac{t}{\delta} + \ln\frac{R(\Pi_{\mathbb{A}})}{\Delta\delta}\right)\right)$$

*and the algorithm never terminates.*

### D.3.1 Guarantees for stochastic environments

**Lemma 37** (Algorithm 4 does not terminate in stochastic environemnts). *Assume the environment is stochastic and there is an optimal policy $\pi_\star$ with a gap $\Delta$ compared to the best policy in $\Pi_{\mathbb{A}}$. If Algorithm 4 is called with inputs $\widehat{\pi} = \pi_\star$ and $\widehat{\Delta} \leq \Delta \leq 2\widehat{\Delta}$, then with probability at least $1 - \mathcal{O}(\delta)$ it never terminates.*

*Proof.* Let $e$ be an arbitrary epoch. By following the steps of Lemma 23, we can show that with probability at least $1 - \mathcal{O}(\delta_e)$ for all time steps in epoch $e$

$$\left|\widetilde{\text{CRew}}_1(t_e + 1, t) - \sum_{\ell=t_e+1}^{t} r_\ell(a_\ell^1, x_\ell)\right| \leq \mathcal{O}\left(\sqrt{\frac{t - t_e}{\rho^e}\ln\frac{\ln(t - t_e)}{\delta_e}} + \frac{1}{\rho^e}\ln\frac{\ln(t - t_e)}{\delta_e}\right)$$

$$\left|\widetilde{\text{CRew}}_0(t_e + 1, t) - \sum_{\ell=t_e+1}^{t} r_\ell(a_\ell^0, x_\ell)\right| \leq \mathcal{O}\left(\sqrt{\frac{t - t_e}{1 - \rho^e}\ln\frac{\ln(t - t_e)}{\delta_e}} + \frac{1}{1 - \rho^e}\ln\frac{\ln(t - t_e)}{\delta_e}\right)$$

and for $i \in \{0, 1\}$

$$\left|\sum_{\ell=t_e+1}^{t} r_\ell(a_\ell^i, x_\ell) - \sum_{\ell=t_e+1}^{t} \mathbb{E}_{a\sim\pi_\ell^i}[r_\ell(a, x_\ell)]\right| \leq \mathcal{O}\left(\sqrt{(t - t_e)\ln\frac{\ln(t - t_e)}{\delta_e}}\right).$$

Combining these bound together with Lemma 42, we have with $\pi_{\star,\mathbb{A}}$ being the best policy in $\Pi_{\mathbb{A}}$

$$\widetilde{\text{CRew}}_0(t_e + 1, t) - \widetilde{\text{CRew}}_1(t_e + 1, t)$$

$$\leq \sum_{\ell=t_e+1}^{t}\left[\mathbb{E}_{a\sim\pi_\ell^0, x\sim\mathcal{D}}[r(a, x)] - \mathbb{E}_{a\sim\pi_\ell^1, x\sim\mathcal{D}}[r(a, x)]\right]$$

$$+ \mathcal{O}\left(\sqrt{\frac{t - t_e}{\rho^e}\ln\frac{\ln(t - t_e)}{\delta_e}} + \frac{1}{\rho^e}\ln\frac{\ln(t - t_e)}{\delta_e}\right)$$

$$= \sum_{\ell=t_e+1}^{t}\left[\mathbb{E}_{a\sim\pi_\star, x\sim\mathcal{D}}[r(a, x)] - \mathbb{E}_{a\sim\pi_{\star,\mathbb{A}}, x\sim\mathcal{D}}[r(a, x)]\right] + \mathcal{O}\left(\sqrt{\frac{t - t_e}{\rho^e}\ln\frac{\ln(t - t_e)}{\delta_e}}\right)$$

$$+ \sum_{\ell=t_e+1}^{t}\left[\mathbb{E}_{a\sim\pi_{\star,\mathbb{A}}, x\sim\mathcal{D}}[r(a, x)] - \mathbb{E}_{a\sim\pi_\ell^1, x\sim\mathcal{D}}[r(a, x)]\right] + \mathcal{O}\left(\frac{1}{\rho^e}\ln\frac{\ln(t - t_e)}{\delta_e}\right)$$

$$= \Delta(t - t_e) + \text{PseudoReg}_{\mathbb{A}}([t_e + 1, t], \Pi_{\mathbb{A}}) + \mathcal{O}\left(\sqrt{\frac{t - t_e}{\rho^e}\ln\frac{\ln(t - t_e)}{\delta_e}} + \frac{1}{\rho^e}\ln\frac{\ln(t - t_e)}{\delta_e}\right)$$

$$\leq \Delta(t - t_e) + \text{Reg}_{\mathbb{A}}([t_e + 1, t], \Pi_{\mathbb{A}}) + \mathcal{O}\left(\sqrt{\frac{t - t_e}{\rho^e} \ln \frac{\ln(t - t_e)}{\delta_e}} + \frac{1}{\rho^e} \ln \frac{\ln(t - t_e)}{\delta_e}\right)$$

(Lemma 42)

$$\leq \Delta(t - t_e) + \mathcal{O}\left(R(\Pi_{\mathbb{A}})\sqrt{\frac{t - t_e}{\rho^e} \ln \frac{t - t_e}{\delta_e}} + \frac{1}{\rho^e} \ln \frac{\ln(t - t_e)}{\delta_e}\right)$$

($\mathbb{A}$ is h-stable)

$$\leq 2\widehat{\Delta}(t - t_e) + \mathcal{O}\left(R(\Pi_{\mathbb{A}})\sqrt{\frac{t - t_e}{\rho^e} \ln \frac{t - t_e}{\delta_e}} + \frac{1}{\rho^e} \ln \frac{\ln(t - t_e)}{\delta_e}\right)$$

($\Delta \leq 2\widehat{\Delta}$)

where we used the fact that $\rho^e \leq 1/2 \leq 1 - \rho^e$ by the choice of constants (see Lemma 39). This chain of inequalities holds with probability $1 - \mathcal{O}(\delta_e)$. Combining this with a union bound, this implies that with this probability at least $1 - \mathcal{O}(\delta)$, the test Line 16 never triggers. Similarly, we can lower-bound the same term as

$$\widetilde{\text{CRew}}_0(t_e + 1, t) - \widetilde{\text{CRew}}_1(t_e + 1, t)$$

$$\geq \sum_{\ell = t_e + 1}^{t} \left[\mathbb{E}_{a \sim \pi_\ell^0, x \sim \mathcal{D}}[r(a, x)] - \mathbb{E}_{a \sim \pi_\ell^1, x \sim \mathcal{D}}[r(a, x)]\right]$$

$$- \mathcal{O}\left(\sqrt{\frac{t - t_e}{\rho^e} \ln \frac{\ln(t - t_e)}{\delta_e}} + \frac{1}{\rho^e} \ln \frac{\ln(t - t_e)}{\delta_e}\right)$$

$$\geq \sum_{\ell = t_e + 1}^{t} \left[\mathbb{E}_{a \sim \pi_\star, x \sim \mathcal{D}}[r(a, x)] - \mathbb{E}_{a \sim \pi_{\star, \mathbb{A}}, x \sim \mathcal{D}}[r(a, x)]\right]$$

($\pi_\ell^0 = \pi_\star$)

$$- \mathcal{O}\left(\sqrt{\frac{t - t_e}{\rho^e} \ln \frac{\ln(t - t_e)}{\delta_e}} + \frac{1}{\rho^e} \ln \frac{\ln(t - t_e)}{\delta_e}\right)$$

$$\geq \Delta(t - t_e) - \mathcal{O}\left(\sqrt{\frac{t - t_e}{\rho^e} \ln \frac{\ln(t - t_e)}{\delta_e}} + \frac{1}{\rho^e} \ln \frac{\ln(t - t_e)}{\delta_e}\right)$$

$$\geq \widehat{\Delta}(t - t_e) - \mathcal{O}\left(\sqrt{\frac{t - t_e}{\rho^e} \ln \frac{\ln(t - t_e)}{\delta_e}} + \frac{1}{\rho^e} \ln \frac{\ln(t - t_e)}{\delta_e}\right)$$

($\widehat{\Delta} \leq \Delta$)

Hence, after combining these statements with a union bound, this implies that with this probability at least $1 - \mathcal{O}(\delta)$, the test Line 14 never triggers. A final union bound for both tests completes the proof. $\qquad\square$

**Lemma 38** (Pseudo-regret of Algorithm 4). *Assume the environment is stochastic and there is an optimal policy $\pi_\star$ with a gap $\Delta$. If Algorithm 4 is called with inputs $\widehat{\pi} = \pi_\star$ and $\widehat{\Delta} = \Theta(\Delta)$, then the pseudo-regret of the algorithm is bounded with probability at least $1 - \mathcal{O}(\delta)$ as*

$$\text{PseudoReg}_{\mathbb{M}}([t_0 + 1, t], \{\widehat{\pi}\} \cup \Pi_{\mathbb{A}}) = \mathcal{O}\left(\frac{R(\Pi_{\mathbb{A}})^2}{\Delta} \left(\ln \frac{R(\Pi_{\mathbb{A}})}{\Delta\delta} + \ln(t) \ln \frac{t}{\delta}\right)\right)$$

*for all rounds $t \geq t_0$ where the algorithm has not terminated.*

*Proof.* The pseudo-regret of Algorithm 4 can be decomposed into the regret in each epoch as

$$\text{PseudoReg}_{\mathbb{M}}([t_0 + 1, t], \{\widehat{\pi}\} \cup \Pi_{\mathbb{A}}) = \sum_{e=0}^{j(t)} \text{PseudoReg}_{\mathbb{M}}([t_e + 1, t_e'], \{\widehat{\pi}\} \cup \Pi_{\mathbb{A}})$$

where $t'_e = \min\{t_{e+1}, t\}$ and $j(t) = \min\{e \in \mathbb{N}: t_{e+1} \geq t\}$ is the epoch at time $t$. We consider the regret in each epoch separately as

$$\text{PseudoReg}_{\mathbb{M}}([t_e + 1, t'_e], \{\widehat{\pi}\} \cup \Pi_{\mathbb{A}})$$

$$= \sum_{\ell=t_e+1}^{t'_e} (\mathbb{E}_{a \sim \pi^\star, x \sim \mathcal{D}}[r(a, x)] - \mathbb{E}_{a \sim \pi_t, x \sim \mathcal{D}}[r(a, x)])$$

$$= \sum_{\ell=t_e+1}^{t'_e} b_\ell \left( \mathbb{E}_{a \sim \pi^\star, x \sim \mathcal{D}}[r(a, x)] - \mathbb{E}_{a \sim \pi_\ell^1, x \sim \mathcal{D}}[r(a, x)] \right) \qquad (\widehat{\pi} = \pi^\star)$$

$$\leq \sum_{\ell=t_e+1}^{t'_e} b_\ell \left( \Delta + \mathbb{E}_{a \sim \pi_{\mathbb{A}}^\star, x \sim \mathcal{D}}[r(a, x)] - \mathbb{E}_{a \sim \pi_\ell^1, x \sim \mathcal{D}}[r(a, x)] \right)$$

where $\pi_{\mathbb{A}}^\star$ is the best policy in $\Pi_{\mathbb{A}}$ which incurs a pseudo-regret of at most $\Delta$ per round. We now apply a concentration argument. Denote $Y_\ell = \mathbf{1}\{\ell > t_e\} \left( \Delta + \mathbb{E}_{a \sim \pi_{\mathbb{A}}^\star, x \sim \mathcal{D}}[r(a, x)] - \mathbb{E}_{a \sim \pi_\ell, x \sim \mathcal{D}}[r(a, x)] \right)$ and let $\mathcal{F}_\ell$ be the sigma-field that includes $\{\pi_t^1\}_{t>t_0}^\ell$, $\{b_t\}_{t>t_0}^{\ell-1}$ and $t_e$. Note that $Y_\ell$ is $\mathcal{F}_\ell$-measurable and $\sum_{\ell=t_e+1}^{t'_e}(b_\ell - \rho^e)Y_\ell$ is a martingale difference sequence w.r.t. $\mathcal{F}_\ell$. Since $|(b_\ell - \rho^e)Y_\ell| \leq 2$ and the sequence of conditional variance is bounded as $V_\ell = \sum_{\ell=t_e+1}^{t'_e} \rho^e(1-\rho^e)Y_\ell \leq \sum_{\ell=t_e+1}^{t'_e} \rho^e Y_\ell$, we can apply Lemma 47 and get with probability at least $1 - \delta_e$ for all rounds $t_e \leq \tilde{t} \leq t_{e+1}$

$$\sum_{\ell=t_e+1}^{\tilde{t}} (b_\ell - \rho^e)Y_\ell \leq \mathcal{O}\left( \sqrt{\sum_{\ell=t_e+1}^{\tilde{t}} \rho^e Y_\ell \ln \frac{\ln(\tilde{t} - t_e)}{\delta_e}} + \ln \frac{\ln(\tilde{t} - t_e)}{\delta_e} \right)$$

$$\leq \sum_{\ell=t_e+1}^{\tilde{t}} \rho^e Y_\ell + \leq \mathcal{O}\left( \ln \frac{\ln(\tilde{t} - t_e)}{\delta_e} \right) \qquad \text{(AM-GM inequality)}$$

This holds in particular for $\tilde{t} = t'_e$ and with shorthand $k'_e = t'_e - t_e$, this gives

$$\text{PseudoReg}_{\mathbb{M}}([t_e + 1, t'_e], \{\widehat{\pi}\} \cup \Pi_{\mathbb{A}})$$

$$\leq 2\rho^e \sum_{\ell=t_e+1}^{t'_e} \left( \Delta + \mathbb{E}_{a \sim \pi_{\mathbb{A}}^\star, x \sim \mathcal{D}}[r(a, x)] - \mathbb{E}_{a \sim \pi_\ell, x \sim \mathcal{D}}[r(a, x)] \right) + \mathcal{O}\left( \ln \frac{\ln(k'_e)}{\delta_e} \right)$$

$$= 2\rho^e k'_e \Delta + \rho^e \text{PseudoReg}_{\mathbb{A}}([t_e + 1, t'_e], \Pi_{\mathbb{A}}) + \mathcal{O}\left( \ln \frac{\ln(k'_e)}{\delta_e} \right)$$

$$\leq 2\rho^e k'_e \Delta + \rho^e \text{Reg}_{\mathbb{A}}([t_e + 1, t'_e], \Pi_{\mathbb{A}}) + \mathcal{O}\left( \rho^e \sqrt{k'_e \ln \frac{\ln(k'_e)}{\delta_e}} + \ln \frac{\ln(k'_e)}{\delta_e} \right) \qquad \text{(Lemma 42)}$$

$$\leq 2\rho^e k'_e \Delta + \mathcal{O}\left( \rho^e R(\Pi_{\mathbb{A}}) \sqrt{\frac{k'_e}{\rho^e} \ln \frac{k'_e}{\delta_e}} + \ln \frac{\ln(k'_e)}{\delta_e} \right) \qquad (\mathbb{A} \text{ is h-stable})$$

$$\leq 2\rho^e k'_e \Delta + \mathcal{O}\left( R(\Pi_{\mathbb{A}}) \sqrt{\rho^e k'_e \ln \frac{k'_e}{\delta_e}} + \ln \frac{\ln(k'_e)}{\delta_e} \right)$$

$$\leq \mathcal{O}\left( \frac{R(\Pi_{\mathbb{A}})^2 \Delta}{\widehat{\Delta}^2} \ln \frac{k_e}{\delta_e} + R(\Pi_{\mathbb{A}}) \sqrt{\frac{R(\Pi_{\mathbb{A}})^2}{\widehat{\Delta}^2} \ln \left( \frac{k_e}{\delta_e} \right) \ln \frac{k'_e}{\delta_e}} + \ln \frac{\ln(k'_e)}{\delta_e} \right)$$

$$\text{(definition of } \rho^e \text{ and } k'_e \leq k_e)$$

$$\leq \mathcal{O}\left( \frac{R(\Pi_{\mathbb{A}})^2}{\Delta} \ln \frac{k_e}{\delta_e} \right) \qquad (\widehat{\Delta} = \Theta(\Delta))$$

Now, we can combine this pseudo-regret bound across all epochs and get

$$
\begin{aligned}
\mathrm{PseudoReg}_{\mathbb{M}}([t_0+1,t],\{\widehat{\pi}\}\cup\Pi_{\mathbb{A}}) &\leq \sum_{e=0}^{j(t)} \mathcal{O}\left(\frac{R(\Pi_{\mathbb{A}})^2}{\Delta}\ln\frac{k_e}{\delta_e}\right)\\
&\leq \mathcal{O}\left(j(t)\frac{R(\Pi_{\mathbb{A}})^2}{\Delta}\ln\frac{tj(t)}{\delta}\right)\\
&= \mathcal{O}\left(\frac{R(\Pi_{\mathbb{A}})^2}{\Delta}\ln(t)\ln\frac{t}{\delta}\right) \qquad (j(t)=\mathcal{O}(\ln(t)))
\end{aligned}
$$

when $j(t)\geq 1$ since $k_e\leq k_{j(t)}\leq 2t$ in this case. In the other case, where $j(t)=0$, we have

$$
\mathrm{PseudoReg}_{\mathbb{M}}([t_0+1,t],\{\widehat{\pi}\}\cup\Pi_{\mathbb{A}}) \leq \mathcal{O}\left(\frac{R(\Pi_{\mathbb{A}})^2}{\Delta}\ln\frac{k_0}{\delta_0}\right) \leq \mathcal{O}\left(\frac{R(\Pi_{\mathbb{A}})^2}{\Delta}\ln\frac{R(\Pi_{\mathbb{A}})}{\Delta\delta}\right).
$$

Hence, combining both cases gives the final bound

$$
\mathrm{PseudoReg}_{\mathbb{M}}([t_0+1,t],\{\widehat{\pi}\}\cup\Pi_{\mathbb{A}}) = \mathcal{O}\left(\frac{R(\Pi_{\mathbb{A}})^2}{\Delta}\left(\ln\frac{R(\Pi_{\mathbb{A}})}{\Delta\delta}+\ln(t)\ln\frac{t}{\delta}\right)\right).
$$

$\qquad\qquad\qquad\qquad\qquad\qquad\qquad\qquad\qquad\qquad\qquad\qquad\qquad\qquad\qquad\qquad\qquad\qquad\qquad\square$

### D.3.2 Exploitation Subroutine Guarantees for Adversarial Environments

**Lemma 39.** *Assume the absolute constant in the length of the initial epoch $k_0$ of Algorithm 4 is chosen large enough. Then the regret of Algorithm 4 against $\Pi_{\mathbb{A}}$ is bounded with probability at least $1-\mathcal{O}(\delta)$ for all rounds $t$ that the algorithm has not terminate yet as*

$$
\mathrm{Reg}_{\mathbb{M}}([t_0+1,t],\Pi_{\mathbb{A}}) = \mathcal{O}(\widehat{\Delta}k_0).
$$

*Proof.* Consider any round $t>t_0$ before the test in Line 14 or 16 triggers. The total reward in relevant rounds can be decomposed into epochs $e=0,\ldots,j(t)$ as follows where $j(t)=\max\{e\in\mathbb{N}\colon t_e<t\}$ is the epoch of round $t$ and $t'_e=\min\{t_{e+1},t\}$.

$$
\sum_{\ell=t_0+1}^{t} r_\ell(a_\ell,x_\ell) = \sum_{e=0}^{j(t)}\sum_{\ell=t_e+1}^{t'_e} r_\ell(a_\ell,x_\ell)
$$

Further, let $k'_e=t'_e-t_e$ be the number of rounds in the $e$-th epoch and consider now a single epoch $e$. We can write

$$
\begin{aligned}
&\sum_{\ell=t_e+1}^{t'_e} r_\ell(a_\ell,x_\ell)\\
&= \rho^e\widetilde{\mathrm{CRew}}_1(t_e+1,t'_e)+(1-\rho^e)\widetilde{\mathrm{CRew}}_0(t_e+1,t'_e)\\
&\geq \rho^e\widetilde{\mathrm{CRew}}_1(t_e+1,t'_e)+(1-\rho^e)\left(\widetilde{\mathrm{CRew}}_1(t_e+1,t'_e)+\widehat{\Delta}k'_e-V(t'_e)k'_e\right)\\
&\hspace{9cm}\text{(first test did not trigger)}\\
&\geq \widetilde{\mathrm{CRew}}_1(t_e+1,t'_e)+(1-\rho^e)\left(\widehat{\Delta}k'_e-V(t'_e)k'_e\right)\\
&\geq \sum_{\ell=t_e+1}^{t'_e} r_\ell(a_\ell^1,x_\ell)+(1-\rho^e)\left(\widehat{\Delta}k'_e-V(t'_e)k'_e\right)-\mathcal{O}\left(\sqrt{\frac{k'_e}{\rho^e}\ln\frac{\ln(k'_e)}{\delta_e}}+\frac{1}{\rho^e}\ln\frac{\ln(k'_e)}{\delta_e}\right)\\
&\hspace{9cm}\text{(see proof of Lemma 37)}\\
&\geq \max_{\pi\in\Pi_{\mathbb{A}}}\sum_{\ell=t_e+1}^{t'_e}\mathbb{E}_{a\sim\pi}[r_\ell(a,x_\ell)]-\mathcal{O}\left(R(\Pi_{\mathbb{A}})\sqrt{\frac{k'_e}{\rho^e}\ln\frac{k'_e}{\delta_e}}+\frac{1}{\rho^e}\ln\frac{\ln(k'_e)}{\delta_e}\right)\\
&\quad + (1-\rho^e)\left(\widehat{\Delta}k'_e-V(t'_e)k'_e\right) \hspace{5.5cm}\text{($\mathbb{A}$ is $h$-stable)}
\end{aligned}
$$

$$\geq \max_{\pi \in \Pi_{\mathbb{A}}} \sum_{\ell=t_e+1}^{t'_e} \mathbb{E}_{a \sim \pi}[r_\ell(a, x_\ell)] + (1 - \rho^e)\widehat{\Delta}k'_e - \mathcal{O}\left(R(\Pi_{\mathbb{A}})\sqrt{\frac{k'_e}{\rho^e}\ln\frac{k'_e}{\delta_e}} + \frac{1}{\rho^e}\ln\frac{\ln(k'_e)}{\delta_e}\right)$$

$$\text{(definition of } V(t'_e))$$

Hence, by rearranging terms, we get that the regret in a single epoch is bounded as

$$\text{Reg}_{\mathbb{M}}([t_e + 1, t'_e], \Pi_{\mathbb{A}}) \leq (\rho^e - 1)\widehat{\Delta}k'_e + \mathcal{O}\left(R(\Pi_{\mathbb{A}})\sqrt{\frac{k'_e}{\rho^e}\ln\frac{k'_e}{\delta_e}} + \frac{1}{\rho^e}\ln\frac{\ln(k'_e)}{\delta_e}\right).$$

To further upper-bound these terms, we first derive useful expression for the inverse probability of playing $\mathbb{A}$. Here, we make the constants in the definition of the initial epoch length $k_0$ explicit. Specifically, we assume that $k_0 = \frac{c_0 R(\Pi_{\mathbb{A}})^2}{\widehat{\Delta}^2}\ln\frac{2c_0 R(\Pi_{\mathbb{A}})}{\widehat{\Delta}\delta}$ where $c_0$ is the absolute constant. We have

$$\frac{1}{\rho^e} = \frac{k_e\widehat{\Delta}^2}{c_\rho R(\Pi_{\mathbb{A}})^2 \ln\frac{k_e}{\delta_e}}$$

Plugging this bound on the inverse probability back into the expression for the regret per epoch above, we get with $c$ as the constant in the $\mathcal{O}$ notation

$$\text{Reg}_{\mathbb{M}}([t_e + 1, t'_e], \Pi_{\mathbb{A}}) \leq (\rho^e - 1)\widehat{\Delta}k'_e + \widehat{\Delta}\sqrt{\frac{6ck'_e k_e}{c_\rho}} + \frac{6c\widehat{\Delta}^2 k_e}{c_0 R(\Pi_{\mathbb{A}})^2}$$

$$= \widehat{\Delta}k'_e\left(-1 + \rho^e + \sqrt{\frac{6ck_e}{c_\rho k'_e}} + \frac{6c\widehat{\Delta}k_e}{k'_e c_\rho R(\Pi_{\mathbb{A}})^2}\right)$$

$$\leq \frac{\widehat{\Delta}}{2}\left(-k'_e + \sqrt{\frac{12c}{c_\rho}k_e k'_e} + \frac{12c}{c_\rho}\frac{\widehat{\Delta}}{R(\Pi_{\mathbb{A}})}k_e\right). \qquad (\rho^e \leq 1/2)$$

In the last step, we used $\rho^e \leq \rho^0 \leq 1/2$ which we can establish by choosing the constants $c_\rho$ and $c_0$ in the definition of $\rho$ and $k_0$ appropriately. Specifically,

$$\rho^e \leq \rho^0 = c_\rho \frac{R(\Pi_{\mathbb{A}})^2 \ln\frac{k_0}{\delta_0}}{\widehat{\Delta}^2 k_0} = \frac{c_\rho}{c_0}\frac{\ln\frac{k_0}{\delta_0}}{\ln\frac{R(\Pi_{\mathbb{A}})}{\delta\widehat{\Delta}}} \leq \frac{c_\rho}{c_0}\frac{\ln\frac{c_0 R(\Pi_{\mathbb{A}})^3}{\delta^2\widehat{\Delta}^3}}{\ln\frac{c_0 R(\Pi_{\mathbb{A}})}{\delta\widehat{\Delta}}} \leq 3\frac{c_\rho}{c_0}.$$

Thus, choosing $c_0 \geq 6c_\rho$ is sufficient to ensure $\rho^e \leq 1/2$. Summing now the bound above over epochs gives

$$\text{Reg}_{\mathbb{M}}([t_0 + 1, t], \Pi_{\mathbb{A}}) \leq \frac{\widehat{\Delta}}{2}\left(-t + t_0 + \sqrt{\frac{12c}{c_\rho}}\sum_{e=0}^{j(t)}\sqrt{k_e k'_e} + \frac{12c}{c_\rho}\frac{\widehat{\Delta}}{R(\Pi_{\mathbb{A}})}\sum_{e=0}^{j(t)}k_e\right).$$

We now distinguish between two cases. First consider the case where $j(t) \geq 1$ and assume that $c_\rho \geq 9 \cdot 12c$. Since in this case, the number of rounds $t - t_0$ is at least half the sum of epoch lengths, $\sum_{e=0}^{j(t)} k_e$, we have

$$\text{Reg}_{\mathbb{M}}([t_0 + 1, t], \Pi_{\mathbb{A}}) \leq \frac{\widehat{\Delta}}{2}\left(-t + t_0 + \frac{1}{3}\sum_{e=0}^{j(t)}\sqrt{k_e k'_e} + \frac{1}{9}\sum_{e=0}^{j(t)}k_e\right)$$

$$\leq \frac{\widehat{\Delta}}{2}\left(-t + t_0 + \frac{4}{9}\sum_{e=0}^{j(t)}k_e\right) \leq \frac{\widehat{\Delta}}{2}\left(-t + t_0 + \frac{8}{9}(t - t_0)\right) \leq 0.$$

In the other case, when $j(t) = 0$, we have

$$\text{Reg}_{\mathbb{M}}([t_0 + 1, t], \Pi_{\mathbb{A}}) \leq \frac{\widehat{\Delta}}{2}\left(-k'_0 + \sqrt{\frac{12c}{c_\rho}}\sqrt{k_0 k'_0} + \frac{12c}{c_\rho}\frac{\widehat{\Delta}}{R(\Pi_{\mathbb{A}})}k_0\right) = \mathcal{O}(\widehat{\Delta}k_0).$$

Hence, the regret against $\Pi_{\mathbb{A}}$ is always bounded as

$$\text{Reg}_{\mathbb{M}}([t_0 + 1, t], \Pi_{\mathbb{A}}) = \mathcal{O}(\widehat{\Delta}k_0).$$

$\square$

**Lemma 40.** *The regret of [Algorithm 4](#) against $\widehat{\pi}$ is bounded with probability at least $1 - \mathcal{O}(\delta)$ for all rounds $t$ that the algorithm has not terminated yet as*

$$\mathrm{Reg}_{\mathbb{M}}([t_0 + 1, t], \{\widehat{\pi}\}) = \mathcal{O}\left(\frac{R(\Pi_{\mathbb{A}})^2}{\widehat{\Delta}}\left(\ln(t)\ln\frac{t}{\delta} + \ln\frac{R(\Pi_{\mathbb{A}})}{\Delta\delta}\right) + \sqrt{(t - t_0)\ln(t)\ln\frac{t}{\delta}}\right).$$

*Proof.* Consider any round $t > t_0$ before the test in Line [14](#) or [16](#) triggers. The total reward in relevant rounds can be decomposed into epochs $e = 0, \ldots, j(t)$ as follows where $j(t) = \max\{e \in \mathbb{N}: t_e < t\}$ is the epoch of round $t$ and $t'_e = \min\{t_{e+1}, t\}$.

$$\sum_{\ell=t_0+1}^{t} r_\ell(a_\ell, x_\ell) = \sum_{e=0}^{j(t)} \sum_{\ell=t_e+1}^{t'_e} r_\ell(a_\ell, x_\ell)$$

Further, let $k'_e = t'_e - t_e$ be the number of rounds in the $e$-th epoch and consider now a single epoch $e$. We can write

$$\sum_{\ell=t_e+1}^{t'_e} r_\ell(a_\ell, x_\ell)$$

$$= (1 - \rho^e)\widetilde{\mathrm{CRew}}_0(t_e + 1, t'_e) + \rho^e\widetilde{\mathrm{CRew}}_1(t_e + 1, t'_e)$$

$$\geq (1 - \rho^e)\widetilde{\mathrm{CRew}}_0(t_e + 1, t'_e) + \rho^e\left(\widetilde{\mathrm{CRew}}_0(t_e + 1, t'_e) - 4\widehat{\Delta}k'_e - V(t'_e)k'_e\right)$$

$$\text{(second test did not trigger)}$$

$$\geq \widetilde{\mathrm{CRew}}_0(t_e + 1, t'_e) - \rho^e\left(4\widehat{\Delta}k'_e + V(t'_e)k'_e\right)$$

$$\geq \sum_{\ell=t_e+1}^{t'_e} r_\ell(a^1_\ell, x_\ell) - \rho^e\left(4\widehat{\Delta}k'_e + V(t'_e)k'_e\right) - \mathcal{O}\left(\sqrt{\frac{k'_e}{1 - \rho^e}\ln\frac{\ln(k'_e)}{\delta_e}} + \frac{1}{1 - \rho^e}\ln\frac{\ln k'_e}{\delta_e}\right)$$

$$\text{(see proof of [Lemma 37](#))}$$

$$\geq \sum_{\ell=t_e+1}^{t'_e} \mathbb{E}_{a\sim\widehat{\pi}}[r_\ell(a, x_\ell)] - \rho^e\left(4\widehat{\Delta}k'_e + V(t'_e)k'_e\right) - \mathcal{O}\left(\sqrt{\frac{k'_e}{1 - \rho^e}\ln\frac{\ln(k'_e)}{\delta_e}} + \frac{1}{1 - \rho^e}\ln\frac{\ln k'_e}{\delta_e}\right).$$

The last step here follows since $\rho^e k'_e V(t'_e) \leq \mathcal{O}(R(\Pi_{\mathbb{A}})\sqrt{\rho^e k'_e \ln\frac{k'_e}{\delta_e}} + \ln\frac{\ln k'_e}{\delta_e})$ By rearranging terms, we can bound the regret against $\widehat{\pi}$ in epoch $e$ as

$$\mathrm{Reg}_{\mathbb{M}}([t_e + 1, t'_e], \{\widehat{\pi}\})$$

$$\leq \mathcal{O}\left(\rho^e k'_e(\widehat{\Delta} + V(t'_e)) + \sqrt{\frac{k'_e}{1 - \rho^e}\ln\frac{k'_e}{\delta_e}} + \frac{1}{1 - \rho^e}\ln\frac{\ln k'_e}{\delta_e}\right)$$

$$\leq \mathcal{O}\left(\frac{R(\Pi_{\mathbb{A}})^2}{\widehat{\Delta}}\ln\frac{k_e}{\delta_e} + \sqrt{\frac{k'_e}{1 - \rho^e}\ln\frac{k'_e}{\delta_e}} + \frac{1}{1 - \rho^e}\ln\frac{\ln k'_e}{\delta_e}\right)$$

$$\text{(definition of } \rho^e, V(t'_e) \text{ and } k_e \geq k'_e)$$

$$\leq \mathcal{O}\left(\frac{R(\Pi_{\mathbb{A}})^2}{\widehat{\Delta}}\ln\frac{k_e}{\delta_e} + \sqrt{\frac{k'_e}{1 - \rho^e}\ln\frac{k'_e}{\delta_e}} + \frac{1}{1 - \rho^e}\ln\frac{\ln k'_e}{\delta_e}\right)$$

$$\leq \mathcal{O}\left(\frac{R(\Pi_{\mathbb{A}})^2}{\widehat{\Delta}}\ln\frac{k_e}{\delta_e} + \sqrt{k'_e \ln\frac{k'_e}{\delta_e}} + \ln\frac{\ln k'_e}{\delta_e}\right) \qquad (\rho^e \leq 1/2)$$

$$\leq \mathcal{O}\left(\frac{R(\Pi_{\mathbb{A}})^2}{\widehat{\Delta}}\ln\frac{k_e}{\delta_e} + \sqrt{k'_e \ln\frac{k'_e}{\delta_e}}\right).$$

Note that $\rho^e \leq 1/2$ holds for appropriate constants in the definition of $k_0$ and $\rho^e$ (see Lemma 39). We can now sum the regret over all epochs $e$ and get

$$\mathrm{Reg}_{\mathbb{M}}([t_0 + 1, t], \{\widehat{\pi}\}) \leq \sum_{e=0}^{j(t)} \mathcal{O}\left( \frac{R(\Pi_{\mathbb{A}})^2}{\widehat{\Delta}} \ln \frac{k_e}{\delta_e} + \sqrt{k_e' \ln \frac{k_e'}{\delta_e}} \right)$$

$$\leq \sum_{e=0}^{j(t)} \mathcal{O}\left( \frac{R(\Pi_{\mathbb{A}})^2}{\widehat{\Delta}} \ln \frac{(j(t) + 1)k_{j(t)}}{\delta} + \sqrt{k_e' \ln \frac{(j(t) + 1)k_{j(t)}}{\delta}} \right)$$

$$\leq \mathcal{O}\left( (j(t) + 1) \frac{R(\Pi_{\mathbb{A}})^2}{\widehat{\Delta}} \ln \frac{(j(t) + 1)k_{j(t)}}{\delta} + \sqrt{(j(t) + 1)(t - t_0) \ln \frac{(j(t) + 1)k_{j(t)}}{\delta}} \right).$$

We now distinguish between two cases. First, $j(t) = 0$, in which case

$$\mathrm{Reg}_{\mathbb{M}}([t_0 + 1, t], \{\widehat{\pi}\}) = \mathcal{O}\left( \frac{R(\Pi_{\mathbb{A}})^2}{\widehat{\Delta}} \ln \frac{R(\Pi_{\mathbb{A}})}{\Delta\delta} + \sqrt{k_0 \ln \frac{R(\Pi_{\mathbb{A}})}{\widehat{\Delta}\delta}} \right)$$

$$= \mathcal{O}\left( \frac{R(\Pi_{\mathbb{A}})^2}{\widehat{\Delta}} \ln \frac{R(\Pi_{\mathbb{A}})}{\widehat{\Delta}\delta} \right)$$

and the case where $j(t) > 0$, where $j(t) = \mathcal{O}(\ln(t - t_0)) = \mathcal{O}(\ln(t))$ and $k_{j(t)} \leq 2(t - t_0)$

$$\mathrm{Reg}_{\mathbb{M}}([t_0 + 1, t], \{\widehat{\pi}\}) = \mathcal{O}\left( \frac{R(\Pi_{\mathbb{A}})^2}{\widehat{\Delta}} \ln(t - t_0) \ln \frac{t - t_0}{\delta} + \sqrt{(t - t_0) \ln(t - t_0) \ln \frac{t - t_0}{\delta}} \right)$$

$$= \mathcal{O}\left( \frac{R(\Pi_{\mathbb{A}})^2}{\widehat{\Delta}} \ln(t) \ln \frac{t}{\delta} + \sqrt{(t - t_0) \ln(t) \ln \frac{t}{\delta}} \right).$$

Combining both cases gives

$$\mathrm{Reg}_{\mathbb{M}}([t_0 + 1, t], \{\widehat{\pi}\}) = \mathcal{O}\left( \frac{R(\Pi_{\mathbb{A}})^2}{\widehat{\Delta}} \left( \ln(t) \ln \frac{t}{\delta} + \ln \frac{R(\Pi_{\mathbb{A}})}{\widehat{\Delta}\delta} \right) + \sqrt{(t - t_0) \ln(t) \ln \frac{t}{\delta}} \right).$$

$\square$

**Lemma 41** (Regret in adversarial environments). *Assume the absolute constant in the length of the initial epoch $k_0$ and sampling probabilities $\rho^e$ of Algorithm 4 is chosen large enough. Then the regret of Algorithm 4 against $\Pi_{\mathbb{A}} \cup \{\widehat{\pi}\}$ is bounded with probability at least $1 - \mathcal{O}(\delta)$ for all rounds $t$ that the algorithm has not terminated yet as*

$$\mathrm{Reg}_{\mathbb{M}}([t_0 + 1, t], \Pi_{\mathbb{A}} \cup \{\widehat{\pi}\}) = \mathcal{O}\left( \frac{R(\Pi_{\mathbb{A}})^2}{\widehat{\Delta}} \left( \ln(t) \ln \frac{t}{\delta} + \ln \frac{R(\Pi_{\mathbb{A}})}{\widehat{\Delta}\delta} \right) + \sqrt{(t - t_0) \ln(t) \ln \frac{t}{\delta}} \right).$$

*Proof.* By combining Lemma 39 and Lemma 40, we have

$$\mathrm{Reg}_{\mathbb{M}}([t_0 + 1, t], \Pi_{\mathbb{A}} \cup \{\widehat{\pi}\})$$

$$= \mathcal{O}\left( \widehat{\Delta}k_0 + \frac{R(\Pi_{\mathbb{A}})^2}{\widehat{\Delta}} \left( \ln(t) \ln \frac{t}{\delta} + \ln \frac{R(\Pi_{\mathbb{A}})}{\widehat{\Delta}\delta} \right) + \sqrt{(t - t_0) \ln(t) \ln \frac{t}{\delta}} \right)$$

and by plugging in the definition of $k_0$, we get

$$\mathrm{Reg}_{\mathbb{M}}([t_0 + 1, t], \Pi_{\mathbb{A}} \cup \{\widehat{\pi}\}) = \mathcal{O}\left( \frac{R(\Pi_{\mathbb{A}})^2}{\widehat{\Delta}} \left( \ln(t) \ln \frac{t}{\delta} + \ln \frac{R(\Pi_{\mathbb{A}})}{\widehat{\Delta}\delta} \right) + \sqrt{(t - t_0) \ln(t) \ln \frac{t}{\delta}} \right).$$

$\square$

## D.4 Concentration Bounds in Stochastic Environments

**Lemma 42.** *In stochastic environments, the regret and pseudo-regret of any algorithm $\mathbb{A}$ against a policy class $\Pi'$ satisfy with probability at least $1 - \delta$ for all rounds $t \in \mathbb{N}$*

$$\text{PseudoReg}_{\mathbb{A}}(t, \Pi') - \text{Reg}_{\mathbb{A}}(t, \Pi') = \mathcal{O}\left(\sqrt{t \ln \frac{\ln t}{\delta}}\right). \tag{33}$$

*Proof.* Let $\pi_\star \in \text{argmax}_{\pi \in \Pi'} \mathbb{E}_{a \sim \pi, x \sim \mathcal{D}}[r(a, x)]$ be the best policy in $\Pi'$. Then

$$\text{PseudoReg}_{\mathbb{A}}(t, \Pi') - \text{Reg}_{\mathbb{A}}(t, \Pi')$$

$$= \sum_{\ell=1}^{t} \mathbb{E}_{a \sim \pi_\star, x \sim \mathcal{D}}[r(a, x)] - \sum_{\ell=1}^{t} \mathbb{E}_{a \sim \pi_\ell, x \sim \mathcal{D}}[r(a, x)]$$

$$- \left( \max_{\pi' \in \Pi'} \sum_{\ell=1}^{t} \mathbb{E}_{a \sim \pi'}[r_\ell(a, x_\ell)] - \sum_{\ell=1}^{t} r_\ell(a_\ell, x_\ell) \right)$$

$$= \sum_{\ell=1}^{t} r_\ell(a_\ell, x_\ell) - \sum_{\ell=1}^{t} \mathbb{E}_{a \sim \pi_\ell, x \sim \mathcal{D}}[r(a, x)]$$

$$+ \sum_{\ell=1}^{t} \mathbb{E}_{a \sim \pi_\star, x \sim \mathcal{D}}[r(a, x)] - \max_{\pi' \in \Pi'} \sum_{\ell=1}^{t} \mathbb{E}_{a \sim \pi'}[r_\ell(a, x_\ell)]$$

$$\leq \sum_{\ell=1}^{t} r_\ell(a_\ell, x_\ell) - \sum_{\ell=1}^{t} \mathbb{E}_{a \sim \pi_\ell, x \sim \mathcal{D}}[r(a, x)]$$

$$+ \sum_{\ell=1}^{t} \mathbb{E}_{a \sim \pi_\star, x \sim \mathcal{D}}[r(a, x)] - \sum_{\ell=1}^{t} \mathbb{E}_{a \sim \pi_\star}[r_\ell(a, x_\ell)] \qquad (\pi_\star \in \Pi')$$

$$\leq 2 \times 1.44 \sqrt{\max(2t, 2)\left(1.4 \ln \ln \left(2\left(\max\left(\frac{2t}{2}, 1\right)\right)\right)\right) + \ln \frac{5.2}{\delta}} \qquad \text{(Lemma 46)}$$

$$= \mathcal{O}\left(\sqrt{t \ln \frac{\ln t}{\delta}}\right).$$

Here, the last main step is to apply the time-uniform Hoeffding bound from Lemma 46 to the the two differences individually. Both are martingale sequences that are bounded, i.e., $r_\ell(a_\ell, x_\ell) - \mathbb{E}_{a \sim \pi_\ell, x \sim \mathcal{D}}[r(a, x)] \in [-1, 1]$ and $\mathbb{E}_{a \sim \pi_\star, x \sim \mathcal{D}}[r(a, x)] - \mathbb{E}_{a \sim \pi_\star}[r_\ell(a, x_\ell)] \in [-1, 1]$. $\qquad \square$

**Lemma 43.** *Let $t_0 \in \mathbb{N}$ be a possibly random time and let $\{\pi_\ell\}_{\ell \geq t_0}$ be a possibly random sequence of policies $\pi_\ell \colon \mathcal{X} \to \Delta_{\mathcal{A}}$ so that for all $\ell \geq t_0$, $\pi_\ell$ is independent of $\{(r_j, x_j)\}_{j \geq \ell}$ and $t_0$ is independent of $\{(r_j, x_j)\}_{j \geq t_0}$. Then with probability at least $1 - 2\delta$*

$$\left| \sum_{\ell=t_0+1}^{t} [\mathbb{E}_{a \sim \pi_\ell}[r_\ell(a, x_\ell)] - \mathbb{E}_{a \sim \pi_\ell, x \sim \mathcal{D}}[r(a, x)]] \right| = \mathcal{O}\left(\sqrt{(t - t_0) \ln \frac{\ln(t - t_0)}{\delta}}\right).$$

*Proof.* Define the sigma-algebra $\mathcal{F}_\ell = \sigma(\{r_j, x_j, \pi_j\}_{j \leq \ell} \cup \{\pi_\ell, t_0\})$ and let

$$Z_\ell = \begin{cases} 0 & \text{if } \ell \leq t_0 \\ \mathbb{E}_{a \sim \pi_\ell}[r_\ell(a, x_\ell)] - \mathbb{E}_{a \sim \pi_\ell, x \sim \mathcal{D}}[r(a, x)] & \text{otherwise.} \end{cases}$$

The sequence $\{Z_\ell\}_{\ell \in \mathbb{N}}$ is a martingale difference sequence w.r.t. $\mathcal{F}_\ell$ and $Z_\ell \in [-\mathbf{1}\{\ell > t_0\}, +\mathbf{1}\{\ell > t_0\}]$. As a result, we can apply Lemma 46 and get that

$$\sum_{\ell=t_0+1}^{t} [\mathbb{E}_{a \sim \pi_\ell}[r_\ell(a, x_\ell)] - \mathbb{E}_{a \sim \pi_\ell, x \sim \mathcal{D}}[r(a, x)]] = \sum_{\ell=1}^{t} Z_t$$

$$\leq 1.44 \sqrt{\max(2(t - t_0), 2)\left(1.4 \ln \ln \left(2 \left(\max\left(\frac{2(t - t_0)}{2}, 1\right)\right)\right) + \ln \frac{5.2}{\delta}\right)}$$

$$= \mathcal{O}\left(\sqrt{(t - t_0) \ln \frac{\ln(t - t_0)}{\delta}}\right)$$

holds for all $t$ with probability at least $1 - \delta$. Applying the same argument to $-Z_\ell$ and a union bound completes the proof. $\qquad \square$

Notice that, because contexts are i.i.d., a simple anytime Hoeffding bound implies the random variable $\mathrm{MaxRew}(t)$ (recall this quantity is defined as the maximum sum of pseudo-expectations over realized contexts) is larger than $t\mathbb{E}_{a \sim \pi_\star(\cdot|x), x \sim \mathcal{D}}[r(a, x)]$ (up to a factor of $\widetilde{\mathcal{O}}(\sqrt{t})$).

**Lemma 44.** *If the environment $\mathbb{B}$ is stochastic then with probability at least $1 - \delta$*

$$t\mathbb{E}_{a \sim \pi_\star(\cdot|x), x \sim \mathcal{D}}[r(a, x)] \leq \max_{\pi \in \Pi} \sum_{\ell=1}^{t} \mathbb{E}_{a \sim \pi}[r_\ell(a, x_\ell)] + \mathcal{O}\left(\sqrt{t \ln \frac{t}{\delta}}\right)$$

*for all $t \in \mathbb{N}$.*

*Proof.* Consider the martingale sequence $\{Z_\ell\}_{\ell=1}^{\infty}$ defined as $Z_\ell = \mathbb{E}_{a \sim \pi_\star(\cdot|x), x \sim \mathcal{D}}[r(a, x)] - \mathbb{E}_{a \sim \pi_\star(\cdot|x_t)}[r(a, x)|x_t]$. By definition $|Z_\ell| \leq 2$ for all $\ell$. Therefore by an anytime Hoeffding bound (Lemma 46 in Appendix E), with probability at least $1 - \delta$,

$$\sum_{\ell=1}^{t} Z_\ell = \mathcal{O}\left(\sqrt{t \ln \frac{t}{\delta}}\right)$$

for all $t \in \mathbb{N}$. Thus,

$$t\mathbb{E}_{a \sim \pi_\star(\cdot|x), x \sim \mathcal{D}}[r(a, x)] \leq \sum_{\ell=1}^{t} \mathbb{E}_{a \sim \pi_\star}[r_\ell(a, x_\ell)] + \mathcal{O}\left(\sqrt{t \ln \frac{t}{\delta}}\right)$$

$$\leq \max_{\pi \in \Pi} \sum_{\ell=1}^{t} \mathbb{E}_{a \sim \pi}[r_\ell(a, x_\ell)] + \mathcal{O}\left(\sqrt{t \ln \frac{t}{\delta}}\right)$$

$$\square$$

Lemma 44 implies that an algorithm that an algorithm that competes with $\mathrm{MaxRew}(t)$ in turn can compete against $t\mathbb{E}_{a \sim \pi_\star(\cdot|x), x \sim \mathcal{D}}[r(a, x)]$. This fact will help us argue that an adversarial algorithm has good performance in a stochastic environment.

Let's show that when the contexts are i.i.d., the counterfactual reward of any algorithm that decides what policy to play before the context is revealed is (up to a $\sqrt{t \ln \frac{t}{\delta}}$ factor) upper bounded by $t\mathbb{E}_{a \sim \pi_\star(\cdot|x), x \sim \mathcal{D}}[r(a, x)]$,

**Lemma 45.** *If the environment is stochastic with i.i.d. contexts and the algorithm decides on its policy $\pi_t$ before observing context $x_t$ then with probability at least $1 - \delta$,*

$$\sum_{\ell=1}^{t} \mathbb{E}_{a \sim \pi_\ell(\cdot|x_\ell)}[r(a, x_\ell)|x_\ell] \leq t\mathbb{E}_{a \sim \pi_\star(\cdot|x), x \sim \mathcal{D}}[r(a, x)] + \mathcal{O}\left(\sqrt{t \ln \frac{t}{\delta}}\right)$$

*for all $t \in \mathbb{N}$.*

*Proof.* Similar to the proof of Lemma 44, consider the martingale sequence $\{Z_\ell\}_{\ell=1}^\infty$ defined as $Z_\ell = \mathbb{E}_{a\sim\pi_\ell(\cdot|x_\ell)}[r(a,x_\ell)|x_\ell] - \mathbb{E}_{a\sim\pi_\ell(\cdot|x),x\sim\mathcal{D}}[r(a,x)]$. By definition $|Z_\ell| \leq 2$. Therefore by the anytime Hoeffding bound of Lemma 46, with probability at least $1-\delta$

$$\sum_{\ell=1}^t Z_\ell = \sum_{\ell=1}^t \mathbb{E}_{a\sim\pi_\ell(\cdot|x_\ell)}[r(a,x_\ell)|x_\ell] - \sum_{\ell=1}^t \mathbb{E}_{a\sim\pi_\ell(\cdot|x),x\sim\mathcal{D}}[r(a,x)] = \mathcal{O}\left(\sqrt{t\ln\frac{t}{\delta}}\right)$$

for all $t \in \mathbb{N}$. By definition $\pi_\star$ is the policy satisfying $\pi_\star = \operatorname{argmax}_\pi \mathbb{E}_{a\sim\pi,x\sim\mathcal{D}}[r(a,x)]$ and therefore for any $\pi_\ell$, the inequality $\mathbb{E}_{a\sim\pi_\ell,x\sim\mathcal{D}}[r(a,x)] \leq \mathbb{E}_{a\sim\pi_\star,x\sim\mathcal{D}}[r(a,x)]$ holds. $\qquad\square$

Lemma 45 implies that in the case of i.i.d. contexts a learner that selects a policy based on historical data (that is the learner selects a policy to play before the context is revealed) cannot do substantially better than playing $\pi_\star = \operatorname{argmax}_{\pi\in\Pi} \mathbb{E}_{a\sim\pi(\cdot|x),x\sim\mathcal{D}}[r(a,x)]$ during all timesteps. This will prove helpful when deriving bounds for the gap estimation phase.

# E  Additional Technical Lemmas

**Lemma 46** (Time-uniform Hoeffding bound). *Let $S_t = \sum_{i=1}^t Y_t$ be a martingale sequence w.r.t. some sigma algebra $\mathcal{F}_t$ and let $Y_t \in [a_t, b_t]$ almost surely for $a_t, b_t$ measurable in $\mathcal{F}_t$. Then with probability at least $1-\delta$ for all $t \in \mathbb{N}$*

$$S_t \leq 1.44\sqrt{\max(W_t, m)\left(1.4\ln\ln\left(2\left(\max\left(\frac{W_t}{m}, 1\right)\right)\right) + \ln\frac{5.2}{\delta}\right)}$$

*where $W_t = \sum_{i=1}^t \frac{(b_i-a_i)^2}{4}$ and $m > 0$ arbitrary but fixed.*

*Proof.* By entry "Hoeffding I" in Table 3 of Howard et al. [18], $S_t$ is a sub-$\psi_N$ process with variance process $W_t$. Further, by Proposition 2 in Howard et al. [19], this implies that it is also a sub-$\psi_P$ process with $c = 0$. We now apply Lemma 48 to achieve the desired result. $\qquad\square$

**Lemma 47** (Time-uniform Bernstein bound). *Let $S_t = \sum_{i=1}^t Y_i$ be a martingale sequence w.r.t. a sigma algebra $\mathcal{F}_t$ and let $Y_t \leq c$ a.s. for some parameter $c > 0$. Then with probability at least $1-\delta$ for all $t \in \mathbb{N}$*

$$S_t \leq 1.44\sqrt{\max(W_t, m)\left(1.4\ln\ln\left(2\left(\max\left(\frac{W_t}{m}, 1\right)\right)\right) + \ln\frac{5.2}{\delta}\right)}$$
$$+ 0.41c\left(1.4\ln\ln\left(2\left(\max\left(\frac{W_t}{m}, 1\right)\right)\right) + \ln\frac{5.2}{\delta}\right)$$

*where $W_t = \sum_{i=1}^t \mathbb{E}[Y_i^2|\mathcal{F}_i]$ and $m > 0$ is arbitrary but fixed.*

*Proof.* By entry "Bennett" in Table 3 of Howard et al. [18], $S_t$ is a sub-$\psi_P$ process with variance process $W_t$ and parameter $c = 0$. We now apply Lemma 48 to achieve the desired result. $\qquad\square$

**Lemma 48** (General concentration result). *In the terminology of Howard et al. [18], let $S_t = \sum_{i=1}^t Y_i$ be a sub-$\psi_P$ process with parameter $c = 0$ and variance process $W_t$. Then with probability at least $1-\delta$ for all $t \in \mathbb{N}$*

$$S_t \leq 1.44\sqrt{\max(W_t, m)\left(1.4\ln\ln\left(2\left(\max\left(\frac{W_t}{m}, 1\right)\right)\right) + \ln\frac{5.2}{\delta}\right)}$$
$$+ 0.41c\left(1.4\ln\ln\left(2\left(\max\left(\frac{W_t}{m}, 1\right)\right)\right) + \ln\frac{5.2}{\delta}\right)$$

*where $m > 0$ is arbitrary but fixed.*

*Proof.* This bound follows from Howard et al. [18] Theorem 1 with Equation 10 in that paper by setting $s = 1.4$ and $\eta = 2$. $\qquad\square$

**Lemma 49.** *Suppose $\{X_t\}_{t=1}^{\infty}$ is a martingale difference sequence with $|X_t| \leq b$. Let*

$$\mathrm{Var}_\ell(X_\ell) = \mathbf{Var}(X_\ell | X_1, \cdots, X_{\ell-1})$$

*Let $V_t = \sum_{\ell=1}^{t} \mathrm{Var}_\ell(X_\ell)$ be the sum of conditional variances of $X_t$. Then we have that for any $\delta' \in (0,1)$ and $t \in \mathbb{N}$*

$$\mathbb{P}\left( \sum_{\ell=1}^{t} X_\ell > 2\sqrt{V_t}A_t + 3bA_t^2 \right) \leq \delta' \,,$$

*where $A_t = \sqrt{2\ln\ln\left(2\left(\max\left(\frac{V_t}{b^2}, 1\right)\right)\right) + \ln\frac{6}{\delta'}}$.*

*Proof.* We are in a position to use 47 (with $c = b$). Let $S_t = \sum_{\ell=1}^{t} X_t$ and $W_t = \sum_{\ell=1}^{t} \mathrm{Var}_\ell(X_\ell)$. Let's set $m = b^2$. It follows that with probability $1 - \delta'$ for all $t \in \mathbb{N}$

$$
\begin{aligned}
S_t &\leq 1.44\sqrt{\max(W_t, b^2)\left(1.4\ln\ln\left(2\left(\max\left(\frac{W_t}{b^2}, 1\right)\right)\right) + \ln\frac{5.2}{\delta'}\right)} \\
&\quad + 0.41b\left(1.4\ln\ln\left(2\left(\max\left(\frac{W_t}{b}, 1\right)\right)\right) + \ln\frac{5.2}{\delta'}\right) \\
&\leq 2\sqrt{\max(W_t, b^2)\left(2\ln\ln\left(2\left(\max\left(\frac{W_t}{b^2}, 1\right)\right)\right) + \ln\frac{6}{\delta'}\right)} \\
&\quad + b\left(2\ln\ln\left(2\left(\max\left(\frac{W_t}{b^2}, 1\right)\right)\right) + \ln\frac{6}{\delta'}\right) \\
&= 2\max(\sqrt{W_t}, b)A_t + bA_t^2 \\
&\leq 2\sqrt{W_t}A_t + 2bA_t + bA_t^2 \\
&\overset{(i)}{\leq} 2\sqrt{W_t}A_t + 3bA_t^2 \,,
\end{aligned}
$$

where $A_t = \sqrt{2\ln\ln\left(2\left(\max\left(\frac{W_t}{b^2}, 1\right)\right)\right) + \ln\frac{6}{\delta'}}$. Inequality $(i)$ follows because $A_t \geq 1$.

Setting $V_t = W_t$ we conclude the proof. $\qquad\square$

In turn, a corollary of the previous lemma is the following.

**Lemma 50.** *Suppose $\{X_t\}_{t=1}^{\infty}$ is a martingale difference sequence with $|X_t| \leq b_t$ with $b_t$ a non-decreasing deterministic sequence. Let*

$$\mathrm{Var}_\ell(X_\ell) = \mathbf{Var}(X_\ell | X_1, \cdots, X_{\ell-1})$$

*Let $V_t = \sum_{\ell=1}^{t} \mathrm{Var}_\ell(X_\ell)$ be the sum of conditional variances of $X_t$. Then for any $\delta \in (0,1)$ and $t \in \mathbb{N}$ we have*

$$\mathbb{P}\left( \sum_{\ell=1}^{t} X_\ell > 2\sqrt{V_t}A_t + 3B_tA_t^2 \right) \leq \delta \,,$$

*where $A_t(\delta) = 2\sqrt{\ln\frac{12t^2}{\delta}}$.*

*Proof.* For any $t$ define a martingale difference sequence $\{X_\ell^{(t)}\}_{\ell=1}^{\infty}$ as follows:

$$X_\ell^{(t)} = \begin{cases} X_\ell & \text{if } \ell \leq t \\ 0 & \text{otherwise} \end{cases}$$

We apply Lemma 49 with parameter $\delta' = \frac{\delta}{2t^2}$ and $b = B_t$, and then overapproximate. A union bound over $t \in \mathbb{N}$ gets the desired result. $\qquad\square$

**Lemma 51.** *Suppose $\{X_t\}_{t=1}^{\infty}$ is a martingale difference sequence with $|X_t| \le b_t$ with $b_t$ a non-decreasing deterministic sequence. Let*

$$\mathrm{Var}_\ell(X_\ell) = \mathbf{Var}(X_\ell | X_1, \cdots, X_{\ell-1})$$

*Let $V_t = \sum_{\ell=1}^{t} \mathrm{Var}_\ell(X_\ell)$ be the sum of conditional variances of $X_t$. Then we have that for any $\delta \in (0,1)$ and $t \in \mathbb{N}$*

$$\mathbb{P}\left(\sum_{\ell=1}^{t} X_\ell > 4\sqrt{V_t \ln \frac{12t^2}{\delta}} + 12b_t \ln \frac{12t^2}{\delta}\right) \le \delta,$$

*Proof.* Notice that by definition $V_t(\mathbf{a}) \le t b_t^2$. Therefore $\max(\frac{V_t(\mathbf{a})}{b_t^2}, 1) \le t$ and

$$A_t(\delta') \le \sqrt{2 \ln \ln 2t + \ln \frac{12t^2}{\delta'}} \le 2\sqrt{\ln \frac{12t^2}{\delta'}} := \tilde{A}_t(\delta').$$

Substituting this upper bound in the statement of Lemma 50 yields the result. $\qquad \square$

**Lemma 52.** *Let $c, \delta \in (0,1]$ and $t \ge \frac{16}{c^2} \ln^2 \frac{2}{c\delta}$. Then*

$$\frac{\ln \frac{t}{\delta}}{\sqrt{t}} \le c$$

*Proof.* The function $f(t) = \frac{\ln \frac{t}{\delta}}{\sqrt{t}}$ is non-increasing on $(e^2 \delta, \infty)$. Thus, it is sufficient to show that the inequality holds for $t = 16 \frac{\ln^2 \frac{2}{c\delta}}{c^2}$. Since $\ln(x) \le \frac{x}{2}$ for all $x \in \mathbb{R}_+$, we can upper-bound this value of $t$ as

$$t \le \frac{16}{c^4 \delta^2}.$$

This implies

$$\ln \frac{t}{\delta} \le \ln \frac{16}{c^4 \delta^4} = 4 \ln \frac{2}{c\delta} = c\sqrt{16 \frac{\ln^2(2/c/\delta)}{c^2}} = c\sqrt{t}$$

which proves the claim. $\qquad \square$

**Lemma 53.** *Let $c, \delta \in (0,1]$ and $t \ge \frac{4}{c} \ln \frac{2}{c\delta}$. Then*

$$\frac{\ln \frac{t}{\delta}}{t} \le c$$

*Proof.* The function $f(t) = \frac{\ln \frac{t}{\delta}}{t}$ is non-increasing on $(e\delta, \infty)$. Thus, it is sufficient to show that the inequality holds for $t = 4 \frac{\ln \frac{2}{c\delta}}{c}$. Since $\ln(x) \le \frac{x}{2}$ for all $x \in \mathbb{R}_+$, we can upper-bound this value of $t$ as

$$t \le \frac{4}{c^2 \delta}.$$

This implies

$$\ln \frac{t}{\delta} \le \ln \frac{4}{c^2 \delta^2} = 2 \ln \frac{2}{c\delta} = \frac{c}{2} t \le ct$$

which proves the claim. $\qquad \square$