# OpenReview forum: "Best of Both Worlds Model Selection"
_NeurIPS.cc/2022/Conference — NeurIPS 2022 Accept_

### Official Review · Reviewer_1K6U · 2022-07-08

**Rating:** 7
**Confidence:** 3
**Soundness:** 3 good
**Presentation:** 2 fair
**Contribution:** 3 good

**Summary:**

This work considers model selection from a set of contextual bandit algorithms with nested policy classes. Each of these base learners has a known regret guarantee, that may or may not hold. The environment may be either stochastic with a reward gap or adversarial, and a best-of-both-worlds high-probability bound is provided for the proposed meta-algorithm. On each round this meta-algorithm chooses a base learner with probability that depends on their known bounds. It then performs mis-specification tests and eliminates the learners that violate their regret bounds. Additionally, it performs a test to reliably detect the stochastic environment and switch to exploitation mode in this case. An important building block is extension of the policy class of each of the base learners to include special actions that delegate the decision to another base learner (with a larger policy class). This allows the elimination test to work in the adversarial environment. In the case that the environment is known to be adversarial, the algorithm can run without the stochastic detection test, in which case it achieves a tighter bound. This bound is a high-probability version of the one known from prior art.

**Questions:**

Can you provide an example of contextual base algorithms (that actually use the context) that are suitable for this framework?

**Limitations:**

The assumptions are clearly stated. A lower bound for stochastic case is provided.

**Strengths And Weaknesses:**

**Originality** To the best of my knowledge, this is the first work to provide best-of-both-worlds guarantees for bandit model selection. It extends the idea of regret balancing to the adversarial environment by modification of the base learners and carefully designed tests.

**Quality** The proofs are provided in the appendix, but I did not verify their soundness. The proof sketches in the main paper seem reasonable. The dependence of the stochastic case bound on the complexity of the largest policy class is justified by proving a lower bound.

**Clarity** This work is highly theoretical, and therefore it is not surprising that it is not easy to understand. However, some improvements to the clarity and the organization would be helpful. For example, the detailed example in lines 46-65 is too technical to be part of the introduction, and is better suited for section 2. Subsection 2.1 feels out of scope for section 2. The extension of the policy space and linking of the base learner performances need more explanation. For example, before the definition of extendability, linking of the learners can be mentioned to explain how the extendability will be used by the meta-algorithm.

**Significance** Demonstration of best-of-both-worlds bounds is important for the understanding of the underlying relations between different settings and eventually designing practical algorithms that can optimally exploit the properties of the environment. On the other hand, switching between operation modes based on tests is intuitively less generic than a framework that tunes its parameters continuously.

Some more discussion on future directions would help to realize the significance of this work. Can it be potentially generalized to other partial information settings? Can some additional assumptions improve the stochastic regret bound?

---

> ### Author Response · Authors · 2022-08-02
> **Response to Reviewer 1K6U**
>
> We want to start by thanking the reviewer for their comments. We will take the reviewer's suggestions regarding the presentation of our manuscript in serious consideration. We will add a pictorial illustration of the different algorithmic components we introduce, including linking and policy space augmentation.
>
> - ``On the other hand, switching between operation modes based on tests is intuitively less generic than a framework that tunes its parameters continuously."
>
> We would like to gently push back against this statement. Regret balancing is a very general technique, the details of which should, of course, be adapted to the specific scenario. One advantage of regret balancing over mirror descent master algorithms from other works such as [3] is interpretability. The same is true for the gap detection tests and exploitation play schedule of Arbe-Gap and Arbe-Exploit, our best-of-both-worlds algorithms. This is very different from [3], where the best of both worlds guarantees for linear bandits are derived using a play schedule based on mirror descent. Besides, regret balancing lends itself to a high probability analysis which is generally harder to obtain with mirror descent.
>
> - ``Some more discussion on future directions would help to realize the significance of this work. Can it be potentially generalized to other partial information settings? Can some additional assumptions improve the stochastic regret bound?"
>
> We believe the significance of this work lies along two axes: 1) we explore in depth the problem of model selection with best-of-both-worlds guarantees. We have documented the level of detail of our results in the main section of this rebuttal. 2) We introduce a variety of novel techniques. In particular this work shows the applicability of the regret balancing in the setting of adversarial bandits. This is important because regret balancing is a very simple and interpretable technique. We hope this can spur the use of these types of algorithms beyond stochastic domains. We believe this could have important consequences in the study of nonstationary bandits and other similar problems. We will add to this effect some comments to the concluding section of the paper.
>
> - ``Can you provide an example of contextual base algorithms (that actually use the context) that are suitable for this framework?"
>
> The Exp4 algorithm (Auer et al., 2003) in contextual bandits with finite action sets. Please recall that this algorithm operates with policies mapping contexts to probability distributions over actions.

---

> > ### Comment · Reviewer_1K6U · 2022-08-08
> > **Response to the rebuttal**
> >
> > Thank you for the response that addresses most of my concerns.
> >
> > Concerning Exp4 – indeed I agree that the expert advice can be seen as the context. Can you please clarify this for the readers (e.g. emphasize it more strongly near line 126)?
> >
> > To summarize, my remaining concerns are clarity and future research directions, both of which you promise to improve.
> >
> > Thank you.

---

> > > ### Author Response · Authors · 2022-08-09
> > > **Follow up**
> > >
> > > Thanks so much for your comments. We will certainly update the manuscript to include the reviewer's suggestions in the camera ready.

---

### Official Review · Reviewer_FmBH · 2022-07-12

**Rating:** 7
**Confidence:** 4
**Soundness:** 3 good
**Presentation:** 4 excellent
**Contribution:** 3 good

**Summary:**

This work studies the problem of model selection with bandit feedback in the presence of a sequence of policy classes, in other words, a monotonically increasing sequence of sets of polices. The goal is to achieve the "best of both worlds" high-probability guarantee between the stochastic world and the adversarial world, in other words, ensuring $polylog(T)$ regret for stochastic setting and $\sqrt{T}$ regret for the adversarial setting. It follows similar ideas of Lee et al. (2021) and Wei et al. (2021) and applies the idea of regret balancing technique discussed in Cutkosky et al. (2021) to design the algorithm Arbe for model selection. To achieve this, the authors extend the techniques to adversarial rewards: they have a few necessary constraints on the algorithms, and enhance the test for mis-specification.

**Questions:**

1. The idea is very similar to the pervious work of Lee et al. (2021) and Wei et al. (2021). I wonder if the authors could emphasize the specific adjustment they made to achieve the best of both worlds guarantee in this setting and what's the specific difficulty this new problem has.

2. Would like to confirm that line 6 of Arbe-GapExploit means that the whole algorithm would only deal with "adversarial" setting and would not go to ArbeGapExploit phase anymore.

3. Can we extend this approach to achieve results in an intermediate setting (e.g., corruption setting)?

**Limitations:**

None.

**Strengths And Weaknesses:**

This work provides the first algorithm which achieves the best of both worlds high-probability guarantees for model selection within (linear) bandit scenarios. The problem is well-defined and very general (for h-stability and extendability). The techniques which are further improved from the previous work of Lee et al. (2021) and Wei et al. (2021) following the idea of regret balancing are very interesting, and may be applied in solving other problems.

Overall, this work does not have any other specific weakness. The paper is clearly written and easy to understand.

---

> ### Author Response · Authors · 2022-08-02
> **Response to Reviewer FmBH**
>
> We want to start by thanking the reviewer for their comments. We will address the reviewer's comments one by one.
>
> - ``This work provides the first algorithm ... within (linear) bandit scenarios".
>
> Our work is $not$ restricted to the adversarial linear bandit setting. Although we use linear bandits as a running example and derive corollaries for model selection and best-of-both-worlds guarantees for linear bandits, all of our theorems hold in much more generality. Please see Section 2 of our manuscript.
>
> - Relationship between our work and Lee et al. (2021) [3] and Wei et al (2021) [4].
>
> As the reviewer has noticed, our work bears some relationship with these two papers. We address them separately.
>
> * Lee et al. (2021) [3]: We share the goal of providing an answer to the best-of-both-worlds problem for linear bandits (main goal for them, corollary for us). Yet, the similarities with that work stop there:
> Our algorithm is much more general than theirs, as it applies to settings beyond linear bandits, and it also achieves model selection guarantees. The onus of our work is in the combination of model selection and best of both worlds guarantees which is not addressed at all by Lee at al. (2021). In fact, our meta algorithms for model selection and best of both worlds are completely different than those in that work, since they are not specialized to just adversarial linear bandits problems. The same is true of the gap detection tests and exploitation play schedule of Arbe-Gap and Arbe-Exploit, our best of both worlds algorithms. This is very different from [3], where the best of both worlds guarantees for linear bandits are derived using a play schedule that is based on mirror descent. Finally, our model selection algorithms are inspired by the regret balancing principle and as a simple elimination algorithm perhaps more interpretable than the mirror descent master algorithms from other works such as [3].
>
> * Wei et al (2021) [4]: Both Wei at al. and our work follow the principle of regret balancing. Therefore, the meta-algorithms are naturally quite similar. Yet, importantly our work applies to the fully adversarial setting with $\sqrt T$ regret while the corruption-robust approach by [4] suffers a linear in $T$ regret if the world is truly fully adversarial. Besides, there are several important technical challenges that we needed to solve that are not present in the corruption setting. Most importantly, the corruption setting can for the most part be handled similar to the stochastic setting (for instance, this does not need linking base learners). In our work we have to link base learners so as to avoid that the adversary fools the meta-algorithm. This may happen when the meta-learner eliminates adapted base learners and incur too much regret because another learner has negative regret. Since [4] measures regret always w.r.t. to the uncorrupted environment, by definition, no learner can achieve negative regret, hence this issue cannot show up in their work. Note that there is no notion of an uncorrupted environment in our setting. Finally, our work deals with general model selection setting (e.g., nested feature classes) while [4] is specific to model selection of the corruption amount.
>
> - ``Would like to confirm that line 6 of Arbe-GapExploit means ...  not go to ArbeGapExploit phase anymore."
>
> This is correct. Once the algorithm detects the environment is not stochastic, it plays Arbe forever, and will not go back into the exploit or gap detection phases anymore. Note that the algorithm will only reach this line if, at some point, there was a statistically significant gap between a policy and all others and this gap disappeared later. This is not possible in the stochastic setting. It is therefore safe to simply proceed with Arbe, the model selection approach for the adversarial setting.
>
> - "Can we extend ... setting (e.g., corruption setting)?"
>
> This is indeed an interesting but also subtle question. First note that, roughly speaking, when the corruption amount $C$ is $\Omega(\sqrt{T})$ then our fully-adversarial results are tighter than those achieved by corruption-robust algorithms which achieve $\min\{\sqrt{T}, \frac{\ln(T)}{\Delta}\} + C$. When the corruption $C$ is $o(\sqrt{T})$, then one might hope to achieve $o(\sqrt{T})$ regret. While model selection of the corruption amount itself is possible in this setting (e.g., [4]), there are lower bounds for general model selection (as we do in this work) in this $o(\sqrt{T})$ regime. Our result in Appendix A can be extended to show this, see also Pacchiano et al 2020. Developing corruption-robust algorithms with general model selection in the entire corruption regime would require additional assumptions, but it certainly is an interesting future direction.

---

### Official Review · Reviewer_Bk6N · 2022-07-13

**Rating:** 6
**Confidence:** 3
**Soundness:** 3 good
**Presentation:** 3 good
**Contribution:** 3 good

**Summary:**

The authors study the best-of-both-worlds guarantees for model selection in linear bandits. The paper proposes a novel algorithm called Arbe, which achieves the first high probability regret bound for adversarial model selection. In addition, the paper proposes a two-stage algorithm called Arbe-Gap, which achieves best-of-both-worlds high probability regret bounds.

**Questions:**

1. Can the proposed algorithms be extended to the infinite arm setting? If not, what is the main difficulty?
2. Does Arbe-Gap work for the stochastic rewards with adversarial corruptions?

**Limitations:**

Since the paper does not provide any empirical comparisons, it is still unknown whether the proposed algorithms are practical.

**Strengths And Weaknesses:**

Strength:

1. The writing is mostly clear.
2. The best-of-both-worlds guarantees for model selection is an interesting and important problem.
3. The proposed algorithms are novel. It is the first algorithm that achieves high probability best-of-both-worlds regret bounds for linear bandits with nested model classes.

Weakness:

1. Both stochastic and adversarial regret bounds are suboptimal with respect to the dimension of the reward vectors.
2. The paper does not contain numerical experiments for the proposed algorithms.

---

> ### Author Response · Authors · 2022-08-02
> **Response to Reviewer Bk6N**
>
> We want to start by thanking the reviewer for their comments. We will address the reviewer's questions one by one.
>
> - ``The authors study the best-of-both-worlds guarantees for model selection in linear bandits."
>
> Our work is $not$ restricted to linear bandits. Although we use linear bandits as a running example and derive corollaries for model selection and best-of-both-worlds guarantees for linear bandits as a consequence of our main results, all of our theorems hold in much more generality. Please see Section 2 of our manuscript.
>
> - ``Both stochastic and adversarial regret bounds are suboptimal with respect to the dimension of the reward vectors."
>
> Happily this is not true. The regret guarantee of Arbe is minimax optimal w.r.t. to the complexity parameter $R(\Pi_\star)$ in the adversarial case. It is not possible to achieve a model selection rate scaling linearly with $R(\Pi_\star)$. The work of [2] (Theorem 4 therein -- this requires some parsing by the reader) shows the best one can hope for is a quadratic dependence on $R(\Pi_\star)$. In the setting of adversarial linear bandits this means a rate with a multiplier of the form $d_*\log(\mathcal{A})$ instead of $\sqrt{d_*\log(\mathcal{A})}$ and in the case of EXP4 learners this means a rate with a multiplier of the form $\log(\Pi_\star)$ instead of $\sqrt{\log(\Pi_\star)}$.
>
> In the case of stochastic environments, our lower bounds show that having a regret scaling with $R^2(\Pi_M)$ is generally unavoidable. As anticipated in the general response section, we have uploaded a slightly more refined version of our paper (both main text and supplemental), where minor modifications to Algorithm 2 have been made. This new version comes with an improved Theorem 5 featuring in the main term, i.e., the one depending on the time horizon, a dependence on $R^2(\Pi_M)$.
>
>
> - ``Can the proposed algorithms be extended to the infinite arm setting? If not, what is the main difficulty?"
>
> Our results are not specific to any finite or infinite action space. In fact, our results for model selection with Arbe in the setting of adversarial linear bandits also holds for the infinite action case. In this setting the $\log(\mathcal{A})$ dependence on the regret bound can be substituted by $d$ via a standard covering argument. Similarly, our best-of-both-worlds algorithms do not require the action sets to be finite but do require the policy gap assumption to hold, something that in the setting of linear bandits is more plausible in the finite arm regime. In summary, the examples given in Sect. 2 are just examples, our technique is not restricted to finite action spaces.
>
> - ``Does Arbe-Gap work for the stochastic rewards with adversarial corruptions?"
>
> We believe it should be possible to adapt Arbe-Gap to work in that setting by treating corruption as adversarial but not in an obvious manner. It sounds like an exciting avenue for future work! Please see also the response to Reviewer FmBH.
>
>
> - ``Empirical comparisons."
>
> We understand this concern, some experiments are planned for the near future. Yet, we consider the main contributions of our work to be of a theoretical nature, and we believe the significance of our work should be be viewed as such.

---

> > ### Comment · Reviewer_Bk6N · 2022-08-07
> > **Optimality of the linear bandit case**
> >
> > Thank you for your detailed response which addresses some of my concerns. But I'm still confused about the optimality of the linear bandit case. In the adversarial setting, we hope to achieve a regret of the form $\tilde{O}(d_* \sqrt{t})$. Am I right? However, in line 265, the paper shows that  Arbe achieves a regret bound of order $\tilde{O}(d_* ^2\sqrt{t})$, which seems suboptimal.

---

> > > ### Author Response · Authors · 2022-08-09
> > > **Follow up**
> > >
> > > Thank you for your comments. We will further clarify the regret rate we can hope for in the linear bandit setting below and in the next revision of the paper. In the absence of model selection, the reviewer is correct that we hope to achieve $\tilde O(d_\star \sqrt{T})$ regret in the adversarial linear bandit setting with very many ($\Omega(2^{d_\star})$) or infinitely many arms. However, there exist lower bounds that suggest that there is a price to pay for model selection and the bound $\tilde O(d_\star \sqrt{T})$ is too much to hope for when $d_\star$ is unknown.
> > >
> > > Specifically, [1] proved a lower bound for the related setting of contextual bandits with $K$ arms where $\tilde O(\sqrt{\ln |\Pi_\star| K T})$ is achievable without model selection, when $\Pi_\star$ is known. Their Theorem~4 shows that if $\Pi_\star$ is unknown, then $\tilde O(\sqrt{\ln |\Pi_\star| K T})$ is unachievable in general and one can only hope to get $\tilde O(\ln |\Pi_\star| \sqrt{ K T})$ (choose $|\Pi_1| = \Theta(1)$ and $\mathfrak C = \Theta(\sqrt{K})$ in their result). Transferring their proof ideas to the linear bandit setting with very many arms suggests that $\tilde O(d_\star \sqrt{T})$ is not achievable there, and we conjecture that only the $\tilde O(d_\star^2 \sqrt{T})$ rate of Arbe is attainable in that case.
> > >
> > > However, we would like to emphasize that our main results are not specific to the linear bandit setting, which is why we did not pursue the direction of formally proving a lower bound for that specific setting. The linear results we present in this work are merely a corollary of our main findings, not the focus of our submission. We will remove that sentence in the abstract that makes it sound otherwise. That being said, as a corollary of our work, we derive the first theoretical results that achieve best of both worlds guarantees while performing, in particular, model selection in adversarial linear bandit scenarios.

---

> > > > ### Comment · Reviewer_Bk6N · 2022-08-09
> > > > **Thanks for the response**
> > > >
> > > > Thank you for the detailed explanation. I'll raise my rating to 6.

---

### Author Response · Authors · 2022-08-02
**General Response**

We want to thank the reviewers for their insightful comments. We will make sure we incorporate this feedback in the final version of our manuscript. First, we want to emphasize that in this work we have the first and rather complete answer to the problem of model selection with best of both worlds guarantees. We now explain why by going over the main results of this submission.

1) We introduce the Arbe algorithm for high probability model selection. This is the first high probability model selection result for adversarial contextual bandit algorithms. Arbe satisfies a high probability regret guarantee as long as each of the base algorithms also satisfies one. Arbe takes inspiration from the balancing and misspecification tests from the Regret Bound Balancing and Elimination algorithm of [1] designed for stochastic contextual bandit (and RL) scenarios, but is not just a corollary of those results. A fair amount of technical hurdles had to be overcome in Arbe's design to make it usable with adversarial base learners. For example, the misspecification test in [1] cannot work as is in an adversarial scenario. This is because, unlike the stochastic case, in the adversarial case the collected reward from any learner may be substantially larger than that of any fixed baseline. This is why in Arbe low complexity learners have to be augmented with an action that allows them to call more complex models every once in a while. The model selection rates that our algorithm achieves are minimax optimal in their dependency on the complexity of the optimal model class as shown by the lower bounds of [2]. See also Item 3 below, as well as the response to Reviewer Bk6N.

2) We introduce the first algorithm for best-of-both-worlds model selection. By leveraging the high probability guarantees of the Arbe algorithm, we develop the first best-of-both-worlds model selection algorithm (Arbe-Gap + Arbe-Exploit) that can retain model selection rates when the environment is adversarial, and obtain logarithmic rates when the world is stochastic with a gap. The logarithmic gap dependent rate of this algorithm exhibits an optimal quadratic dependence w.r.t the largest policy class $R(\Pi_M)$ (see below the response to Reviewer Bk6N). Our algorithm requires a couple of innovations, first a careful design of a gap identification subroutine aimed at identifying a candidate optimal policy and gap estimator (Arbe-Gap) and, second, a very precise schedule of play for exploiting this knowledge and test its truthfulness (Arbe-Exploit). The algorithm is quite intricate and required us to overcome a variety of technical hurdles. These are better explained in the submission text.

3) We show via a lower bound for stochastic environments that, in the presence of a gap, perfect model selection between multiple logarithmic rate learners is impossible. This can be found in Appendix A (line 517 therein should instead read $\frac{R^2(\Pi_\star)\log T}{\Delta}$ -- sorry for the bad typo).
Thus the optimal dependence in the complexity parameter of the largest policy class must be quadratic.
Our algorithms (Arbe-Gap + Arbe-Exploit) achieve exactly this dependence on the complexity of the largest class when the environment is stochastic and has a gap.
To this effect, {\em we have uploaded a slightly refined version of our paper, where we made minor modification to Algorithm 2, with an improved version of Theorem 5, where the main term now depends on $R^2(\Pi_M)$.

Again, we want to emphasize that although our work is related to both [3] and [4], it is not a direct consequence of any of these works.
There was a considerable amount of technical difficulty in making each of our contributions, and we hope we have further clarified that in this rebuttal.

[1] Cutkosky, Dann, Das, Gentile, Pacchiano, Purohit, ``Dynamic Balancing for Model Selection in Bandits and RL", ICML 2021.

[2] Marinov, Zimmert, ``The Pareto Frontier of model selection for general Contextual Bandits", Neurips 2021.

[3] Lee, Luo, Wei, Zhang, Zhang, ``Achieving Near Instance-Optimality and Minimax-Optimality in Stochastic and Adversarial Linear Bandits Simultaneously", ICML 2021.

[4] Wei, Dann, Zimmert, ``A Model Selection Approach for Corruption Robust Reinforcement Learning", ALT 2022.

---

### Meta-Review · Area_Chair_H1R1 · 2022-08-26

**Recommendation:** Accept
**Confidence:** Certain

**Metareview:**

This work advances the direction on model selection for bandit problems with nested model classes. Reviewers all agree that the results are significant, the contribution is solid, and the paper is well written. Clear accept.

**Award:**

No

---

### Decision · Program_Chairs · 2022-09-14

Accept